# Refined Generalization Analysis of the Deep Ritz Method and Physics-Informed Neural Networks

## Abstract

In this paper, we derive refined generalization bounds for the Deep Ritz Method (DRM) and Physics-Informed Neural Networks (PINNs). For the DRM, we focus on two prototype elliptic partial differential equations (PDEs): Poisson equation and static Schrödinger equation on the $d$-dimensional unit hypercube with the Neumann boundary condition. Furthermore, sharper generalization bounds are derived based on the localization techniques under the assumptions that the exact solutions of the PDEs lie in the Barron spaces or the general Sobolev spaces. For the PINNs, we investigate the general linear second order elliptic PDEs with Dirichlet boundary condition using the local Rademacher complexity in the multi-task learning setting. Finally, we discuss the generalization error in the setting of over-parameterization when solutions of PDEs belong to Barron space.

## 1 Introduction

Partial Differential Equations (PDEs) play a pivotal role in modeling phenomena across physics, biology and engineering. However, solving PDEs numerically has been a longstanding challenge in scientific computing. Classical numerical methods like finite difference, finite element, finite volume and spectral methods may suffer from the curse of dimensionality when dealing with high-dimensional PDEs. Recent years, the remarkable successes of deep learning in diverse fields like computer vision, natural language processing and reinforcement learning have sparked interest in applying machine learning techniques to solve various types of PDEs. In fact, the idea of using machine learning to solve PDEs dates back to the last century (Lagaris et al., 1998), but it has recently gained renewed attention due to the significant advancements in hardware technology and the algorithm development.

There are numerous methods proposed to solve PDEs using neural networks. One popular method, known as PINNs (Raissi et al., 2019), utilizes neural network to represent the solution and enforces the neural network to satisfy the PDE constraints, initial conditions and boundary conditions by encoding these conditions into the loss function. The flexibility and scalability of the PINNs make it a widely used framework for addressing PDE-related problems. The Deep Ritz method (Yu et al., 2018), on the other hand, incorporates the variational formulation into training the neural networks due to the widespread use of the variational formulation in traditional methods. In comparison to PINNs, the form of DRM has a lower derivative order, but the fact that not all PDEs have variational forms limits its applications. Both methods hinge on the approximation ability of the deep neural networks.

The approximation power of feed-forward neural networks (FNNs) with diverse activation functions has been studied for different types of functions, including smooth functions (Lu et al., 2021a), continuous functions (Shen et al., 2022), Sobolev functions (Belomestny et al., 2023; Yang et al., 2023b;a; Yarotsky, 2017), Barron functions (Barron, 1993). It was proven in the last century that a sufficiently large neural network can approximate a target function in a certain function class with any given tolerance. Specifically, it has been shown in Hornik (1991) that the two-layer neural network with ReLU activation function is a universal approximator for continuous functions. More recently, specific approximate rate of neural networks has been shown for different function classes in terms of depth and width. Lu et al. (2021a) showed that a ReLU FNN with width

$\mathcal{O}(N \log N)$ and depth $\mathcal{O}(L \log L)$ can achieve approximation rate $\mathcal{O}(N^{-2s/d}L^{-2s/d})$ for the function class $C^s([0, 1]^d)$ in the $L^\infty$ norm, which is nearly optimal. In the context of applying neural networks to solve PDEs, the focus shifts to the approximation rates in the Sobolev norms. Belomestny et al. (2023) utilized multivariate spline to derive the required depth, width, and sparsity of a ReLU$^2$ deep neural network to approximate any Hölder smooth function in Hölder norms with the given approximation error. And the weights of the neural network are also controlled, which is essential to derive generalization error. Yang et al. (2023b) derived the nearly optimal approximation results of deep neural networks in Sobolev spaces with Sobolev norms. Specifically, deep ReLU neural networks with width $\mathcal{O}(N \log N)$ and depth $\mathcal{O}(L \log L)$ can achieve approximation rate $\mathcal{O}(N^{-2(n-1)/d}L^{-2(n-1)/d})$ for functions in $W^{n,\infty}((0, 1)^d)$ with $W^{1,\infty}$ norm. For higher order approximation in Sobolev spaces, Yang et al. (2023a) introduced deep super ReLU networks for approximating functions in Sobolev spaces under Sobolev norms $W^{m,p}$ for $m \in \mathbb{N}$ with $m \geq 2$. The optimality was also established by estimating the VC-dimension of the function class consisting of higher-order derivatives of deep super ReLU networks.

In this work, we focus on the DRM and PINNs, aiming to derive sharper generalization bounds. Compared to Jiao et al. (2021); Duan et al. (2021b), the localized analysis utilized in this paper leads to improved generalization bounds. We believe that this study provides a unified framework for deriving generalization bounds for methods that solve PDEs involving machine learning.

## 1.1 RELATED WORKS

**Deep learning based PDE solvers:** Solving high-dimensional PDEs has been a long-standing challenge in scientific computing due to the curse of dimensionality. Inspired by the ability and flexibility of neural networks for representing high dimensional functions, numerous studies have focused on developing efficient deep learning-based PDE solvers. In recent years, the PINNs have emerged as a flexible framework for addressing problems related to PDEs and have achieved impressive results in numerous tasks. Despite their success, there are areas where further improvements can be made, such as developing better optimization targets (Chiu et al., 2022) and neural network architectures (Ren et al., 2022; Zhang et al., 2020). Inspired by the use of weak formulation in traditional solvers, Zang et al. (2020) proposed to solve the weak formulation of PDEs via an adversarial network and the DRM (Yu et al., 2018) trains a neural network to minimize the variational formulations of PDEs.

**Fast rates in machine learning:** In statistical learning, the excess risk is expressed as the form $(\frac{\mathrm{COMP}_n(\mathcal{F})}{n})^\alpha$, where $n$ is the sample size, $\mathrm{COMP}_n(\mathcal{F})$ measures the complexity of the function class $\mathcal{F}$ and $\alpha \in [\frac{1}{2}, 1]$ represents the learning rate. The slow learning rate $\frac{1}{\sqrt{n}}$ ($\alpha = \frac{1}{2}$) can be easily derived by invoking Rademacher complexity (Bartlett & Mendelson, 2002), but achieving the fast rate $\frac{1}{n}$ ($\alpha = 1$) is much more challenging. Based on localization techniques, the local Rademacher complexity (Bartlett et al., 2005; Koltchinskii, 2006) was introduced to statistical learning and has become a popular tool to derive fast rates. It has been successfully applied across a variety of tasks, like clustering (Li & Liu, 2021), learning kernels (Cortes et al., 2013), multi-task learning (Yousefi et al., 2018), empirical variance minimization (Belomestny et al., 2017), among others. Variants of Rademacher complexity, such as shifted Rademacher complexity (Zhivotovskiy & Hanneke, 2018) and offset Rademacher complexity (Liang et al., 2015), also offer a potential direction for achieving the fast rates (Duan et al., 2023; Kanade et al., 2022; Yang et al., 2019). In this paper, our results are based on the localized analysis in Bartlett et al. (2005); Koltchinskii (2006; 2011).

**Generalization bounds for machine learning based PDE solvers:** Based on the probabilistic space filling arguments (Calder, 2019), Shin et al. (2020) demonstrated the consistency of PINNs for the linear second order elliptic and parabolic type PDEs. An abstract framework was introduced in Mishra & Molinaro (2022) and stability properties of the underlying PDEs were leveraged to derive upper bounds on the generalization error of PINNs. Following similar methods widely used in machine learning for deriving generalization bounds, the convergence rate of PINNs was derived in Jiao et al. (2021) by decomposing the error and estimating related Rademacher complexity. For the DRM, when the solutions are in the spectral Barron space, Lu et al. (2021c) demonstrated the generalization error bounds of two-layer neural networks for solving the Poisson equation and static Schrödinger equation, but in expectation and with the slow rates. When solutions of the PDEs fall in general Sobolev spaces, Duan et al. (2021b) established non-asymptotic convergence rate for DRM using a method similar to Jiao et al. (2021). The most relevant work to ours is Lu et al. (2021b),

which used peeling methods to derive sharper generalization bounds of the DRM and PINNs for the Schrödinger equation on a hypercube with zero Dirichlet boundary condition. However, Lu et al. (2021b) assumed that the function class of neural networks is a subset of $H_0^1$, which is challenging to achieve. For the DRM, the peeling method in Lu et al. (2021b) cannot be applied to derive the generalization error of the Poisson equation, as in this scenario, the population loss isn't the expectation of the empirical loss. For the PINNs, Lu et al. (2021b) required the strong convexity and only considered the static Schrödinger equation with zero Dirichlet boundary condition, but our approach does not need this condition and works for general linear second order elliptic PDEs.

## 1.2 CONTRIBUTIONS

- For the aspect of approximation via neural networks, we show that the functions in $\mathcal{B}^2(\Omega)$ can be well approximated in the $H^1$ norm by two-layer ReLU neural networks with controlled weights, and similar results are also presented for functions in $\mathcal{B}^3(\Omega)$ in the $H^2$ norm. Compared to the results in Lu et al. (2021c), our approximation rate is faster and the Barron space in our setting is larger than the spectral Barron space in Lu et al. (2021c). Compared with other approximation results for Barron functions (Siegel & Xu, 2022a; Siegel, 2023), the constant in our result is independent of the dimension.

- For the DRM, we derive sharper generalization bounds for the Poisson equation and Schrödinger equation with Neumann boundary condition, regardless of whether the solutions fall in Barron spaces or Sobolev spaces. Our methods rely on the strongly convex property of the variational form and the localized analysis of Bartlett et al. (2005); Koltchinskii (2006). However, these methods cannot be applied directly, as for the Poisson equation, the expectation of empirical loss is not equal to the variational formulation. Additionally, for the static Schrödinger equation, the strongly convex property cannot be simply regarded as the Bernstein condition in Bartlett et al. (2005), as the solutions of the PDEs often do not belong to the function class of neural networks in our setting.

- For the PINNs, we regard this framework as a scenario within multi-task learning (MTL). At this time, there are two key points: one is that the loss functions are non-negative and the other one is that a non-exact oracle inequality suffices. To achieve our goal, we extend the entropy method to derive a Talagrand-type concentration inequality for MTL, which offers better constants than those provided by Theorem 1 in Yousefi et al. (2018). Consequently, similar results to those in single-task setting can be established, yielding a non-exact oracle inequality tailored for PINNs. Unlike Lu et al. (2021b), which required the strong convexity, our approach does not impose this requirement. While we have only presented results for the linear second order elliptic equations with Dirichlet boundary conditions, our method can serve as a framework for PINNs for a wide range of PDEs, as well as other methods that share similar forms with PINNs.

- In the Discussion section, we investigate the complexity of over-parameterized two-layer neural networks when approximating functions in Barron space, and demonstrate meaningful generalization errors in the setting of over-parameterization. Additionally, we discuss other boundary conditions for Deep Ritz Method.

## 1.3 NOTATION

For $x \in \mathbb{R}^d$, $|x|_p$ denotes its $p$-norm and we use $|x|$ as shorthand for $|x|_2$. We denote the inner product of vectors $x, y \in \mathbb{R}^d$ by $x \cdot y$. For the $d$-dimensional ball with radius $r$ in the $p$-norm and the boundary of this ball, we denote them by $B_p^d(r)$ and $\partial B_p^d(r)$ respectively. For a set $\mathcal{F}$ that is a subset of a metric space with metric $d$, we use $\mathcal{N}(\mathcal{F}, d, \epsilon)$ to denote its covering number with given radius $\epsilon$ and the metric $d$. For given probability measure $P$ and a sequence of random variables $\{X_i\}_{i=1}^n$ distributed according to $P$, we denote the empirical measure of $P$ by $P_n$, i.e. $P_n = \frac{1}{n} \sum_{i=1}^n \delta_{X_i}$. For the activation functions, we write $\sigma_k(x)$ for the ReLU$^k$ activation function, i.e., $\sigma_k(x) := (\max(0, x))^k$. And we use $\sigma$ for $\sigma_1$ for simplicity. Given a domain $\Omega \subset \mathbb{R}^d$, we denote $|\Omega|$ and $|\partial\Omega|$ the measure of $\Omega$ and its boundary $\partial\Omega$, respectively.

## 2 DEEP RITZ METHOD

### 2.1 SET UP

Let $\Omega = (0,1)^d$ be the unit hypercube on $\mathbb{R}^d$ and $\partial\Omega$ be the boundary of $\Omega$. We consider the Poisson equation and static Schrödinger equation on $\Omega$ with Neumann boundary condition.

Poisson equation:

$$-\Delta u = f \ in \ \Omega, \ \ \frac{\partial u}{\partial \nu} = 0 \ on \ \partial\Omega. \tag{1}$$

Static Schrödinger equation:

$$-\Delta u + Vu = f \ in \ \Omega, \ \ \frac{\partial u}{\partial \nu} = 0 \ on \ \partial\Omega. \tag{2}$$

In this section, we follow the framework established in Lu et al. (2021c), which characterizes the solutions through variational formulations. For completeness, the detailed results are presented as follows.

**Proposition 1** (Proposition 1 in Lu et al. (2021c) ). *(1) Assume that $f \in L^2(\Omega)$ with $\int_\Omega f dx = 0$. Then there exists a unique weak solution $u_P^* \in H_*^1(\Omega) := \{u \in H^1(\Omega) : \int_\Omega u dx = 0\}$ to the Poisson equation. Moreover, we have that*

$$u_P^* = \underset{u \in H^1(\Omega)}{\arg\min} \, \mathcal{E}_P(u) := \underset{u \in H^1(\Omega)}{\arg\min} \left\{ \int_\Omega |\nabla u|^2 dx + \left(\int_\Omega u dx\right)^2 - 2\int_\Omega f u dx \right\}, \tag{3}$$

*and that for any $u \in H^1(\Omega)$,*

$$\mathcal{E}_P(u) - \mathcal{E}_P(u_P^*) \leq \|u - u_P^*\|_{H^1(\Omega)}^2 \leq \max\{2c_P + 1, 2\}(\mathcal{E}_P(u) - \mathcal{E}_P(u_P^*)), \tag{4}$$

*where $c_P$ is the Poincaré constant on the domain $\Omega$.*

*(2) Assume that $f, V \in L^\infty(\Omega)$ and that $0 < V_{min} \leq V(x) \leq V_{max} < \infty$ for all $x \in \Omega$ and some constants $V_{min}$ and $V_{max}$. Then there exists a unique weak solution $u_S^* \in H^1(\Omega)$ to the static Schrödinger equation. Moreover, we have that*

$$u_S^* = \underset{u \in H^1(\Omega)}{\arg\min} \, \mathcal{E}_S(u) := \underset{u \in H^1(\Omega)}{\arg\min} \left\{ \int_\Omega |\nabla u|^2 + V|u|^2 dx - 2\int_\Omega f u dx \right\}, \tag{5}$$

*and that for any $u \in H^1(\Omega)$,*

$$\frac{1}{\max(1, V_{max})}(\mathcal{E}_S(u) - \mathcal{E}_S(u_S^*)) \leq \|u - u_S^*\|_{H^1(\Omega)}^2 \leq \frac{1}{\min(1, V_{min})}(\mathcal{E}_S(u) - \mathcal{E}_S(u_S^*)). \tag{6}$$

Throughout the paper, we assume that $f \in L^\infty(\Omega)$ and $V \in L^\infty(\Omega)$ with $0 < V_{min} \leq V(x) \leq V_{max} < \infty$. The boundedness is essential in our method for deriving fast rates and it also leads to the strongly convex property in Proposition 1(2). There are also some methods for deriving generalization error beyond boundedness, as discussed in Mendelson (2015; 2018); Lecué & Mendelson (2013). However, these approaches often require additional assumptions, such as specific properties of the data distributions or function classes, which can be difficult to verify in practice.

The core concept of DRM involves substituting the function class of neural networks for Sobolev spaces and then training the neural networks to minimize the variational formulations. Subsequently, we can employ Monte-Carlo method to compute the high-dimensional integrals, as traditional quadrature methods are constrained by the curse of dimensionality in this context.

Let $\{X_i\}_{i=1}^n$ be an i.i.d. sequence of random variables distributed uniformly in $\Omega$. As in our setting, the volume of $\Omega$ is 1, thus the empirical losses can be written directly as

$$\mathcal{E}_{n,P}(u) = \frac{1}{n}\sum_{i=1}^n (|\nabla u(X_i)|^2 - 2f(X_i)u(X_i)) + \left(\frac{1}{n}\sum_{i=1}^n u(X_i)\right)^2 \tag{7}$$

and

$$\mathcal{E}_{n,S}(u) = \frac{1}{n}\sum_{i=1}^{n}(|\nabla u(X_i)|^2 + V(X_i)|u(X_i)|^2 - 2f(X_i)u(X_i)), \tag{8}$$

where we write $\mathcal{E}_{n,P}$ and $\mathcal{E}_{n,S}$ for the empirical losses of the Poisson equation and static Schrödinger equation respectively. Note that the expectation of $\mathcal{E}_{n,P}(u)$ is not equal to $\mathcal{E}_P(u)$, which limits most methods for deriving a fast rate for the Poisson equation.

## 2.2 MAIN RESULTS

The aim of this section is to establish a framework for deriving improved generalization bounds for the DRM. In the setting where the solutions lie in the Barron space $\mathcal{B}^2(\Omega)$, we demonstrate that the generalization error between the empirical solutions from minimizing the empirical losses and the exact solutions grows polynomially with the underlying dimension, enabling the DRM to overcome the curse of dimensionality in this context. Furthermore, when the solutions fall in the general Sobolev spaces, we provide tight generalization bounds through the localization analysis.

We begin by presenting the definition of the Barron space, as introduced in Barron (1993).

$$\mathcal{B}^s(\Omega) := \{f : \Omega \to \mathbb{C} : \|f\|_{\mathcal{B}^s(\Omega)} := \inf_{f_e|_\Omega = f}\int_{\mathbb{R}^d}(1+|\omega|_1)^s|\hat{f}_e(\omega)|d\omega < \infty\}, \tag{9}$$

where the infimum is over extensions $f_e \in L^1(\mathbb{R}^d)$ and $\hat{f}_e$ is the Fourier transform of $f_e$. Note that we choose 1-norm for $\omega$ in the definition just for simplicity.

There are also several different definitions of Barron space (Ma et al., 2022) and the relationships between them have been studied in Siegel & Xu (2023). The most important property of functions in the Barron space is that those functions can be efficiently approximated by two-layer neural networks without the curse of dimensionality. It has been shown in Barron (1993) that two-layer neural networks with sigmoidal activation functions can achieve approximation rate $\mathcal{O}(1/\sqrt{m})$ under the $L^2$ norm, where $m$ is the number of neurons. And the results have been extended to the Sobolev norms (Siegel & Xu, 2022a;b). However, some constants in these extensions implicitly depend on the dimension and there is a possibility that the weights may be unbounded. To address these concerns, we demonstrate the approximation results for functions in the Barron space under the $H^1$ norm. Additionally, for completeness, the approximation result in $W^{k,\infty}(\Omega)$ with $W^{1,\infty}$ norm is also presented, which was originally derived in Yang et al. (2023b).

**Proposition 2** (Approximation results in the $H^1$ norm).

*(1) Barron space: For any $f \in \mathcal{B}^2(\Omega)$, there exists a two-layer neural network $f_m \in \mathcal{F}_{m,1}(5\|f\|_{\mathcal{B}^2(\Omega)})$ such that*

$$\|f - f_m\|_{H^1(\Omega)} \le c\|f\|_{\mathcal{B}^2(\Omega)}m^{-(\frac{1}{2}+\frac{1}{3d})}, \tag{10}$$

*where $\mathcal{F}_{m,1}(B) := \{\sum_{i=1}^{m}\gamma_i\sigma(\omega_i \cdot x + t_i) : |\omega_i|_1 = 1, t_i \in [-1,1), \sum_{i=1}^{m}|\gamma_i| \le B\}$ for any positive constant $B$ and $c$ is a universal constant.*

*(2) Sobolev space: For any $f \in \mathcal{W}^{k,\infty}(\Omega)$ with $k \in \mathbb{N}$, $k \ge 2$ and $\|f\|_{\mathcal{W}^{k,\infty}(\Omega)} \le 1$, any $N, L \in \mathbb{N}_+$, there exists a ReLU neural network $\phi$ with the width $(34+d)2^d k^{d+1}(N+1)\log_2(8N)$ and depth $56d^2k^2(L+1)\log_2(4L)$ such that*

$$\|f(x) - \phi(x)\|_{\mathcal{W}^{1,\infty}(\Omega)} \le C(k,d)N^{-2(k-1)/d}L^{-2(k-1)/d}, \tag{11}$$

*where $C(k,d)$ is the constant independent with $N, L$.*

**Remark 1.** *When approximation functions in $\mathcal{B}^2(\Omega)$, our derived bound exhibits a faster rate than the bound of $m^{-\frac{1}{2}}$ presented in Xu (2020). Although our bound is slower than the bound $m^{-(\frac{1}{2}+\frac{1}{2(d+1)})}$ shown in Siegel & Xu (2022a), it is important to note that the constant within the approximation rate of Siegel & Xu (2022a) may depend exponentially on the dimension and the weights of two-layer neural network could potentially be unbounded. In contrast, the constant in our approximation is dimension-independent and the weights are controlled.*

For the convenience of expression, we write $\Phi(N, L, B)$ for the function class of ReLU neural networks in Proposition 2(2) with width $(34 + d)2^d k^{d+1}(N + 1)\log_2(8N)$, depth $56d^2 k^2(L + 1)\log_2(4L)$ and $W^{1,\infty}$ norm bounded by $B$ such that the approximation result in Proposition 2(2) holds for any $f \in \mathcal{W}^{k,\infty}(\Omega)$ with $\|f\|_{\mathcal{W}^{k,\infty}(\Omega)} \leq 1$.

With the approximation results above, we can derive the generalization error for the Poisson equation and the static Schrödinger equation through the localized analysis.

**Theorem 3** (Generalization error for the Poisson equation).

*Let $u_P^* \in H_*^1(\Omega)$ solve the Poisson equation and $u_{n,P}$ be the minimizer of the empirical loss $\mathcal{E}_{n,P}$ in the function class $\mathcal{F}$.*

*(1) For $u_P^* \in \mathcal{B}^2(\Omega)$, taking $\mathcal{F} = \mathcal{F}_{m,1}(5\|u_P^*\|_{\mathcal{B}^2(\Omega)})$, then with probability as least $1 - e^{-t}$*

$$\mathcal{E}_P(u_{n,P}) - \mathcal{E}_P(u_P^*) \leq CM^2 \log M \left( \frac{md \log n}{n} + \left( \frac{1}{m} \right)^{1+\frac{2}{3d}} + \frac{t}{n} \right), \tag{12}$$

*where $C$ is a universal constant and $M$ is the upper bound for $\|f\|_{L^\infty}$, $\|u_P^*\|_{\mathcal{B}^2(\Omega)}$.*

*By taking $m = \left( \frac{n}{d} \right)^{\frac{3d}{2(3d+1)}}$, we have*

$$\mathcal{E}_P(u_{n,P}) - \mathcal{E}_P(u_P^*) \leq CM^2 \log M \left( \left( \frac{d}{n} \right)^{\frac{3d+2}{2(3d+1)}} \log n + \frac{t}{n} \right). \tag{13}$$

*(2) For $u_P^* \in \mathcal{W}^{k,\infty}(\Omega)$, taking $\mathcal{F} = \Phi(N, L, B\|u_P^*\|_{\mathcal{W}^{k,\infty}(\Omega)})$, then with probability at least $1 - e^{-t}$*

$$\mathcal{E}_P(u_{n,P}) - \mathcal{E}_P(u_P^*) \leq C \left( \frac{(NL)^2(\log N \log L)^3}{n} + (NL)^{-4(k-1)/d} + \frac{t}{n} \right), \tag{14}$$

*where $n \geq C(NL)^2(\log N \log L)^3$ and $C$ is a constant independent of $N, L, n$.*

*By taking $N = L = n^{\frac{d}{4(d+2(k-1))}}$, we have*

$$\mathcal{E}_P(u_{n,P}) - \mathcal{E}_P(u_P^*) \leq C \left( n^{-\frac{2k-2}{d+2k-2}} (\log n)^6 + \frac{t}{n} \right). \tag{15}$$

The generalization error for the static Schrödinger equation shares similar form with that in Theorem 3. For readability and brevity, we put it in Appendix (see Theorem 9).

**Remark 2.** *By utilizing the strong convexity of the Ritz functional and localized analysis, we improve the convergence rate $n^{-\frac{2k-2}{d+4k-4}}$ as shown in Duan et al. (2021b) to $n^{-\frac{2k-2}{d+2k-2}}$. Furthermore, when the solution belongs to $\mathcal{B}^2(\Omega)$, our convergence rate $\left( \frac{d}{n} \right)^{\frac{3d+2}{2(3d+1)}}$ is faster than $n^{-\frac{1}{3}}$ in Lu et al. (2021c) and explicitly demonstrates its dependency on the dimension.*

**Remark 3.** *Due to the equivalence between $H^1$-error and the energy excess as shown in Proposition 1, we are able to deduce the generalization error for both the Poisson equation and the static Schrödinger equation under the $H^1$ norm. For example, one can derive that for the Poisson equation, if $u_P^* \in \mathcal{B}^2(\Omega)$, then*

$$\|u_{n,P} - u_P^*\|_{H^1(\Omega)}^2 \leq CM^2 \log M \left( \left( \frac{d}{n} \right)^{\frac{3d+2}{2(3d+1)}} \log n + \frac{t}{n} \right). \tag{16}$$

In the setting of over-parameterization, the generalization bound in (12) becomes meaningless. Fortunately, the function class $\mathcal{F}$ in Theorem 3(1) has constraints on the weights of the two-layer neural networks, thus we can obtain width-independent upper bounds on their covering number. See Discuss section D.1 in the appendix for more details.

## 3 PHYSICS-INFORMED NEURAL NETWORKS

### 3.1 SET UP

In this section, we will consider the following linear second order elliptic equation with Dirichlet boundary condition.

$$\begin{cases} -\sum_{i,j=1}^{d} a_{ij}\partial_{ij}u + \sum_{i=1}^{d} b_i\partial_i u + cu = f, & in\ \Omega, \\ \\ u = g, & on\ \partial\Omega, \end{cases} \tag{17}$$

where $a_{ij} \in C(\bar{\Omega})$, $b_i, c, f \in L^\infty(\Omega)$, $g \in L^\infty(\partial\Omega)$ and $\Omega \subset (0,1)^d$ is an open bounded domain with properly smooth boundary. Additionally, we assume that the strictly elliptic condition holds, i.e., there exists a constant $\lambda > 0$ such that $\sum_{i,j=1}^{d} a_{ij}\xi_i\xi_j \geq \lambda|\xi|^2$ for $\forall\, x \in \Omega, \xi \in \mathbb{R}^d$.

In the framework of PINNs, we train the neural network $u$ with the following loss function.

$$\mathcal{L}(u) := \int_\Omega \left( -\sum_{i,j=1}^{d} a_{ij}(x)\partial_{ij}u(x) + \sum_{i=1}^{d} b_i(x)\partial_i u(x) + c(x)u(x) - f(x) \right)^2 dx + \int_{\partial\Omega} (u(y)-g(y))^2 dy. \tag{18}$$

By employing the Monte Carlo method, the empirical version of $\mathcal{L}$ can be written as

$$\mathcal{L}_N(u) := \frac{|\Omega|}{N_1} \sum_{k=1}^{N_1} \left( -\sum_{i,j=1}^{d} a_{ij}(X_k)\partial_{ij}u(X_k) + \sum_{i=1}^{d} b_i(X_k)\partial_i u(X_k) + c(X_k)u(X_k) - f(X_k) \right)^2$$

$$+ \frac{|\partial\Omega|}{N_2} \sum_{k=1}^{N_2} (u(Y_k) - g(Y_k))^2, \tag{19}$$

where $N = (N_1, N_2)$, $\{X_k\}_{k=1}^{N_1}$ and $\{Y_k\}_{k=1}^{N_2}$ are i.i.d. random variables distributed according to the uniform distribution $U(\Omega)$ on $\Omega$ and $U(\partial\Omega)$ on $\partial\Omega$, respectively.

Given the empirical loss $\mathcal{L}_N$, the empirical minimization algorithm aims to seek $u_N$ which minimizes $\mathcal{L}_N$, that is:

$$u_N \in \arg\min_{u \in \mathcal{F}} \mathcal{L}_N(u),$$

where $\mathcal{F}$ is a parameterized hypothesis function class.

### 3.2 MAIN RESULTS

We begin by presenting the approximation results in the $H^2$ norm.

**Proposition 4** (Approximation results in the $H^2$ norm).

*(1) Barron space: For any $f \in \mathcal{B}^3(\Omega)$, there exists a two-layer neural network $f_m \in \mathcal{F}_{m,2}(c\|f\|_{\mathcal{B}^3(\Omega)})$ such that*

$$\|f - f_m\|_{H^2(\Omega)} \leq c\|f\|_{\mathcal{B}^3(\Omega)} m^{-(\frac{1}{2}+\frac{1}{3d})}, \tag{20}$$

*where $\mathcal{F}_{m,2}(B) := \{\sum_{i=1}^{m} \gamma_i \sigma_2(\omega_i \cdot x + t_i) :\ |\omega_i|_1 = 1, t_i \in [-1,1), \sum_{i=1}^{m} |\gamma_i| \leq B\}$ for any positive constant $B$ and $c$ is a universal constant.*

*(2) Sobolev space: For any $f \in \mathcal{W}^{k,\infty}(\Omega)$ with $k > 3$ and any integer $K \geq 2$, there exists some sparse ReLU$^3$ neural network $\phi \in \Phi(L, W, S, B; H)$ with $L = \mathcal{O}(1), W = \mathcal{O}(K^d), S = \mathcal{O}(K^d), B = 1, H = \mathcal{O}(1)$, such that*

$$\|f(x) - \phi(x)\|_{H^2(\Omega)} \leq \frac{C}{K^{k-2}}, \tag{21}$$

*where $C$ is a constant independent of $K$, $\Phi(L, W, S, B; H)$ denote the function class of ReLU$^3$ neural networks with depth $L$, width $W$ and at most $S$ non-zero weights taking their values in $[-B, B]$. Moreover, the $W^{2,\infty}$ norms of functions in $\Phi(L, W, S, B; H)$ have the upper bound $H$.*

The framework of PINNs can be regarded as a form of multi-task learning (MTL), as a single neural network is designed to simultaneously learn multiple related tasks, involving the enforcement of physical laws and constraints within the learning process. In contrast to traditional single-task learning, MTL encompasses $T$ supervised learning tasks sampled from the input-output space $\mathcal{X}_1 \times \mathcal{Y}_1, \cdots, \mathcal{X}_T \times \mathcal{Y}_T$ respectively. Each task $t$ is represented by an independent random vector $(X_t, Y_t)$ distributed according to a probability distribution $\mu_t$.

Before presenting our results, we first introduce some notations. Let $(X_t^i, Y_t^i)_{i=1}^{N_t}$ be a sequence of i.i.d. random samples drawn from the distribution $\mu_t$ for $t = 1, \cdots, T$. For any vector-valued function $\boldsymbol{f} = (f_1, \cdots, f_T)$, we denote its expectation and its empirical part as

$$P\boldsymbol{f} := \frac{1}{T}\sum_{t=1}^{T} Pf_t, \ P_N\boldsymbol{f} := \frac{1}{T}\sum_{t=1}^{T} P_{N_t}f_t, \tag{22}$$

where $N = (N_1, \cdots, N_T)$, $Pf_t := \mathbb{E}[f_t(X_t)]$ and $P_{N_t}f_t := \frac{1}{N_t}\sum_{i=1}^{N_t} f_t(X_t^i)$. We denote the component-wise exponentiation of $\boldsymbol{f}$ as $\boldsymbol{f}^\alpha = (f_1^\alpha, \cdots, f_T^\alpha)$ for any $\alpha \in \mathbb{R}$. In the following, we use bold lowercase letters to represent vector-valued functions and bold uppercase letters to indicate the class of functions consisting of vector-valued functions.

To derive sharper generalization bounds for the PINNs, we require results from the field of MTL, with a core component being the Talagrand-type concentration inequality. Yousefi et al. (2018) has established a Talagrand-type inequality for MTL, which is based on so-called Logarithmic Sobolev inequality on log-moment generating function. Of independent interest, we provide a proof using the entropy method. This not only demonstrates the entropy method's capability in proving results for the single-task scenario but also shows that it can be readily adapted to the multi-task scenario. Additionally, the concentration inequality yields better constants compared to those offered by Theorem 1 in Yousefi et al. (2018).

**Theorem 5.** *Let $\mathcal{F} = \{\boldsymbol{f} := (f_1, \cdots, f_T)\}$ be a class of vector-valued functions satisfying $\max_{1 \leq t \leq T} \sup_{x \in \mathcal{X}_t} |f_t(x)| \leq b$. Also assume that $X := (X_t^i)_{(t,i)=(1,1)}^{(T, N_t)}$ is a vector of $\sum_{t=1}^{T} N_t$ independent random variables. Let $\{\sigma_t^i\}_{t,i}$ be a sequence of independent Rademacher variables. If $\frac{1}{T} \sup_{\boldsymbol{f} \in \mathcal{F}} \sum_{t=1}^{T} Var(f_t(X_t^1)) \leq r$, then for every $x > 0$, with probability at least $1 - e^{-x}$,*

$$\sup_{\boldsymbol{f} \in \mathcal{F}}(P\boldsymbol{f} - P_N\boldsymbol{f}) \leq \inf_{\alpha > 0}\left(2(1+\alpha)\mathcal{R}(\mathcal{F}) + 2\sqrt{\frac{xr}{nT}} + \left(1 + \frac{4}{\alpha}\right)\frac{bx}{nT}\right), \tag{23}$$

*where $n = \min_{1 \leq t \leq T} N_t$ and the multi-task Rademacher complexity of function class $\mathcal{F}$ is defined as*

$$\mathcal{R}(\mathcal{F}) := \mathbb{E}_{X,\sigma}\left[\sup_{\boldsymbol{f} \in \mathcal{F}} \frac{1}{T}\sum_{t=1}^{T}\frac{1}{N_t}\sum_{i=1}^{N_t}\sigma_t^i f_t(X_t^i)\right]. \tag{24}$$

*Moreover, the same bound also holds for $\sup_{\boldsymbol{f} \in \mathcal{F}}(P_N\boldsymbol{f} - P\boldsymbol{f})$.*

**Remark 4.** *In comparison with the concentration inequality provided in Yousefi et al. (2018), which is stated as*

$$\sup_{\boldsymbol{f} \in \mathcal{F}}(P\boldsymbol{f} - P_N\boldsymbol{f}) \leq 4\mathcal{R}(\mathcal{F}) + \sqrt{\frac{8xr}{nT}} + \frac{12bx}{nT}, \tag{25}$$

*our result exhibits improved constants by taking $\alpha = 1$.*

Note that the loss functions of the PINNs are all non-negative, which facilitates the derivation of analogous results to those obtained in the single-task context. With the results in MTL, the generalization error for the PINNs can be established.

**Theorem 6** (Generalization error for PINN loss of the linear second order elliptic equation)**.**

*Let $u^*$ be the solution of the linear second order elliptic equation and $n = \min(N_1, N_2)$.*

*(1) If $u^* \in \mathcal{B}^3(\Omega)$, taking $\mathcal{F} = \mathcal{F}_{m,2}(c\|u^*\|_{\mathcal{B}^3(\Omega)})$, then with probability at least $1 - e^{-t}$,*

$$\mathcal{L}(u_N) \leq cC_1(\Omega, M) \left( \frac{m \log n}{n} + \left( \frac{1}{m} \right)^{1 + \frac{2}{3d}} + \frac{t}{n} \right), \tag{26}$$

*where $c$ is a universal constant and $C_1(\Omega, M) := \max\{d^2 M^2, C(Tr, \Omega), |\Omega| d^2 M^4 + |\partial\Omega| M^2\}$, $C(Tr, \Omega)$ is the constant in the Trace theorem for $\Omega$.*

*By taking $m = n^{\frac{3d}{2(3d+1)}}$, we have*

$$\mathcal{L}(u_N) \leq cC_1(\Omega, M) \left( \left( \frac{1}{n} \right)^{\frac{3d+2}{2(3d+1)}} \log n + \frac{t}{n} \right). \tag{27}$$

*(2) If $u^* \in \mathcal{W}^{k,\infty}(\Omega)$ for $k > 3$, taking $\mathcal{F} = \Phi(L, W, S, B; H)$ with $L = \mathcal{O}(1), W = \mathcal{O}(K^d), S = \mathcal{O}(K^d), B = 1, H = \mathcal{O}(1)$, then with probability at least $1 - e^{-t}$,*

$$\mathcal{L}(u_N) \leq C \left( \frac{K^d (\log K + \log n)}{n} + \left( \frac{1}{K} \right)^{2k-4} + \frac{t}{n} \right), \tag{28}$$

*where $C$ is a constant independent of $K, N$.*

*By taking $K = n^{\frac{1}{d+2k-4}}$, we have*

$$\mathcal{L}(u_N) \leq C \left( n^{-\frac{2k-4}{d+2k-4}} \log n + \frac{t}{n} \right). \tag{29}$$

**Remark 5.** *The convergence rate $n^{-\frac{2k-4}{d+2k-4}}$ is faster than $n^{-\frac{2k-4}{d+2k-8}}$ presented in Jiao et al. (2021) and is same as that in Lu et al. (2021b) for the static Schrödinger equation with zero Dirichlet boundary condition. However, our result does not require the strong convexity of the objective function. Furthermore, the objective function in Lu et al. (2021b) only involves one task.*

Note that in certain cases, for instance, when $\Omega = (0, 1)^d$, the constant $C(Tr, \Omega)$ is at most $d$, at this time, $\mathcal{L}(u_N)$ in Theorem 6(1) only depends polynomially with the underlying dimension.

Although Theorem 6 provides a generalization error for the loss function of PINNs, it is often necessary to measure the generalization error between the empirical solution and the true solution under a certain norm. Fortunately, from Lemma 17, we can deduce that

$$\|u_N - u^*\|^2_{H^{\frac{1}{2}}(\Omega)} \leq C_\Omega(\|\mathcal{L}u_N - f\|^2_{L^2(\Omega)} + \|u_N - g\|^2_{L^2(\partial\Omega)}) = C_\Omega \mathcal{L}(u_N). \tag{30}$$

Therefore, under the settings of Theorem 6, we can obtain the generalization error for the linear second order elliptic equation in the $H^{\frac{1}{2}}$ norm.

For the PINNs, we only focus on the $L^2$ loss, as considered in the original study (Raissi et al., 2019). Actually, the design of the loss function should incorporate some priori estimation, which serves as a form of stability property (Wang et al., 2022). Specifically, the design of the loss function should follow the principle that if the loss of PINNs $\mathcal{L}(u)$ is small for some function $u$, then $u$ should be close to the true solution under some appropriate norm. For instance, Theorem 1.2.19 in Garroni & Menaldi (2002) demonstrates that, under some suitable conditions for domain $\Omega$ and related functions $a_{ij}, b_i, c, f, g$, the solution $u^*$ of the linear second order elliptic equation satisfies that

$$\|u^*\|_{H^2(\Omega)} \leq C(\|f\|_{L^2(\Omega)} + \|g\|_{H^{\frac{3}{2}}(\partial\Omega)}). \tag{31}$$

Thus, if we apply the loss

$$\mathcal{L}(u) = \|Lu - f\|^2_{L^2(\Omega)} + \|u - g\|^2_{H^{\frac{3}{2}}(\partial\Omega)}, \tag{32}$$

we may obtain the generalization error in the $H^2$ norm. However, this term $\|g\|_{H^{\frac{3}{2}}(\partial\Omega)}$ is challenging to compute because it also requires ensuring Lipschitz continuity with respect to the parameters, which is essential for estimating the covering number. We leave this as a direction for future work.

On the other hand, some variants of PINNs do not fit the standard MTL framework. For instance, within the extended physics-informed neural networks (XPINNs) framework, to ensure continuity, samples from adjacent regions have cross-correlations. The detailed theoretical framework for XPINNs remains an area for future research.

## 4 CONCLUSION

In this paper, we have refined the generalization bounds for the DRM and PINNs through the localization techniques. For the DRM, our attention was centered on the Poisson equation and the static Schrödinger equation on the $d$-dimensional unit hypercube with Neumann boundary condition. As for the PINNs, our focus shifted to the general linear second elliptic PDEs with Dirichlet boundary condition. Additionally, in both neural networks based approaches for solving PDEs, we considered two scenarios: when the solutions of the PDEs belong to the Barron spaces and when they belong to the Sobolev spaces. Furthermore, we believe that the methodologies established in this paper can be extended to a variety of other methods involving machine learning for solving PDEs.

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

APPENDIX

The Appendix is organized into four parts: Proof of Section 2, Proof of Section 3, Auxiliary Lemmas, and Discussion.

## A  PROOF OF SECTION 2

### A.1  PROOF OF PROPOSITION 2

The proof follows a similar procedure to that in Barron (1993), but the method in Barron (1993) can only yield a slow rate of approximation. We start with a sketch of the proof. For any function in the Barron space, we first prove that it belongs to the $H^1(\Omega)$ closure of the convex hull of some set. Then estimating the metric entropy of the set and applying Theorem 1 in Makovoz (1996) (see Lemma 10) leads to the fast rate of approximation.

For the function $f \in \mathcal{B}^2(\Omega)$, according to the definition of Barron space, we can assume that the infimum can be attained at the function $f_e$. To simplify the notation, we write $f_e$ as $f$, since $f_e|_\Omega = f$. From the formula of Fourier inverse transform and the fact that $f$ is real-valued,

$$
\begin{aligned}
f(x) &= Re \int_{\mathbb{R}^d} e^{i\omega \cdot x} \hat{f}(\omega) d\omega \\
&= Re \int_{\mathbb{R}^d} e^{i\omega \cdot x} e^{i\theta(\omega)} |\hat{f}(\omega)| d\omega \\
&= \int_{\mathbb{R}^d} \cos(\omega \cdot x + \theta(\omega)) |\hat{f}(\omega)| d\omega \\
&= \int_{\mathbb{R}^d} \frac{B \cos(\omega \cdot x + \theta(\omega))}{(1 + |\omega|_1)^2} \Lambda(d\omega) \\
&= \int_{\mathbb{R}^d} g(x, \omega) \Lambda(d\omega),
\end{aligned}
\tag{33}
$$

where $B = \int_{\mathbb{R}^d} (1 + |\omega|_1)^2 |\hat{f}(\omega)| d\omega$, $\Lambda(d\omega) = \frac{(1 + |\omega|_1)^2 |\hat{f}(\omega)| d\omega}{B}$ is a probability measure , $e^{i\theta(\omega)}$ is the phase of $\hat{f}(\omega)$ and

$$
g(x, \omega) = \frac{B \cos(\omega \cdot x + \theta(\omega))}{(1 + |\omega|_1)^2}.
\tag{34}
$$

From the integral representation of $f$ and the form of $g$, i.e. (33) and (34), we can deduce that $f$ is in the $H^1(\Omega)$ closure of the convex hull of the function class

$$
\mathcal{G}_{cos}(B) := \left\{ \frac{B \cos(\omega \cdot x + t)}{(1 + |\omega|_1)^2} : \omega \in \mathbb{R}^d, t \in \mathbb{R} \right\}.
\tag{35}
$$

It could be easily verified via the probabilistic method. Assume that $\{\omega_i\}_{i=1}^n$ is a sequence of i.i.d. random variables distributed according to $\Lambda$, then

$$
\begin{aligned}
&\mathbb{E}\left[ \| f(x) - \frac{1}{n} \sum_{i=1}^n g(x, \omega_i) \|_{H^1(\Omega)}^2 \right] \\
&= \int_\Omega \mathbb{E}\left[ |f(x) - \frac{1}{n} \sum_{i=1}^n g(x, \omega_i)|^2 + |\nabla f(x) - \frac{1}{n} \sum_{i=1}^n \nabla g(x, \omega_i)|^2 \right] dx \\
&= \frac{1}{n} \int_\Omega Var(g(x, \omega)) dx + \frac{1}{n} \int_\Omega Tr(Cov[\nabla g(x, \omega)]) dx \\
&\leq \frac{\mathbb{E}[\| g(x, \omega) \|_{H^1(\Omega)}^2]}{n} \\
&\leq \frac{2B^2}{n},
\end{aligned}
$$

where the first equality follows from Fubini's theorem and the last inequality holds due to the facts that $|g(x,\omega)| \le B$ and $|\nabla g(x,\omega)| \le B$ for any $x, \omega$.

Then, for any given tolerance $\epsilon > 0$, by Markov's inequality,

$$P\left(\|f(x) - \frac{1}{n}\sum_{i=1}^{n} g(x,\omega_i)\|_{H^1(\Omega)} > \epsilon\right) \le \frac{1}{\epsilon^2}\mathbb{E}\left[\|f(x) - \frac{1}{n}\sum_{i=1}^{n} g(x,\omega_i)\|_{H^1(\Omega)}^2\right] \le \frac{2B^2}{n\epsilon^2}.$$

By choosing a large enough $n$ such that $\frac{2B^2}{n\epsilon^2} < 1$, we have

$$P\left(\|f(x) - \frac{1}{n}\sum_{i=1}^{n} g(x,\omega_i)\|_{H^1(\Omega)} \le \epsilon\right) > 0,$$

which implies that there exist realizations of the random variables $\{\omega_i\}_{i=1}^{n}$ such that $\|f(x) - \frac{1}{n}\sum_{i=1}^{n} g(x,\omega_i)\|_{H^1(\Omega)} \le \epsilon$. Therefore, the conclusion holds.

Next, we are going to show that those functions in $\mathcal{G}_{cos}(B)$ are in the $H^1(\Omega)$ closure of the convex hull of the function class $\mathcal{F}_\sigma(5B) \cup \mathcal{F}_\sigma(-5B) \cup \{0\}$, where

$$\mathcal{F}_\sigma(b) := \{b\sigma(\omega \cdot x + t) : |\omega|_1 = 1, t \in [-1,1]\} \tag{36}$$

for any constant $b \in \mathbb{R}$.

Note that although $\mathcal{G}_{cos}(B)$ consists of high-dimensional functions, those functions depend only on the projection of multivariate variable $x$. Specifically, each function $g(x,\omega) = \frac{B\cos(\omega \cdot x + t)}{(1+|\omega|_1)^2} \in \mathcal{G}_{cos}(B)$ is the composition of a one-dimensional function $g(z) = \frac{B\cos(|\omega|_1 z + t)}{(1+|\omega|_1)^2}$ and a linear function $z = \frac{\omega}{|\omega|_1} \cdot x$ with value in $[-1,1]$. Therefore, it suffices to prove that the conclusion holds for $g(z)$ on $[-1,1]$, i.e., to prove that for each $\omega$, $g$ is in the $H^1([-1,1])$ closure of convex hull of $\mathcal{F}_\sigma^1(5B) \cup \mathcal{F}_\sigma^1(-5B) \cup \{0\}$, where

$$\mathcal{F}_\sigma^1(b) := \{b\sigma(\epsilon z + t) : \epsilon = -1 \ or \ 1, t \in [-1,1]\} \tag{37}$$

for any constant $b \in \mathbb{R}$. Then applying the variable substitution leads to the conclusion for $g(x,\omega)$.

In fact, it is easier to handle that in one-dimension due to the relationship between the ReLU functions and the basis function in the finite element method (FEM) (He et al., 2018), specifically the basis functions in the FEM can be represented by ReLU functions. To make it more precise, let us consider the uniform mesh of interval $[-1,1]$ by taking $m+1$ points

$$-1 = x_0 < x_1 < \cdots < x_m = 1,$$

and set $h = \frac{2}{m}$, $x_{-1} = -1 - h$, $x_{m+1} = 1 + h$. For $0 \le i \le m$, introduce the function $\varphi_i(z)$, which is defined as follows:

$$\varphi_i(z) = \begin{cases} \frac{1}{h}(z - z_{i-1}), & if \ z \in [z_{i-1}, z_i], \\ \frac{1}{h}(z_{i+1} - z), & if \ z \in [z_i, z_{i+1}], \\ 0, & otherwise. \end{cases} \tag{38}$$

Clearly, the set $\{\varphi_0, \cdots, \varphi_m\}$ is a basis of $\mathcal{P}_h^1$, which is a vector space of continuous, piece-wise linear functions ($\mathbb{P}_1$ Lagrange finite element, see Chapter 1 of Ern & Guermond (2004) for more details). And $\varphi_i$ can be written as

$$\varphi_i(z) = \frac{\sigma(z - z_{i-1}) - 2\sigma(z - z_i) + \sigma(z - z_{i+1})}{h}. \tag{39}$$

Now, we are ready to present the definition of interpolation operator and the estimation of interpolation error (Ern & Guermond, 2004) (Proposition 1.5 in Ern & Guermond (2004)).

Consider the so-called interpolation operator

$$\mathcal{I}_h^1 : v \in C([-1,1]) \to \sum_{i=0}^{m} v(z_i)\varphi_i \in P_h^1. \tag{40}$$

Then for all $h$ and $v \in H^2([-1,1])$, the interpolation error can be bounded as

$$\|v - \mathcal{I}_h^1 v\|_{L^2([-1,1])} \leq h^2 \|v^{''}\|_{L^2([-1,1])} \ and \ \|v^{'} - (\mathcal{I}_h^1 v)^{'}\|_{L^2([-1,1])} \leq h \|v^{''}\|_{L^2([-1,1])}. \quad (41)$$

By invoking the interpolation operator and the connection between the ReLU functions and the basis functions, we can establish the following conclusion for one-dimensional functions.

**Lemma 1.** *Let $g \in \mathcal{C}^2([-1,1])$ with $\|g^{(s)}\|_{L^\infty} \leq B$ for $s = 0, 1, 2$. Then there exists a two-layer ReLU network $g_m$ of the form*

$$g_m(z) = \sum_{i=1}^{6m-1} a_i \sigma(\epsilon_i z + t_i), \quad (42)$$

*with $|a_i| \leq \frac{2B}{m}$, $\sum_{i=1}^{6m-1} |a_i| \leq 5B$, $|t_i| \leq 1$, $\epsilon_i \in \{-1,1\}$, $1 \leq i \leq 6m-1$ such that*

$$\|g - g_m\|_{H^1([-1,1])} \leq \frac{4\sqrt{2}B}{m}. \quad (43)$$

*Therefore, $g$ is in the $H^1([-1,1])$ closure of the convex hull of $\mathcal{F}_\sigma^1(5B) \cup \mathcal{F}_\sigma^1(-5B) \cup \{0\}$.*

*Proof.* Note that from (39) and (40), the interpolant of $g$ can be written as a combination of ReLU functions as follows

$$\mathcal{I}_h^1(g) = \sum_{i=0}^m g(z_i) \varphi_i(z)$$

$$= \sum_{i=0}^m g(z_i) \frac{\sigma(z - z_{i-1}) - 2\sigma(z - z_i) + \sigma(z - z_{i+1})}{h}$$

$$= \frac{g(z_0)(\sigma(z - z_{-1}) - 2\sigma(z - z_0))}{h} + \frac{g(z_1)\sigma(z - z_0)}{h} + \sum_{i=1}^{m-1} \frac{g(z_{i-1}) - 2g(z_i) + g(z_{i+1})}{h} \sigma(z - z_i)$$

$$= g(z_0) + \frac{g(z_1) - g(z_0)}{h} \sigma(z - z_0) + \sum_{i=1}^{m-1} \frac{g(z_{i-1}) - 2g(z_i) + g(z_{i+1})}{h} \sigma(z - z_i). \quad (44)$$

By the mean value theorem, there exist $\xi_0 \in [z_0, z_1]$ and $\xi_i \in [z_{i-1}, z_{i+1}]$ for $1 \leq i \leq m-1$ such that $g(z_1) - g(z_0) = g^{'}(\xi_0)h$ and $g(z_{i-1}) - 2g(z_i) + g(z_{i+1}) = g^{''}(\xi_i)h^2$ for $1 \leq i \leq m-1$.

Therefore, $\mathcal{I}_h^1(g)$ can be rewritten as

$$\mathcal{I}_h^1(g) = g(z_0) + g^{'}(\xi_0)\sigma(z - z_0) + \sum_{i=1}^{m-1} g^{''}(\xi_i)\sigma(z - z_i)h. \quad (45)$$

On the other hand, the constant can also be represented as a combination of ReLU functions on $[-1,1]$. By the observation that $\sigma(z) + \sigma(-z) = |z|$, we have that for any $z \in [-1,1]$

$$1 = \frac{|1+z| + |1-z|}{2} = \frac{\sigma(z+1) + \sigma(-z-1) + \sigma(-z+1) + \sigma(z-1)}{2}. \quad (46)$$

Plugging (46) into (45) yields that

$$\mathcal{I}_h^1(g) = \sum_{i=1}^m \frac{g(z_0)(\sigma(z+1) + \sigma(-z-1) + \sigma(-z+1) + \sigma(z-1))}{2m} + \sum_{i=1}^m \frac{g^{'}(\xi_0)\sigma(z - z_0)}{m}$$

$$+ \sum_{i=1}^{m-1} \frac{2g^{''}(\xi_i)\sigma(z - z_i)}{m}. \quad (47)$$

Combining the expression of $\mathcal{I}_h^1(g)$ and the estimation for interpolation error, i.e. (47) and (41), leads to that there exists a two-layer neural network $g_m$ of the form

$$g_m(z) = \mathcal{I}_h^1(g) = \sum_{i=1}^{6m-1} a_i \sigma(\epsilon_i z + t_i),$$

with $|a_i| \leq \frac{2B}{m}$, $\sum_{i=1}^{6m-1} |a_i| \leq 5B$, $|t_i| \leq 1$, $\epsilon_i \in \{-1,1\}$, $1 \leq i \leq 6m - 1$ such that

$$\|g - g_m\|_{H^1([-1,1])} \leq \frac{4\sqrt{2}B}{m}.$$

$\square$

Although the interpolation operator can be view as a piece-wise linear interpolation of $g$, which is similar to Lemma 18 in Lu et al. (2021c), our result does not require $g'(0) = 0$ and the value of $g$ at the certain point is also expressed as a combination of ReLU functions. Specifically, the $g_m$ in Lemma 18 of Lu et al. (2021c) has the form $g_m(z) = c + \sum_{i=1}^{2m} a_i \sigma(\epsilon_i z + t_i)$, where $c = g(0)$ and they partition $[-1, 1]$ by $2m$ points with $z_0 = -1, z_m = 0, z_{2m} = 1$. And our result can also be extended in $W^{1,\infty}([-1,1])$ norm like Lemma 18 of Lu et al. (2021c). Note that on $[z_{i-1}, z_i]$

$$\mathcal{I}_h^1(g)(z) = g(z_{i-1})\frac{z_i - z}{h} + g(z_i)\frac{z - z_{i-1}}{h},$$

which is the piece-wise linear interpolation of $g$. Then by bounding the remainder in Lagrange interpolation formula, we have $\|\mathcal{I}_h(g) - g\|_{L^\infty[z_{i-1},z_i]} \leq \frac{h^2}{8}\|g''\|_{L^\infty[z_{i-1},z_i]}$ and

$$
\begin{aligned}
|(\mathcal{I}_h^1(g))'(z) - g'(z)| &= |\frac{g(z_i) - g(z_{i-1})}{h} - g''(z)| \\
&\leq |g'(\xi_i) - g'(z_i)| \\
&\leq h\|g''\|_{L^\infty[z_{i-1},z_i]},
\end{aligned}
\tag{48}
$$

where the first inequality follows from the mean value theorem.

Therefore, $\|\mathcal{I}_h^1(g) - g\|_{W^{1,\infty}([-1,1])} \leq \frac{2B}{m}$.

Lemma 1 implies that for any $\omega$, the one-dimension function $g(z) = \frac{B\cos(|\omega|_1 z + t)}{(1 + |\omega|_1)^2}$ is in the $H^1([-1,1])$ closure of convex hull of $\mathcal{F}_\sigma^1(5B) \cup \mathcal{F}_\sigma^1(-5B) \cup \{0\}$. Then applying the variable substitution yields that those functions in $\mathcal{G}_{cos}(B)$ are in the $H^1(\Omega)$ closure of the convex hull of the function class $\mathcal{F}_\sigma(5B) \cup \mathcal{F}_\sigma(-5B) \cup \{0\}$. Specifically, for any function $h : \mathbb{R} \to \mathbb{R}$ and $\omega \in \mathbb{R}^d$ with $|\omega|_1 = 1$, without loss of generality, we can assume that $\omega_1 > 0$. Then for the integral

$$\int_\Omega |h(\omega \cdot x)|^2 dx = \int_{[0,1]^d} |h(\omega \cdot x)|^2 dx,$$

let $y_1 = \omega \cdot x, y_2 = x_2, \cdots, y_d = x_d$, we have

$$\int_{[0,1]^d} |h(\omega \cdot x)|^2 dx = \frac{1}{\omega_1} \int_0^1 \cdots \int_{\omega_2 \cdot y_2 + \cdots \omega_d \cdot y_d}^{\omega_2 \cdot y_2 + \cdots \omega_d \cdot y_d + \omega_1} |h(y_1)|^2 dy_1 \cdots dy_d \leq \frac{1}{\omega_1} \int_{-1}^1 |h(y_1)|^2 dy_1.$$

Therefore, the conclusion holds for $\mathcal{G}_{cos}(B)$. Recall that $f$ is in the $H^1(\Omega)$ closure of the convex hull of $\mathcal{G}_{cos}(B)$, thus we have the following conclusion.

**Proposition 7.** *For any given function $f$ in $\mathcal{B}^2(\Omega)$, $f$ is in the $H^1(\Omega)$ closure of the convex hull of $\mathcal{F}_\sigma(5\|f\|_{\mathcal{B}^2(\Omega)}) \cup \mathcal{F}_\sigma(-5\|f\|_{\mathcal{B}^2(\Omega)}) \cup \{0\}$, i.e., for any $\epsilon > 0$, there exist $m \in \mathbb{N}$ and $\omega_i, t_i, a_i, 1 \leq i \leq m$ such that*

$$\|f(x) - \sum_{i=1}^m a_i \sigma(\omega_i \cdot x + t_i)\|_{H^1(\Omega)} \leq \epsilon,
\tag{49}$$

*where $|\omega_i|_1 = 1, t_i \in [-1,1], 1 \leq i \leq m$ and $\sum_{i=1}^m |a_i| \leq 5\|f\|_{\mathcal{B}^2(\Omega)}$.*

Proposition 7 implies that functions in $\mathcal{B}^2(\Omega)$ can be approximated by a linear combination of functions in $\mathcal{F}_\sigma(1)$.

Recall that $\mathcal{F}_\sigma(1) = \{\sigma(\omega \cdot x + t) : |\omega|_1 = 1, t \in [-1, 1]\}$. For simplicity, we write $\mathcal{F}_\sigma$ for $\mathcal{F}_\sigma(1)$.

Then to invoke Theorem 1 in Makovoz (1996) (see Lemma 10), it remains to estimate the metric entropy of the function class $\mathcal{F}_\sigma$, which is defined as

$$\epsilon_n(\mathcal{F}_\sigma) := \inf\{\epsilon : \mathcal{F}_\sigma \ can \ be \ covered \ by \ at \ most \ n \ sets \ of \ diameter \ \leq \epsilon \ under \ the \ H^1 \ norm\}. \tag{50}$$

By Lemma 12, we just need to estimate the covering number of $\mathcal{F}_\sigma$, which is easier to handle.

**Proposition 8** (Estimation of the metric entropy). *For any $n \in \mathbb{N}$,*

$$\epsilon_n(\mathcal{F}_\sigma) \leq cn^{-\frac{1}{3d}},$$

*where $c$ is a universal constant.*

*Proof.* For $(\omega_1, t_1), (\omega_2, t_2) \in \partial B_1^d(1) \times [-1, 1]$, we have

$$\|\sigma(\omega_1 \cdot x + t_1) - \sigma(\omega_2 \cdot x + t_2)\|_{H^1(\Omega)}^2$$

$$= \int_\Omega |\sigma(\omega_1 \cdot x + t_1) - \sigma(\omega_2 \cdot x + t_2)|^2 dx + \int_\Omega |\nabla\sigma(\omega_1 \cdot x + t_1) - \nabla\sigma(\omega_2 \cdot x + t_2)|^2 dx$$

$$\leq \int_\Omega |(\omega_1 - \omega_2) \cdot x + (t_1 - t_2)|^2 dx + \int_\Omega |\omega_1 I_{\{\omega_1 \cdot x + t_1 \geq 0\}} - \omega_2 I_{\{\omega_2 \cdot x + t_2 \geq 0\}}|^2 dx$$

$$\leq 2(|\omega_1 - \omega_2|_1^2 + |t_1 - t_2|^2) + \int_\Omega |(\omega_1 - \omega_2) I_{\{\omega_1 \cdot x + t_1 \geq 0\}} + \omega_2 (I_{\{\omega_1 \cdot x + t_1 \geq 0\}} - I_{\{\omega_2 \cdot x + t_2 \geq 0\}})|^2 dx$$

$$\leq 2(|\omega_1 - \omega_2|_1^2 + |t_1 - t_2|^2) + 2|\omega_1 - \omega_2|_1^2 + 2\int_\Omega |I_{\{\omega_1 \cdot x + t_1 \geq 0\}} - I_{\{\omega_2 \cdot x + t_2 \geq 0\}}|^2 dx$$

$$\leq 4(|\omega_1 - \omega_2|_1^2 + |t_1 - t_2|^2) + 2\int_\Omega |I_{\{\omega_1 \cdot x + t_1 \geq 0\}} - I_{\{\omega_2 \cdot x + t_2 \geq 0\}}|^2 dx, \tag{51}$$

where the first inequality is due to that $\sigma$ is 1-Lipschitz continuous, the second and the third inequalities follow the from the mean inequality and the fact that the 2-norm is dominated by the 1-norm.

It is challenging to handle the first and second terms simultaneously due to the discontinuity of indicator functions, thus we turn to handle two terms separately. Note that the first term is related to the covering of $\partial B_1^d(1) \times [-1, 1]$ and the second term is related to the covering of a VC-class of functions (see Chapter 2.6 of Vaart & Wellner (2023) or Chapter 9 of Kosorok (2008)). Therefore, we consider a new space $\mathcal{G}_1$ defined as

$$\mathcal{G}_1 := \{((\omega, t), I_{\{\omega \cdot x + t \geq 0\}}) : \omega \in \partial B_1^d(1), t \in [-1, 1]\}.$$

Obviously, it is a subset of the metric space

$$\mathcal{G}_2 := \{((\omega_1, t_1), I_{\{\omega_2 \cdot x + t_2 \geq 0\}}) : \omega_1, \omega_2 \in \partial B_1^d(1), t_1, t_2 \in [-1, 1]\}$$

with the metric $d$ that for $b_1 = \left((\omega_1^1, t_1^1), I_{\{\omega_2^1 \cdot x + t_2^1 \geq 0\}}\right), b_2 = \left((\omega_1^2, t_1^2), I_{\{\omega_2^2 \cdot x + t_2^2 \geq 0\}}\right)$,

$$d(b_1, b_2) := \sqrt{2(|\omega_1^1 - \omega_1^2|_1^2 + |t_1^1 - t_1^2|^2)} + \|I_{\{\omega_2^1 \cdot x + t_2^1 \geq 0\}} - I_{\{\omega_2^2 \cdot x + t_2^2 \geq 0\}}\|_{L^2(\Omega)}.$$

The key point is that $\mathcal{G}_2$ can be seen as a product space of $\partial B_1^d(1) \times [-1, 1]$ and the function class $\mathcal{F}_1 := \{I_{\{\omega \cdot x + t \geq 0\}} : (\omega, t) \in \partial B_1^d(1) \times [-1, 1]\}$ is a VC-class. Therefore, we can handle the two terms separately.

By defining the metric $d_1$ in $\partial B_1^d(1) \times [-1, 1]$ as

$$d_1\left((\omega_1^1, t_1^1), (\omega_1^2, t_1^2)\right) = \sqrt{2(|\omega_1^1 - \omega_1^2|_1^2 + |t_1^1 - t_1^2|^2)}$$

and the metric $d_2$ in $\mathcal{F}_1$ as

$$d_2\left(I_{\{\omega_2^1 \cdot x + t_2^1 \geq 0\}}, I_{\{\omega_2^2 \cdot x + t_2^2 \geq 0\}}\right) = \|I_{\{\omega_2^1 \cdot x + t_2^1 \geq 0\}} - I_{\{\omega_2^2 \cdot x + t_2^2 \geq 0\}}\|_{L^2(\Omega)},$$

the covering number of $\mathcal{G}_2$ can be bounded as

$$\mathcal{N}(\mathcal{G}_2, d, \epsilon) \leq \mathcal{N}(\partial B_1^d(1) \times [-1, 1], d_1, \frac{\epsilon}{2}) \cdot \mathcal{N}(\mathcal{F}_1, d_2, \frac{\epsilon}{2}).$$

As $\mathcal{F}_1$ is a subset of the collection of all indicator functions of sets in a class with finite VC-dimension, then Theorem 2.6.4 in Vaart & Wellner (2023) implies

$$\mathcal{N}(\mathcal{F}_1, d_2, \epsilon) \leq K(d+1)(4e)^{d+1}\left(\frac{2}{\epsilon}\right)^{2d}$$

with a universal constant $K$, since the collection of all half-spaces in $\mathbb{R}^d$ is a VC-class of dimension $d+1$ (see Lemma 9.12(i) in Kosorok (2008)).

By the inequality $\sqrt{|a| + |b|} \leq \sqrt{|a|} + \sqrt{|b|}$, we have

$$\sqrt{2|\omega_1^1 - \omega_1^2|_1^2 + |t_1^1 - t_1^2|^2} \leq \sqrt{2}(|\omega_1^1 - \omega_1^2|_1 + |t_1^1 - t_1^2|),$$

therefore

$$\mathcal{N}(\partial B_1^d(1) \times [-1, 1], d_1, \epsilon) \leq \mathcal{N}(\partial B_1^d(1), |\cdot|_1, \frac{\sqrt{2}}{2}\epsilon) \cdot \mathcal{N}([-1, 1], |\cdot|, \frac{\sqrt{2}}{2}\epsilon).$$

Combining all results above and Lemma 11, we can compute an upper bound for the covering number of $\mathcal{G}_1$.

$$\begin{aligned}
\mathcal{N}(\mathcal{G}_1, d, \epsilon) &\leq \mathcal{N}(\mathcal{G}_2, d, \frac{\epsilon}{2}) \\
&\leq \mathcal{N}(\partial B_1^d(1) \times [-1, 1], d_1, \frac{\epsilon}{4}) \cdot \mathcal{N}(\mathcal{F}_1, d_2, \frac{\epsilon}{4}) \\
&\leq \mathcal{N}(\partial B_1^d(1), |\cdot|_1, \frac{\sqrt{2}}{8}\epsilon) \cdot \mathcal{N}([-1, 1], |\cdot|, \frac{\sqrt{2}}{8}\epsilon) \cdot \mathcal{N}(\mathcal{F}_1, d_2, \frac{\epsilon}{4}) \\
&\leq K(d+1)(4e)^{d+1}\left(\frac{c}{\epsilon}\right)^{3d},
\end{aligned}$$

where $c$ is a universal constant.

Therefore, applying Lemma 12 yields the desired conclusion.

Note that in Proposition 2(1), we require $t_i \in [-1, 1)$ instead of $t_i \in [-1, 1]$ due to the measurability (see Remark 6). At this time, the approximation result does not change. In fact, for any $\omega \in \mathbb{R}^d$, taking a sequence $\{t_n\}_{n \in \mathbb{N}}$ that is monotonically increasing and tends to 1, we can deduce that $\|\sigma(\omega \cdot x + t_n)\|_{H^1(\Omega)} \to \|\sigma(\omega \cdot x + 1)\|_{H^1(\Omega)}$. It suffices to prove that

$$\int_\Omega |I_{\{\omega \cdot x + t_n \geq 0\}} - I_{\{\omega \cdot x + 1 \geq 0\}}|^2 dx = \int_\Omega |I_{\{\omega \cdot x + t_n > 0\}} - I_{\{\omega \cdot x + 1 > 0\}}|^2 dx \to 0.$$

Since the function $t \mapsto I_{\{u < t\}}$ is left-continuous for any $u \in \mathbb{R}$, so that $I_{\{\omega \cdot x + t_n > 0\}} \to I_{\{\omega \cdot x + 1 > 0\}}$ for all $x \in \Omega$. Then, applying the dominated convergence theorem leads to the conclusion. $\square$

## A.2 PROOF OF THEOREM 3

The proof is based on a new error decomposition and the peeling method. The key point is the fact that $\int_\Omega u^*(x) dx = 0$, thus for any $u \in H^1(\Omega)$,

$$\left(\int_\Omega u(x) dx\right)^2 = \left(\int_\Omega (u(x) - u^*(x)) dx\right)^2 \leq \int_\Omega (u(x) - u^*(x))^2 dx \leq \|u - u^*\|_{H^1(\Omega)}^2, \quad (52)$$

which implies that if $u$ is close enough to $u^*$ in the $H^1$ norm, then $\left(\int_\Omega u(x)dx\right)^2$ is also proportionately small. Furthermore, if $u$ is bounded, we can also prove that the empirical part of $\left(\int_\Omega u(x)dx\right)^2$, i.e., $\left(\frac{1}{n}\sum_{i=1}^n u(X_i)\right)^2$ is also small in high probability via the Hoeffding inequality.

In the proof, we omit the notation for the Poisson equation, i.e., we write $\mathcal{E}$ and $\mathcal{E}_n$ for the population loss $\mathcal{E}_P$ and empirical loss $\mathcal{E}_{n,P}$ respectively. Additionally, we assume that there is a constant $M$ such that $|u^*|, |\nabla u^*|, |f| \leq M$.

Assume that $u_n$ is the minimal solution obtained by minimizing the empirical loss $\mathcal{E}_n$ in the function class $\mathcal{F}$, here we just take $\mathcal{F}$ as a parameterized hypothesis function class. When considering the specific setting, we can choose $\mathcal{F}$ to be the function class of two-layer neural networks or deep neural networks. Additionally, we assume that those functions in $\mathcal{F}$ and their gradients are bounded by $M$ in absolute value and 2-norm.

Recall that the population loss and its empirical part are

$$\mathcal{E}(u) = \int_\Omega |\nabla u(x)|^2 dx - \int_\Omega 2f(x)u(x)dx + \left(\int_\Omega u(x)dx\right)^2 \tag{53}$$

and

$$\mathcal{E}_n(u) = \frac{1}{n}\sum_{i=1}^n |\nabla u(X_i)|^2 - \frac{2}{n}\sum_{i=1}^n f(X_i)u(X_i) + \left(\frac{1}{n}\sum_{i=1}^n u(X_i)\right)^2. \tag{54}$$

By taking $u_\mathcal{F} \in \arg\min_{u\in\mathcal{F}} \|u - u^*\|_{H^1(\Omega)}$, we have the following error decomposition:

$$\begin{aligned}
\mathcal{E}(u_n) - \mathcal{E}(u^*) &= \mathcal{E}(u_n) - \lambda\mathcal{E}_n(u_n) + \lambda(\mathcal{E}_n(u_n) - \mathcal{E}_n(u_\mathcal{F})) + \lambda\mathcal{E}_n(u_\mathcal{F}) - \mathcal{E}(u^*) \\
&\leq \mathcal{E}(u_n) - \lambda\mathcal{E}_n(u_n) + \lambda\mathcal{E}_n(u_\mathcal{F}) - \mathcal{E}(u^*) \\
&= \mathcal{E}(u_n) - \lambda\mathcal{E}_n(u_n) + \lambda(\mathcal{E}_n(u_\mathcal{F}) - \mathcal{E}_n(u^*)) + \lambda\mathcal{E}_n(u^*) - \mathcal{E}(u^*) \\
&= (\mathcal{E}(u_n) - \mathcal{E}(u^*)) - \lambda(\mathcal{E}_n(u_n) - \mathcal{E}_n(u^*)) + \lambda(\mathcal{E}_n(u_\mathcal{F}) - \mathcal{E}_n(u^*)) \\
&\leq \sup_{u\in\mathcal{F}}[(\mathcal{E}(u) - \mathcal{E}(u^*)) - \lambda(\mathcal{E}_n(u) - \mathcal{E}_n(u^*))] + \lambda(\mathcal{E}_n(u_\mathcal{F}) - \mathcal{E}_n(u^*)),
\end{aligned} \tag{55}$$

where the first inequality follows from the definition of $u_n$ and $\lambda$ is a constant to be determined.

In the following, we estimate the two terms separately.

Rearranging the term $\mathcal{E}_n(u_\mathcal{F}) - \mathcal{E}_n(u^*)$ yields

$$\begin{aligned}
&\mathcal{E}_n(u_\mathcal{F}) - \mathcal{E}_n(u^*) \\
&= \frac{1}{n}\sum_{i=1}^n |\nabla u_\mathcal{F}(X_i)|^2 + \left(\frac{1}{n}\sum_{i=1}^n u_\mathcal{F}(X_i)\right)^2 - \frac{2}{n}\sum_{i=1}^n f(X_i)u_\mathcal{F}(X_i) \\
&\quad - \left[\frac{1}{n}\sum_{i=1}^n |\nabla u^*(X_i)|^2 + (\frac{1}{n}\sum_{i=1}^n u^*(X_i))^2 - \frac{2}{n}\sum_{i=1}^n f(X_i)u^*(X_i)\right] \\
&= \frac{1}{n}\sum_{i=1}^n \left[(|\nabla u_\mathcal{F}(X_i)|^2 - 2f(X_i)u_\mathcal{F}(X_i)) - (|\nabla u^*(X_i)|^2 - 2f(X_i)u^*(X_i))\right] \\
&\quad + \left[\left(\frac{1}{n}\sum_{i=1}^n u_\mathcal{F}(X_i)\right)^2 - \left(\frac{1}{n}\sum_{i=1}^n u^*(X_i)\right)^2\right] \\
&:= \phi_n^1 + \phi_n^2,
\end{aligned} \tag{56}$$

where in the last equality, we denote the right two terms in the second equality as $\phi_n^1$ and $\phi_n^2$ respectively.

Define
$$h(x) = (|\nabla u_\mathcal{F}(x)|^2 - 2f(x)u_\mathcal{F}(x)) - (|\nabla u^*(x)|^2 - 2f(x)u^*(x)),$$

then by the boundedness of $u_{\mathcal{F}}, |\nabla u_{\mathcal{F}}|, u^*, |\nabla u^*|$ and $f$, we can deduce that

$$Var(h) \le P(h^2) \le 8M^2\|u_{\mathcal{F}} - u^*\|^2_{H^1(\Omega)} = 8M^2\epsilon^2_{app} \text{ and } |h - \mathbb{E}[h]| \le 2\sup|h| \le 12M^2, \quad (57)$$

where $\epsilon_{app}$ denotes the approximation error in the $H^1(\Omega)$ norm, i.e., $\epsilon_{app} = \|u_{\mathcal{F}} - u^*\|_{H^1(\Omega)}$.

Therefore, from Bernstein inequality (see Lemma 7) and (57), we have that with probability at least $1 - e^{-t}$,

$$\phi_n^1 = \frac{1}{n}\sum_{i=1}^n [(|\nabla u_{\mathcal{F}}(X_i)|^2 - 2f(X_i)u_{\mathcal{F}}(X_i)) - (|\nabla u^*(X_i)|^2 - 2f(X_i)u^*(X_i))]$$

$$\le \mathbb{E}[h(X)] + \sqrt{\frac{24M^2t}{n}\epsilon^2_{app}} + \frac{4M^2t}{n}$$

$$= \mathcal{E}(u_{\mathcal{F}}) - \mathcal{E}(u^*) - \left(\int_\Omega u_{\mathcal{F}}dx\right)^2 + \sqrt{\frac{24M^2t}{n}\epsilon^2_{app}} + \frac{4M^2t}{n} \quad (58)$$

$$\le C\left(\epsilon^2_{app} + \frac{M^2t}{n}\right),$$

where the last inequality follows by the basic inequality $2\sqrt{ab} \le a + b$ for any $a, b > 0$ and Proposition 1(1).

For $\phi_n^2$, the Hoeffding inequality (see Lemma 8) implies

$$P\left(\left|\frac{1}{n}\sum_{i=1}^n u_{\mathcal{F}}(X_i) - \int_\Omega u_{\mathcal{F}}(x)dx\right| \ge 2M\sqrt{\frac{2t}{n}}\right) \le 2e^{-t}.$$

Therefore with probability at least $1 - 2e^{-t}$,

$$\phi_n^2 = \left(\frac{1}{n}\sum_{i=1}^n u_{\mathcal{F}}(X_i)\right)^2 - \left(\frac{1}{n}\sum_{i=1}^n u^*(X_i)\right)^2$$

$$\le \left(\frac{1}{n}\sum_{i=1}^n u_{\mathcal{F}}(X_i)\right)^2$$

$$\le 2\left(\left|\frac{1}{n}\sum_{i=1}^n u_{\mathcal{F}}(x_i) - \int_\Omega u_{\mathcal{F}}(x)dx\right|^2 + \left|\int_\Omega u_{\mathcal{F}}(x)dx\right|^2\right) \quad (59)$$

$$\le C\left(\epsilon^2_{app} + \frac{M^2t}{n}\right).$$

Combining the upper bounds for $\phi_n^1$ and $\phi_n^2$, i.e. (58) and (59), we can deduce that with probability as least $1 - 3e^{-t}$,

$$\mathcal{E}_n(u_{\mathcal{F}}) - \mathcal{E}_n(u_*) \le C\left(\epsilon^2_{app} + \frac{M^2t}{n}\right). \quad (60)$$

Plugging this into the error decomposition (55) yields that with probability as least $1 - 3e^{-t}$,

$$\mathcal{E}(u_n) - \mathcal{E}(u^*) \le \sup_{u \in \mathcal{F}}[(\mathcal{E}(u) - \mathcal{E}(u^*)) - \lambda(\mathcal{E}_n(u) - \mathcal{E}_n(u^*))] + \lambda C\left(\epsilon^2_{app} + \frac{M^2t}{n}\right). \quad (61)$$

For the first term in the right of (61), we employ the peeling technique to establish an upper bound for it.

Let $\rho_0$ be a positive constant to be determined and $\rho_k = 2\rho_{k-1}$ for $k \ge 1$.

Consider the sets $\mathcal{F}_k := \{u \in \mathcal{F} : \rho_{k-1} < \|u - u^*\|^2_{H^1(\Omega)} \le \rho_k\}$ for $k \ge 1$ and $\mathcal{F}_0 = \{u \in \mathcal{F} : \|u - u^*\|^2_{H^1(\Omega)} \le \rho_0\}$ for $k = 0$.

The boundedness of the functions in $\mathcal{F}$, $u^*$ and their respective gradients implies that

$$K := \max k \leq C \log \frac{M^2}{\rho_0},$$

since $\rho_K = 2^K \rho_0$ and $\sup_{u \in \mathcal{F}} \|u - u^*\|^2_{H^1(\Omega)} \leq 4M^2$.

Then for the fixed constant $\delta \in (0, 1)$, set $\delta_k = \frac{\delta}{K+1}$ for $0 \leq k \leq K$.

From Lemma 14, we know that with probability at least $1 - \delta_k$

$$\sup_{u \in \mathcal{F}_k} (\mathcal{E}(u) - \mathcal{E}(u^*)) - (\mathcal{E}_n(u) - \mathcal{E}_n(u^*))$$

$$\leq C \left( \frac{\alpha M^2 \log(2\beta\sqrt{n})}{n} + \sqrt{\frac{M^2 \rho_k \alpha \log(2\beta\sqrt{n})}{n}} + \sqrt{\frac{M^2 \rho_k \log \frac{1}{\delta_k}}{n}} \right. \tag{62}$$

$$\left. + \frac{M^2 \log \frac{1}{\delta_k}}{n} + \sqrt{\frac{a M^2 \rho_k}{n} \log \frac{4b}{M}} \right),$$

where $\alpha, \beta, a, b$ are constants depending on the complexity of $\mathcal{F}$ (see the definitions in Lemma 14).

Note that

$$\begin{aligned}
\rho_k &\leq \max\{\rho_0, 2\rho_{k-1}\} \\
&\leq \max\{\rho_0, 2\|u - u^*\|^2_{H^1(\Omega)}\} \\
&\leq \max\{\rho_0, 2C_P(\mathcal{E}(u) - \mathcal{E}(u^*))\} \\
&\leq \rho_0 + 2C_P(\mathcal{E}(u) - \mathcal{E}(u^*))
\end{aligned} \tag{63}$$

holds for any $u \in \mathcal{F}_k$ and

$$\log \frac{1}{\delta_k} = \log \frac{K+1}{\delta} \leq \log \frac{1}{\delta} + C \log \log \frac{M^2}{\rho_0}. \tag{64}$$

Therefore, setting $\rho_0 = 1/n$, then with (63) for $\rho_k$, for the right terms in (62), we can deduce that the following inequality holds for all $u \in \mathcal{F}_k$.

$$\begin{aligned}
&C\sqrt{\frac{M^2 \rho_k \alpha \log(2\beta\sqrt{n})}{n}} \\
&\leq C\sqrt{\frac{M^2(\rho_0 + 2C_P(\mathcal{E}(u) - \mathcal{E}(u^*)))\alpha \log(2\beta\sqrt{n})}{n}} \\
&\leq C\sqrt{\frac{M^2 \rho_0 \alpha \log(2\beta\sqrt{n})}{n}} + C\sqrt{\frac{2M^2 C_P(\mathcal{E}(u) - \mathcal{E}(u^*))\alpha \log(2\beta\sqrt{n})}{n}} \\
&\leq C\sqrt{\frac{M^2 \rho_0 \alpha \log(2\beta\sqrt{n})}{n}} + C\left(\frac{\mathcal{E}(u) - \mathcal{E}(u^*)}{4C} + \frac{2CM^2 C_P \alpha \log(2\beta\sqrt{n})}{n}\right) \\
&= \frac{\mathcal{E}(u) - \mathcal{E}(u^*)}{4} + C\left(\sqrt{\frac{M^2 \rho_0 \alpha \log(2\beta\sqrt{n})}{n}} + \frac{M^2 C_P \alpha \log(2\beta\sqrt{n})}{n}\right) \\
&\leq \frac{\mathcal{E}(u) - \mathcal{E}(u^*)}{4} + \frac{CM^2 C_P \alpha \log(2\beta\sqrt{n})}{n},
\end{aligned} \tag{65}$$

where the third inequality follows from the basic inequality $2\sqrt{ab} \leq a + b$ for any $a, b \geq 0$.

Similarly, with the upper bound for $\log \frac{1}{\delta_k}$ (64), we can deduce that

$$C\sqrt{\frac{M^2 \rho_k \log \frac{1}{\delta_k}}{n}} \leq \frac{\mathcal{E}(u) - \mathcal{E}(u^*)}{4} + \frac{CC_P M^2 (\log \frac{1}{\delta} + \log \log(nM^2))}{n}, \tag{66}$$

$$\frac{M^2 \log \frac{1}{\delta_k}}{n} \leq \frac{M^2 (\log \frac{1}{\delta} + \log \log(nM^2))}{n} \tag{67}$$

and

$$C\sqrt{\frac{aM^2\rho_k}{n}\log\frac{4b}{M}} \leq \frac{\mathcal{E}(u) - \mathcal{E}(u^*)}{4} + \frac{CM^2C_Pa\log\frac{4b}{M}}{n}. \tag{68}$$

Combining (65), (66), (67), (68) and (62) yields that with probability at least $1 - \delta_k$ for all $u \in \mathcal{F}_k$,

$$(\mathcal{E}(u) - \mathcal{E}(u^*)) - 4(\mathcal{E}_n(u) - \mathcal{E}_n(u^*))$$

$$\leq C\left(\frac{M^2C_P\alpha\log(2\beta\sqrt{n})}{n} + \frac{CC_PM^2(\log\frac{1}{\delta} + \log\log(nM^2))}{n} + \frac{M^2C_Pa\log\frac{4b}{M}}{n}\right). \tag{69}$$

Note that $\sum_{k=0}^{K}\delta_k = \delta$, therefore the above inequality (69) holds with probability at least $1 - \delta$ uniformly for all $u \in \mathcal{F}$, i.e.,

$$\sup_{u\in\mathcal{F}}(\mathcal{E}(u) - \mathcal{E}(u^*)) - 4(\mathcal{E}_n(u) - \mathcal{E}_n(u^*))$$

$$\leq C\left(\frac{M^2C_P\alpha\log(2\beta\sqrt{n})}{n} + \frac{C_PM^2(\log\frac{1}{\delta} + \log\log(nM^2))}{n} + \frac{M^2C_Pa\log\frac{4b}{M}}{n}\right). \tag{70}$$

By taking $\lambda = 4$ and $\delta = e^{-t}$ in (70), together with the error decomposition (55), we have that with probability at least $1 - 4e^{-t}$,

$$\mathcal{E}(u_n) - \mathcal{E}(u^*)$$

$$\leq C\left(\frac{M^2C_P\alpha\log(2\beta\sqrt{n})}{n} + \frac{C_PM^2(t + \log\log(nM^2))}{n} + \frac{M^2C_Pa\log\frac{4b}{M}}{n} + \epsilon_{app}^2 + \frac{M^2t}{n}\right). \tag{71}$$

From Lemma 15, we know that

(1) when $\mathcal{F} = \mathcal{F}_{m,1}(5\|u_P^*\|_{\mathcal{B}_2(\Omega)})$,

$$b = cM, a = cmd, \beta = cM^2, \alpha = cmd,$$

where $c$ is a universal constant.

(2) when $\mathcal{F} = \Phi(N, L, B\|u_P^*\|_{W^{k,\infty}(\Omega)})$,

$$b = Cn, a = CN^2L^2(\log N\log L)^3, \beta = Cn, \alpha = CN^2L^2(\log N\log L)^3,$$

where $n \geq CN^2L^2(\log N\log L)^3$ and $C$ is a constant independent of $N, L$.

Finally, recall the tensorization of variance:

$$Var[f(X_1, \cdots, X_n)] \leq \mathbb{E}\left[\sum_{i=1}^{n} Var_i f(X_1, \cdots, X_n)\right]$$

whenever $X_1, \cdots, X_n$ are independent, where

$$Var_i f(x_1, \cdots, x_n) := Var[f(x_1, \cdots, x_{i-1}, X_i, x_{i+1}, \cdots, x_n)].$$

Combining this fact and the observation of the product structure of $[0,1]^d$ yields that the Poincaré constant is a universal constant.

Hence, the conclusion follows.

**Remark 6.** *In the proof of Theorem 3, we have made an implicit assumption that the empirical processes are measurable. Typically, when considering some empirical process, corresponding functions are Lipschitz continuous with respect to the parameters and the parameter space is separable, thus the measurability holds directly. However, in our setting where ReLU neural networks are used in the DRM, the functions fail to satisfy the Lipschitz continuity with respect to the parameters. Thus, it's necessary to discuss the measurability of the empirical processes. For simplicity, we only consider the two-layer neural networks.*

*Here, we require the concept of pointwise measurability. Recall that a function class $\mathcal{F}$ of measurable functions in $\mathcal{X}$ is pointwise measurable if there exists a countable subset $\mathcal{G} \subset \mathcal{F}$ such that for*

*every $f \in \mathcal{F}$, there exists a sequence $\{g_m\} \in \mathcal{G}$ with $g_m(x) \to f(x)$ for every $x \in \mathcal{X}$ (see Chapter 2.3 in Vaart & Wellner (2023) or Chapter 8.2 in Kosorok (2008)).*

*Note that when applying two-layer neural networks in the DRM, the term $I_{\{\omega \cdot x + t \geq 0\}}$ is not Lipschitz continuous with respect to $\omega$ and $t$. Fortunately, we can adapt the proof of Lemma 8.12 in Kosorok (2008) to show that the function class is pointwise measurable. Specifically, consider the function class*

$$\mathcal{G} = \{I_{\{-\omega \cdot x \leq t\}} : \ \omega \in \partial B_1^d(1) \cap \mathbb{Q}^d, t \in [-1, 1) \cap \mathbb{Q}\},$$

*where $\mathbb{Q}$ is the set consisting of all rationals.*

*Fix $\omega$ and $t$, we can construct $\{(\omega_m, t_m)\}$ as follows: pick $\omega_m \in \partial B_1^d(1) \cap \mathbb{Q}^d$ such that $|\omega_m - \omega|_1 \leq 1/(2m)$ and pick $t_m \in (t + 1/(2m), t + 1/m]$. Now, for any $x \in [0, 1]^d$, we have that*

$$I_{\{-\omega_m \cdot x \leq t_m\}} = I_{\{-\omega \cdot x \leq t_m + (\omega_m - \omega) \cdot x\}}.$$

*Since $|(\omega_m - \omega) \cdot x| \leq |\omega_m - \omega|_1 \leq 1/(2m)$, we have that $r_m := t_m + (\omega_m - \omega) \cdot x - t > 0$ for all $m$ and $r_m \to 0$ as $m \to \infty$. Note that the function $t \mapsto I_{\{u \leq t\}}$ is right-continuous for any $u \in \mathbb{R}$, so that $I_{\{-\omega_m \cdot x \leq t_m\}} \to I_{\{-\omega \cdot x \leq t\}}$ for all $x \in [0, 1]^d$. Thus, the pointwise measurability is established.*

*Therefore, for the function class of two-layer neural networks $\mathcal{F}_{m,1}(B)$,*

$$\mathcal{F}_{m,1}(B) = \left\{\sum_{i=1}^m \gamma_i \sigma(\omega_i \cdot x + t_i) : \ |\omega_i|_1 = 1, t_i \in [-1, 1), \sum_{i=1}^m |\gamma_i| \leq B\right\},$$

*we can pick $\gamma_i, \omega_i, t_i$ to be rationals. To prove the measurability for the empirical processes of the form $\sup_{u \in \mathcal{F}}(\mathcal{E}(u) - \lambda \mathcal{E}_n(u))$, where $\mathcal{F}$ is related to ReLU functions and their gradients, it remains to focus on the term $Pf$.*

*Note that for $u, \hat{u} \in \mathcal{F}_{m,1}(B)$ with the forms*

$$u(x) = \sum_{i=1}^m \gamma_i \sigma(\omega_i \cdot x + t_i), \hat{u}(x) = \sum_{i=1}^m \hat{\gamma}_i \sigma(\hat{\omega}_i \cdot x + \hat{t}_i),$$

*we have that*

$$|P(|\nabla u|^2 - 2fu) - P(|\nabla \hat{u}|^2 - 2f\hat{u})|$$
$$\leq C(P|\nabla u - \nabla \hat{u}| + P|u - \hat{u}|)$$
$$\leq C\left(\sum_{i=1}^m |\gamma_i - \hat{\gamma}_i| + |\omega_i - \hat{\omega}_i|_1 + |t_i - \hat{t}_i| + P|I_{\{\omega_i \cdot x + t_i \geq 0\}} - I_{\{\hat{\omega}_i \cdot x + \hat{t}_i \geq 0\}}|\right).$$

*The dominated convergence theorem implies that*

$$P|I_{\{\omega \cdot x + t \geq 0\}} - I_{\{\omega_m \cdot x + t_m \geq 0\}}| \to 0.$$

*Therefore, with a little abuse of notation, we have $\sup_{u \in \mathcal{F}}(\mathcal{E}(u) - \lambda \mathcal{E}_n(u)) = \sup_{u \in \mathcal{G}}(\mathcal{E}(u) - \lambda \mathcal{E}_n(u))$, which implies that the empirical processes in the proof of Theorem 3 are measurable, as the parameters in $\mathcal{F}$ can be replaced by rationals.*

### A.3 PROOF OF THEOREM 9

**Theorem 9.** *Let $u_S^*$ solve the static Schrödinger and $u_{n,S}$ be the minimizer of the empirical loss $\mathcal{E}_{n,S}$ in the function class $\mathcal{F}$.*

*(1) For $u_S^* \in \mathcal{B}^2(\Omega)$, taking $\mathcal{F} = \mathcal{F}_{m,1}(5\|u_S^*\|_{\mathcal{B}^2(\Omega)})$, then with probability as least $1 - e^{-t}$*

$$\mathcal{E}_S(u_{n,S}) - \mathcal{E}_S(u_S^*) \leq CM^2\left(\frac{md\log n}{n} + \left(\frac{1}{m}\right)^{1 + \frac{2}{3d}} + \frac{t}{n}\right), \tag{72}$$

*where $C$ is a universal constant and $M$ is the upper bound for $\|f\|_{L^\infty}, \|u_S^*\|_{\mathcal{B}^2(\Omega)}, \|V\|_{L^\infty}$.*

*By taking $m = (\frac{n}{d})^{\frac{3d}{2(3d+1)}}$, we have*

$$\mathcal{E}_S(u_{n,S}) - \mathcal{E}_S(u_S^*) \le CM^2 \left( (\frac{d}{n})^{\frac{3d+2}{2(3d+1)}} \log n + \frac{t}{n} \right). \tag{73}$$

*(2) For $u_S^* \in \mathcal{W}^{k,\infty}(\Omega)$, taking $\mathcal{F} = \Phi(N, L, B\|u_S^*\|_{\mathcal{W}^{k,\infty}(\Omega)})$, then with probability at least $1 - e^{-t}$*

$$\mathcal{E}_S(u_{n,S}) - \mathcal{E}_S(u_S^*) \le C \left( \frac{(NL)^2(\log N \log L)^3}{n} + (NL)^{-4(k-1)/d} + \frac{t}{n} \right), \tag{74}$$

*where $n \ge C(NL)^2(\log N \log L)^3$ and $C$ is a constant independent of $N, L, n$.*

*By taking $N = L = n^{\frac{1}{4(d+2(k-1))}}$, we have*

$$\mathcal{E}_S(u_{n,S}) - \mathcal{E}_S(u_S^*) \le C \left( n^{-\frac{2k-2}{d+2k-2}} (\log n)^6 + \frac{t}{n} \right). \tag{75}$$

*Proof.* For the static Schrödinger equation, we can also use the method in the proof of Theorem 3 or other methods in Lu et al. (2021b) and Farrell et al. (2021), due to the similarity between the problem and the generalization error of $L^2$ regression with bounded noise. However, the methods mentioned above are quite complex. Here, we provide a simple proof through a different error decomposition and LRC, which can be easily adapted for other problems with similar strongly convex structures.

As before, in the proof, we write $\mathcal{E}$ and $\mathcal{E}_n$ for the population loss $\mathcal{E}_S$ and empirical loss $\mathcal{E}_{n,S}$ respectively. Additionally, we assume that $|u^*|, |\nabla u^*|, |V|, |f| \le M$ for some positive constant $M$.

Recall that

$$u^* = \arg\min_{u \in H^1(\Omega)} \mathcal{E}(u) := \int_\Omega |\nabla u|^2 + V|u|^2 dx - 2 \int_\Omega fu dx \tag{76}$$

and $u_n$ is the minimal solution to the empirical loss $\mathcal{E}_n$ in the function class $\mathcal{F}$. We also assume that $\sup_{u \in \mathcal{F}} |u|, \sup_{u \in \mathcal{F}} |\nabla u| \le M$.

Through an error decomposition, the same as that for the Poisson equation (55), we have

$$\begin{aligned}
\mathcal{E}(u_n) - \mathcal{E}(u^*) &= \mathcal{E}(u_n) - \lambda \mathcal{E}_n(u_n) + \lambda(\mathcal{E}_n(u_n) - \mathcal{E}_n(u_{\mathcal{F}})) + \lambda \mathcal{E}_n(u_{\mathcal{F}}) - \mathcal{E}(u^*) \\
&\le \mathcal{E}(u_n) - \lambda \mathcal{E}_n(u_n) + \lambda \mathcal{E}_n(u_{\mathcal{F}}) - \mathcal{E}(u^*) \\
&= \mathcal{E}(u_n) - \lambda \mathcal{E}_n(u_n) + \lambda(\mathcal{E}_n(u_{\mathcal{F}}) - \mathcal{E}_n(u^*)) + \lambda \mathcal{E}_n(u^*) - \mathcal{E}(u^*) \\
&= (\mathcal{E}(u_n) - \mathcal{E}(u^*)) - \lambda(\mathcal{E}_n(u_n) - \mathcal{E}_n(u^*)) + \lambda(\mathcal{E}_n(u_{\mathcal{F}}) - \mathcal{E}_n(u^*)) \\
&\le \sup_{u \in \mathcal{F}} [(\mathcal{E}(u) - \mathcal{E}(u^*)) - \lambda(\mathcal{E}_n(u) - \mathcal{E}_n(u^*))] + \lambda(\mathcal{E}_n(u_{\mathcal{F}}) - \mathcal{E}_n(u^*)),
\end{aligned} \tag{77}$$

where the first inequality follows from the definition of $u_n$ and $\lambda$ is a constant to be determined.

Let $\epsilon_{app} := \|u_{\mathcal{F}} - u^*\|_{H^1(\Omega)}$ be the approximation error.

From the Bernstein inequality, we can deduce that with probability at least $1 - e^{-t}$

$$(\mathcal{E}_n(u_{\mathcal{F}}) - \mathcal{E}_n(u^*)) - (\mathcal{E}(u_{\mathcal{F}}) - \mathcal{E}(u^*)) \le \sqrt{\frac{2tVar(g)}{n}} + \frac{t\|g\|_{L^\infty}}{3n}, \tag{78}$$

where

$$g(x) := (|\nabla u_{\mathcal{F}}|^2 + V(x)|u_{\mathcal{F}}(x)|^2 - 2f(x)u_{\mathcal{F}}(x)) - (|\nabla u^*(x)|^2 + V(x)|u^*(x)|^2 - 2f(x)u^*(x)).$$

From the boundedness of $u_{\mathcal{F}}, u^*, \nabla u_{\mathcal{F}}, \nabla u^*, f$ and $V$, we can deduce that $|g| \le 8M^2$ and

$$Var(g) \le Pg^2 \le cM^2 \|u_{\mathcal{F}} - u^*\|_{H^1(\Omega)}^2 = cM^2 \epsilon_{app}^2. \tag{79}$$

Therefore, plugging (79) into (78) yields that with probability at least $1 - e^{-t}$

$$\begin{aligned}
\mathcal{E}_n(u_{\mathcal{F}}) - \mathcal{E}_n(u^*) &\le c\epsilon_{app}^2 + \sqrt{\frac{2tcM^2 \epsilon_{app}^2}{n}} + \frac{8tM^2}{3n} \\
&\le c \left( \epsilon_{app}^2 + \frac{tM^2}{n} \right),
\end{aligned} \tag{80}$$

where the first inequality follows from Proposition 1(2) and the second inequality follows from the mean inequality.

Plugging (80) into the error decomposition (77) yields that

$$\mathcal{E}(u_n) - \mathcal{E}(u^*) \leq \sup_{u \in \mathcal{F}} [(\mathcal{E}(u) - \mathcal{E}(u^*)) - \lambda(\mathcal{E}_n(u) - \mathcal{E}_n(u^*))] + \lambda c \left( \epsilon_{app}^2 + \frac{M^2 t}{n} \right) \qquad (81)$$

holds with probability at least $1 - e^{-t}$.

Note that $(\mathcal{E}(u) - \mathcal{E}(u^*)) - \lambda(\mathcal{E}_n(u) - \mathcal{E}_n(u^*))$ can be rewritten as

$$(\mathcal{E}(u) - \mathcal{E}(u^*)) - \lambda(\mathcal{E}_n(u) - \mathcal{E}_n(u^*)) = Ph - \lambda P_n h, \qquad (82)$$

where $h(x) := (|\nabla u(x)|^2 + V(x)|u(x)|^2 - 2f(x)u(x)) - (|\nabla u^*(x)|^2 + V(x)|u^*(x)|^2 - 2f(x)u^*(x))$. And the form (82) motivates the use of LRC.

To invoke the LRC, we begin by defining the function class

$$\mathcal{H} := \{(|\nabla u(x)|^2 + V(x)|u(x)|^2 - 2f(x)u(x)) - (|\nabla u^*(x)|^2 + V(x)|u^*(x)|^2 - 2f(x)u^*(x)) : u \in \mathcal{F}\}$$

and a functional on $\mathcal{H}$ as $T(h) := Ph^2$. It is easy to check that

$$Var(h) \leq T(h) \leq cM^2 Ph, \qquad (83)$$

as $Ph^2 \leq cM^2 \|u - u^*\|_{H^1(\Omega)}^2 \leq cM^2(\mathcal{E}(u) - \mathcal{E}(u^*)) = cM^2 Ph$. It implies that the functional $T$ satisfies the condition of Theorem 3.3 in Bartlett et al. (2005).

Following the procedure of Theorem 3.3 in Bartlett et al. (2005), we are going to seek a sub-root function and compute its fixed point.

Define the sub-root function

$$\psi(r) := 80M^2 \mathbb{E}\mathcal{R}_n(h \in star(\mathcal{H}, 0) : Ph^2 \leq r) + 704 \frac{M^4 \log n}{n}, \qquad (84)$$

where $star(\mathcal{H}, 0) := \{\alpha h : \alpha \in [0, 1], h \in \mathcal{H}\}$ and invoking the star-hull of $\mathcal{H}$ around 0 is to make $\psi$ to be a sub-root function.

Next, our goal is to bound the fixed point of $\psi$.

If $r \geq \psi(r)$, then Corollary 2.2 in Bartlett et al. (2005) implies that with probability at least $1 - \frac{1}{n}$,

$$\{h \in star(\mathcal{H}, 0) : Ph^2 \leq r\} \subset \{h \in star(\mathcal{H}, 0) : P_n h^2 \leq 2r\},$$

and thus

$$\mathbb{E}\mathcal{R}_n(h \in star(\mathcal{H}, 0) : Ph^2 \leq r) \leq \mathbb{E}\mathcal{R}_n(h \in star(\mathcal{H}, 0) : P_n h^2 \leq 2r) + \frac{8M^2}{n}. \qquad (85)$$

Assume that $r^*$ is the fixed point of $\psi$, then

$$r^* = \psi(r^*) \leq cM^2 \mathbb{E}\mathcal{R}_n(h \in star(\mathcal{H}, 0) : P_n h^2 \leq 2r^*) + c \frac{M^4 \log n}{n}, \qquad (86)$$

where we use a universal constant $c$ to represent the upper bound for the constants in the definition of $\psi(r)$, i.e. (84).

To estimate the first term in (86), we need the assumption about the empirical covering number of $\mathcal{H}$.

**Assumption 1.** *For any $\epsilon > 0$, assume that*

$$\mathcal{N}(\mathcal{H}, L_2(P_n), \epsilon) \leq \left( \frac{\beta}{\epsilon} \right)^{\alpha} \quad a.s.,$$

*for some constant $\beta > \sup_{h \in \mathcal{H}} |h|$.*

Then by Dudley's theorem,

$$\mathbb{E}\mathcal{R}_n(h \in star(\mathcal{H}, 0) : P_n h^2 \leq 2r^*)$$

$$\leq \frac{c}{\sqrt{n}}\mathbb{E}\int_0^{\sqrt{2r^*}} \sqrt{\log\mathcal{N}(\epsilon, star(\mathcal{H},0), L_2(P_n))}d\epsilon$$

$$\leq \frac{c}{\sqrt{n}}\mathbb{E}\int_0^{\sqrt{2r^*}} \sqrt{\log\mathcal{N}(\frac{\epsilon}{2}, \mathcal{H}, L_2(P_n))\left(\frac{2}{\epsilon}+1\right)}d\epsilon$$

$$\leq c\sqrt{\frac{\alpha}{n}}\int_0^{\sqrt{2r^*}} \sqrt{\log\left(\frac{\beta}{\epsilon}\right)}d\epsilon$$

$$= c\beta\sqrt{\frac{\alpha}{n}}\int_0^{\frac{\sqrt{2r^*}}{\beta}} \sqrt{\log\left(\frac{1}{\epsilon}\right)}d\epsilon$$

$$\leq c\sqrt{\frac{\alpha}{n}}\sqrt{r^*\log\left(\frac{\beta}{\sqrt{r^*}}\right)}$$

$$\leq c\sqrt{\frac{\alpha}{n}}\sqrt{r^*\log\left(\frac{\sqrt{n}\beta}{M^2}\right)},$$

where the fourth inequality follows from Lemma 13 and the last inequality follows by the fact that $r^* = \psi(r^*) \geq c\frac{M^4\log n}{n}$.

Therefore,

$$r^* \leq cM^2\sqrt{\frac{\alpha}{n}}\sqrt{r^*\log\left(\frac{\sqrt{n}\beta}{M^2}\right)} + c\frac{M^4\log n}{n},$$

which implies

$$r^* \leq cM^4\left(\frac{\alpha}{n}\log\left(\frac{\sqrt{n}\beta}{M^2}\right) + \frac{\log n}{n}\right).$$

The final step is to estimate the empirical covering numbers of the function classes of two-layer neural networks and deep neural networks, i.e., to determine $\alpha$ and $\beta$ for $\mathcal{F} = \mathcal{F}_m(5\|u_S^*\|_{\mathcal{B}^2(\Omega)})$ and $\mathcal{F} = \Phi(N, L, B\|u_S^*\|_{W^{1,\infty}(\Omega)})$.

(1) When $\mathcal{F} = \mathcal{F}_{m,1}(5\|u_S^*\|_{\mathcal{B}^2(\Omega)})$, estimation of the covering number of $\mathcal{H}$ is almost same as the estimation of $\mathcal{G}$ for the two-layer neural networks in Lemma 15(1). It is not difficult to deduce that $\alpha = cmd$, $\beta = cM^2$. For simplicity, we omit the proof.

(2) When $\mathcal{F} = \Phi(N, L, B\|u_S^*\|_{W^{k,\infty}(\Omega)})$, we can also deduce that $\alpha = CN^2L^2(\log N\log L)^3, \beta = Cn$ by a similar method as that in Lemma 15(2).

As a result, given the upper bound for $r^*$, applying Theorem 3.3 in Bartlett et al. (2005) with $\lambda = 2$ allows us to reach the conclusion. □

# B    PROOF OF SECTION 3

## B.1    PROOF OF PROPOSITION 4

*Proof.* (1) The proof mainly follows the procedure in the proof the Proposition 2(1), but the tools from the FEM may not work for ReLU$^2$ functions. Therefore, we turn to use Taylor's theorem with integral remainder, which enables us to establish a connection between the one-dimensional $C^2$ functions and the ReLU$^2$ functions. And the method has been also used in Klusowski & Barron (2018); Xu (2020).

Recall that Taylor's theorem with integral remainder states that for $f : \mathbb{R} \to \mathbb{R}$ that has $k + 1$ continuous derivatives in some neighborhood $U$ of $x = a$, then for $x \in U$

$$f(x) = f(a) + f'(a)(x - a) + \cdots + \frac{f^{(k)}(a)}{k!}(x-a)^k + \int_a^x f^{(k+1)}(t)\frac{(x-t)^k}{k!}dt.$$

Similar as the proof of Proposition 2(1), for any $f \in \mathcal{B}^3(\Omega)$, we have

$$f(x) = \int_{\mathbb{R}^d} g(x, \omega) \Lambda(d\omega),$$

where $B = \int_{\mathbb{R}^d} (1 + |\omega|_1)^3 |\hat{f}(\omega)| d\omega$, $\Lambda(d\omega) = (1 + |\omega|_1)^3 |\hat{f}(\omega)|/B$ and

$$g(x, \omega) = \frac{B \cos(\omega \cdot x + \theta(\omega))}{(1 + |\omega|_1)^3}.$$

Therefore, $f$ is in the $H^2(\Omega)$ closure of the convex hull of the function class

$$\mathcal{G}_{cos}(B) := \left\{ \frac{B \cos(\omega \cdot x + t)}{(1 + |\omega|_1)^3} : \omega \in \mathbb{R}^d, t \in \mathbb{R} \right\}.$$

Note that any function $g(x, \omega) = \frac{B \cos(\omega \cdot x + t)}{(1 + |\omega|_1)^3}$ is a composition of a one-dimensional function $g(z) = \frac{B \cos(|\omega|_1 z + t)}{(1 + |\omega|_1)^3}$ and a linear function $z = \frac{\omega}{|\omega|_1} \cdot x$ with value in $[-1, 1]$. Therefore, in order to prove that $f$ is in the $H^2(\Omega)$ closure of the convex hull of the function class $\mathcal{F}_{\sigma_2}(cB) \cup \mathcal{F}_{\sigma_2}(-cB) \cup \{0\}$, it suffices to prove that $g$ is in the $H^2([-1, 1])$ closure of the convex hull of the function class $\mathcal{F}^1_{\sigma_2}(cB) \cup \mathcal{F}^1_{\sigma_2}(-cB) \cup \{0\}$, where

$$\mathcal{F}_{\sigma_2}(b) := \{b\sigma_2(\omega \cdot x + t) : |\omega|_1 = 1, t \in [-1, 1]\} \ and \ \mathcal{F}^1_{\sigma_2}(b) := \{b\sigma_2(\epsilon z + t) : \epsilon = +1 \ or \ 1, t \in [-1, 1]\}$$

for any constant $b \in \mathbb{R}$.

For

$$g(z) = \frac{B \cos(|\omega|_1 z + t)}{(1 + |\omega|_1)^3} = \frac{B(\cos(|\omega|_1 z) \cos t - \sin(|\omega|_1 z) \sin t)}{(1 + |\omega|_1)^3}$$

with $z \in [-1, 1]$, applying Taylor's theorem with integral remainder for $\cos(|\omega|_1 z)$ and $\sin(|\omega|_1 z)$ at the point 0, we have

$$\cos(|\omega|_1 z) = 1 - \frac{|\omega|_1^2}{2} z^2 + \int_0^z |\omega|_1^3 \sin(|\omega|_1 s) \frac{(z - s)^2}{2} ds$$

and

$$\sin(|\omega|_1 z) = |\omega|_1 z - \int_0^z |\omega|_1^3 \cos(|\omega|_1 s) \frac{(z - s)^2}{2} ds.$$

Note that $z^2, z, 1$ can be represented by combinations of $\mathrm{ReLU}^2$ functions, specifically

$$z^2 = \sigma_2(z) + \sigma_2(-z), z = \frac{(z + 1)^2 - (z - 1)^2}{4}, 1 = \frac{(z + 1)^2 + (z - 1)^2}{2} - z^2.$$

Therefore, we only need to prove that the integral remainders are in the $H^2([-1, 1])$ closure of the convex hull of the function class $\mathcal{F}^1_{\sigma_2}(cB) \cup \mathcal{F}^1_{\sigma_2}(-cB) \cup \{0\}$. In the following, the constant $c$ may change line by line, but it is still a universal constant, so we still denote it by $c$.

Due to the form of the integral remainder, we consider the general form $h(z) = \int_0^z \varphi(s)(z - s)^2 ds$ with $\varphi \in C([-1, 1])$. By the fact that $(z - s)^2 = (z - s)_+^2 + (-z + s)_+^2$, we have

$$\int_0^z \varphi(s)(z - s)^2 ds = \int_0^z \varphi(s)(z - s)_+^2 ds + \int_0^z \varphi(s)(-z + s)_+^2 ds := A_1 + A_2$$

In the following, we aim to prove that

$$A_1 + A_2 = \int_0^1 \varphi(s)(z - s)_+^2 ds - \int_0^1 \varphi(-s)(-z - s)_+^2 ds := B_1 + B_2,$$

which enables the method used in the proof of Proposition 2(1) to be feasible.

(1) When $z \geq 0$, it is easy to obtain that

$$A_1 = \int_0^z \varphi(s)(z - s)_+^2 ds = \int_0^1 \varphi(s)(z - s)_+^2 ds = B_1, \ and \ A_2 = 0, B_2 = 0. \qquad (87)$$

Therefore, $A_1 + A_2 = B_1 + B_2$.

(2) When $z < 0$, it is easy to check that $A_1 = B_1 = 0$. Therefore, it remains only to check that $A_2 = B_2$.

For $A_2$, we can deduce that

$$
\begin{aligned}
A_2 &= \int_0^z \varphi(s)(-z+s)_+^2 ds \\
&= -\int_z^0 \varphi(s)(-z+s)_+^2 ds \\
&= -\left[ \int_z^0 \varphi(s)(-z+s)_+^2 ds + \int_{-1}^z \varphi(s)(-z+s)_+^2 ds \right] \\
&= -\int_{-1}^0 \varphi(s)(-z+s)_+^2 ds \\
&= -\int_0^1 \varphi(-y)(-z-y)_+^2 dy = B_2,
\end{aligned}
\tag{88}
$$

where the third equality follows by that $\int_{-1}^z \varphi(s)(-z+s)_+^2 ds = 0$ and the fifth equality is due to the variable substitution $s = -y$.

Combining (87) and (88), we can deduce that

$$
h(z) = \int_0^z \varphi(s)(z-s)^2 ds = \int_0^1 \varphi(s)(z-s)_+^2 ds - \int_0^1 \varphi(-s)(-z-s)_+^2 ds. \tag{89}
$$

The next step is to prove that $h$ is the $H^2([-1,1])$ closure of convex hull of $\mathcal{F}_{\sigma_2}^1(cB) \cup \mathcal{F}_{\sigma_2}^1(-cB) \cup \{0\}$.

Let $h_1(z) = \int_0^1 \varphi(s)(z-s)_+^2 ds$, $h_2(z) = \int_0^1 \varphi(-s)(-z-s)_+^2 ds$, then $h(z) = h_1(z) - h_2(z)$.

Note that $h_1'(z) = \int_0^1 2\varphi(s)(z-s)_+ ds$ and $h_1''(z) = \int_0^1 2\varphi(s) I_{\{z-s \geq 0\}} ds$ a.e., since $(z-s)_+$ is differentiable for $s$ a.e. .

Let $\{s_i\}_{i=1}^n$ be an i.i.d. sequence of random variables distributed according the uniform distribution of the interval $[0,1]$, then by Fubini's theorem

$$
\mathbb{E} \left\| h_1(z) - \sum_{i=1}^n \frac{\varphi(s_i)(z-s_i)_+^2}{n} \right\|_{H^2([-1,1])}^2
$$

$$
= \int_{-1}^1 \mathbb{E} \left[ |h_1(z) - \sum_{i=1}^n \frac{\varphi(s_i)(z-s_i)_+^2}{n}|^2 + |h_1'(z) - \sum_{i=1}^n \frac{2\varphi(s_i)(z-s_i)_+}{n}|^2 + +|h_1''(z) - \sum_{i=1}^n \frac{2\varphi(s_i)I_{\{z-s_i \geq 0\}}}{n}|^2 \right] dz
$$

$$
= \int_{-1}^1 \frac{Var(\varphi(\cdot)(z-\cdot)_+^2) + Var(2\varphi(\cdot)(z-\cdot)_+) + Var(2\varphi(\cdot)I_{\{z-\cdot \geq 0\}})}{n} dz
$$

$$
\leq \frac{C}{n},
$$

where the last inequality follows from the boundedness of $\varphi$. And the same conclusion also holds for $h_2(z)$ and $h(z)$. Therefore, we can deduce that $h$ is in the $H^2([-1,1])$ closure of convex hull of the function class $\mathcal{F}_{\sigma_2}^1(cB) \cup \mathcal{F}_{\sigma_2}^1(-cB) \cup \{0\}$.

Then applying the variable substitution yields that for any $f \in \mathcal{B}^3(\Omega)$ and $\epsilon > 0$, there exists a two-layer $\sigma_2$ neural network such that

$$
\| f(x) - \sum_{i=1}^m a_i \sigma_2(\omega_i \cdot x + t_i) \|_{H^2(\Omega)} \leq \epsilon, \tag{90}
$$

where $|\omega_i|_1 = 1, |t_i| \leq 1, \sum_{i=1}^m |a_i| \leq c\|f\|_{\mathcal{B}^3(\Omega)}$ and $c$ is a universal constant.

Just as the proof of Proposition 2(1), it remains only to estimate the metric entropy of the function class

$$\mathcal{F}_2 := \{\sigma_2(\omega \cdot x + t) : |\omega|_1 = 1, t \in [-1, 1]\}$$

under the $H^2$ norm.

For $(\omega_1, t_1), (\omega_2, t_2) \in \partial B_1^d(1) \times [-1, 1]$, we have

$$\|\sigma_2(\omega_1 \cdot x + t_1) - \sigma_2(\omega_2 \cdot x + t_2)\|_{H^2(\Omega)}^2$$

$$= \|\sigma_2(\omega_1 \cdot x + t_1) - \sigma_2(\omega_2 \cdot x + t_2)\|_{L^2(\Omega)}^2 + \||2\omega_1\sigma(\omega_1 \cdot x + t_1) - 2\omega_2\sigma(\omega_2 \cdot x + t_2)|\|_{L^2(\Omega)}^2$$

$$+ \sum_{i=1}^{d}\sum_{j=1}^{d} \|2\omega_{1i}\omega_{1j}I_{\{\omega_1 \cdot x + t_1 \geq 0\}} - 2\omega_{2i}\omega_{2j}I_{\{\omega_2 \cdot x + t_2 \geq 0\}}\|_{L^2(\Omega)}^2$$

$$:= (i) + (ii) + (iii),$$

where we denote the $i$-th element of the vector $\omega_k$ by $\omega_{ki}$ for $k = 1, 2, 1 \leq i \leq d$.

For $(i)$, since $\sigma_2$ is 4-Lipschitz in $[-2, 2]$,

$$\begin{aligned}(i) &= \|\sigma_2(\omega_1 \cdot x + t_1) - \sigma_2(\omega_2 \cdot x + t_2)\|_{L^2(\Omega)}^2 \\ &\leq 16\|(\omega_1 - \omega_2) \cdot x + (t_1 - t_2)\|_{L^2(\Omega)}^2 \\ &\leq 32(|\omega_1 - \omega_2|_1^2 + |t_1 - t_2|^2).\end{aligned} \tag{91}$$

For $(ii)$,

$$\begin{aligned}(ii) &= \||2\omega_1\sigma(\omega_1 \cdot x + t_1) - 2\omega_2\sigma(\omega_2 \cdot x + t_2)|\|_{L^2(\Omega)}^2 \\ &= 2\||(\omega_1 - \omega_2)\sigma(\omega_1 \cdot x + t_1) + \omega_2(\sigma(\omega_1 \cdot x + t_1) - \sigma(\omega_2 \cdot x + t_2))|\|_{L^2(\Omega)}^2 \\ &\leq 4\||(\omega_1 - \omega_2)\sigma(\omega_1 \cdot x + t_1)|\|_{L^2(\Omega)}^2 + 4\||\omega_2(\sigma(\omega_1 \cdot x + t_1) - \sigma(\omega_2 \cdot x + t_2))|\|_{L^2(\Omega)}^2 \\ &\leq 16|\omega_1 - \omega_2|_1^2 + 8(|\omega_1 - \omega_2|_1^2 + |t_1 - t_2|^2),\end{aligned} \tag{92}$$

where the first inequality follows from the mean inequality and the boundedness of $\sigma$.

For $(iii)$,

$$\begin{aligned}(iii) &= \sum_{i=1}^{d}\sum_{j=1}^{d} \|2\omega_{1i}\omega_{1j}I_{\{\omega_1 \cdot x + t_1 \geq 0\}} - 2\omega_{2i}\omega_{2j}I_{\{\omega_2 \cdot x + t_2 \geq 0\}}\|_{L^2(\Omega)}^2 \\ &= 4\sum_{i=1}^{d}\sum_{j=1}^{d} \|(\omega_{1i}\omega_{1j} - \omega_{2i}\omega_{2j})I_{\{\omega_1 \cdot x + t_1 \geq 0\}} + \omega_{2i}\omega_{2j}(I_{\{\omega_1 \cdot x + t_1 \geq 0\}} - I_{\{\omega_2 \cdot x + t_2 \geq 0\}})\|_{L^2(\Omega)}^2 \\ &\leq 8\sum_{i=1}^{d}\sum_{j=1}^{d} |\omega_{1i}\omega_{1j} - \omega_{2i}\omega_{2j}|^2 + (\omega_{2i}\omega_{2j})^2\|I_{\{\omega_1 \cdot x + t_1 \geq 0\}} - I_{\{\omega_2 \cdot x + t_2 \geq 0\}}\|_{L^2(\Omega)}^2 \\ &\leq 8\sum_{i=1}^{d}\sum_{j=1}^{d} |\omega_{1i} - \omega_{2i}|^2|\omega_{1j}|^2 + |\omega_{1j} - \omega_{2j}|^2|\omega_{2i}|^2 + (\omega_{2i}\omega_{2j})^2\|I_{\{\omega_1 \cdot x + t_1 \geq 0\}} - I_{\{\omega_2 \cdot x + t_2 \geq 0\}}\|_{L^2(\Omega)}^2 \\ &\leq 16|\omega_1 - \omega_2|_1^2 + 8\|I_{\{\omega_1 \cdot x + t_1 \geq 0\}} - I_{\{\omega_2 \cdot x + t_2 \geq 0\}}\|_{L^2(\Omega)}^2,\end{aligned} \tag{93}$$

where the last inequality follows from the fact that $|\omega_1| \leq |\omega_1|_1 = 1, |\omega_2| \leq |\omega_2|_1 = 1$.

Combining the upper bounds for $(i), (ii), (iii)$, we obtain that

$$\|\sigma_2(\omega_1 \cdot x + t_1) - \sigma_2(\omega_2 \cdot x + t_2)\|_{H^2(\Omega)}^2 \leq 72(|\omega_1 - \omega_2|_1^2 + |t_1 - t_2|^2) + 8\|I_{\{\omega_1 \cdot x + t_1 \geq 0\}} - I_{\{\omega_2 \cdot x + t_2 \geq 0\}}\|_{L^2(\Omega)}^2. \tag{94}$$

Therefore, based on the same method used in the proof of Proposition 8, we can deduce that

$$\epsilon_n(\mathcal{F}_2) \leq cn^{-\frac{1}{3d}}.$$

Finally, applying Theorem 1 in Makovoz (1996) (Lemma 10) yields the conclusion for $f$ in $\mathcal{B}^3(\Omega)$.

(2) Recall that based on the spline theory, Belomestny et al. (2023) has demonstrated the approximation rates for Hölder continuous functions with sparse ReLU[2] neural networks. Then Belomestny et al. (2024) extended these results for sparse ReLU[3] neural networks. In fact, the approximation results also hold for Sobolev functions, we only need to replace the Theorem 3 in Belomestny et al. (2023) with the results from Schumaker (2007) on approximating Sobolev functions with multivariate splines. For simplicity, we omit the proof. $\square$

## B.2 Proof of Theorem 5

Before the proof, we first provide some preliminaries about the entropy method, which is a common method to derive concentration inequalities. For $\Omega = \prod_{k=1}^n \Omega_k, \mu = \prod_{k=1}^n \mu_k$, where $\mu_k$ is a probability measure, let $(\Omega, \Sigma)$ be a measurable space and $\mathcal{A}(\Omega)$ denote the algebra of bounded, measurable real valued function on $\Omega$. For $f \in \mathcal{A}, \beta \in \mathbb{R}$, define the expectation functional as

$$\mathbb{E}_{\beta f}[g] = \frac{\mathbb{E}[g e^{\beta f}]}{\mathbb{E}[e^{\beta f}]} = Z_{\beta f}^{-1} \mathbb{E}[g e^{\beta f}], for\ g \in \mathcal{A},$$

where $Z_{\beta f} = \mathbb{E}[e^{\beta f}]$ is the normalizing quantity. Then, we can define the entropy as

$$Ent_f(\beta) := \beta \mathbb{E}_{\beta f}[f] - \log Z_{\beta f}.$$

The connection between the entropy and the exponential moment makes the entropy method popular for deriving concentration inequalities, i.e.,

$$\log \mathbb{E}[e^{\beta(f - \mathbb{E}f)}] \leq \beta \int_0^\beta \frac{Ent_f(\gamma)}{\gamma^2} d\gamma \tag{95}$$

holds for any $f \in \mathcal{A}$ and $\beta \geq 0$.

For any real-valued function $F$ on $\Omega$ and $y \in \Omega_k$ for $k \in \{1, \cdots, n\}$, define the substitution operator $S_y^k$ on $F$ as

$$S_y^k(F)(x_1, \cdots, x_n) := F(x_1, \cdots, x_{k-1}, y, x_{k+1}, \cdots, x_n), \tag{96}$$

i.e., the $k$-th argument is simply replaced by $y$. And define the operator $V_+^2 : \mathcal{A} \to \mathcal{A}$ by

$$V_+^2 F(x) := \sum_{k=1}^n \mathbb{E}_{y \sim \mu_k} \left[ \left( (F(x) - S_y^k F(x))_+ \right)^2 \right]. \tag{97}$$

*Proof.* Assume that $\sup_{1 \leq t \leq T} \sup_{x \in \mathcal{X}_t} |f_t(x)| \leq b$ and $\frac{1}{T} \sup_{\boldsymbol{f} \in \boldsymbol{\mathcal{F}}} \sum_{t=1}^T Var(f_t(X_t^1)) \leq r$.

Let

$$Z := \sup_{\boldsymbol{f} \in \boldsymbol{\mathcal{F}}} \frac{1}{T} \sum_{t=1}^T (P_n - P) f_t = \sup_{\boldsymbol{f} \in \boldsymbol{\mathcal{F}}} \frac{1}{T} \sum_{t=1}^T \frac{1}{N_t} \sum_{i=1}^{N_t} f_t(X_t^i) - \mathbb{E}f_t(X_t^i) \tag{98}$$

and

$$F(x) := \frac{1}{2b} \sup_{\boldsymbol{f} \in \boldsymbol{\mathcal{F}}} \sum_{t=1}^T \frac{n}{N_t} \sum_{i=1}^{N_t} f_t(x_t^i) - \mathbb{E}f_t(X_t^i), \tag{99}$$

where $n = \min_{1 \leq i \leq T} N_t$ and $x = (x_1^1, \cdots, x_t^i, \cdots, x_T^{N_t})$.

Define

$$W(x) := \frac{1}{4b^2} \sup_{\boldsymbol{f} \in \boldsymbol{\mathcal{F}}} \sum_{t=1}^T \frac{n^2}{N_t^2} \sum_{i=1}^{N_t} (f_t(x_t^i) - \mathbb{E}f_t(X_t^i))^2 + \mathbb{E}(f_t(X_t^i) - \mathbb{E}f_t(X_t^i))^2. \tag{100}$$

Similar to Theorem 38 in Maurer (2021) for the single task, fix $(x_{t,i})_{1 \leq t \leq T, 1 \leq i \leq n}$ and assume that the maximum in the definition of $F$ is achieved at $\hat{\boldsymbol{f}} = (\hat{f}_1, \cdots, \hat{f}_T) \in \boldsymbol{\mathcal{F}}$.

Then for any $y$, $(F(x) - S_y^{t,i}F(x))_+ \le \frac{n}{2bN_t}(\hat{f}_t(x_{t,i}) - \hat{f}_t(y))_+$, therefore

$$
\begin{aligned}
V_+^2 F(x) &= \sum_{t=1}^{T} \sum_{i=1}^{N_t} \mathbb{E}_{y \sim \mu_{t,i}} \left[ \left( (F - S_y^{t,i}F)_+ \right)^2 \right] \\
&\le \frac{1}{4b^2} \sum_{t=1}^{T} \frac{n^2}{N_t^2} \sum_{i=1}^{N_t} \mathbb{E}_{y \sim \mu_{t,i}} \left[ \left( (\hat{f}_t(x_t^i) - \hat{f}_t(y))_+ \right)^2 \right] \\
&\le \frac{1}{4b^2} \sum_{t=1}^{T} \frac{n^2}{N_t^2} \sum_{i=1}^{N_t} \mathbb{E}_{y \sim \mu_{t,i}} \left[ (\hat{f}_t(x_t^i) - \hat{f}_t(y))^2 \right] \\
&= \frac{1}{4b^2} \sum_{t=1}^{T} \frac{n^2}{N_t^2} \sum_{i=1}^{N_t} \mathbb{E}_{y \sim \mu_{t,i}} \left[ \left( \hat{f}_t(x_t^i) - \mathbb{E}\hat{f}_t(X_t^i) - (\hat{f}_t(y) - \mathbb{E}\hat{f}_t(X_t^i)) \right)^2 \right] \\
&= \frac{1}{4b^2} \sum_{t=1}^{T} \frac{n^2}{N_t^2} \sum_{i=1}^{N_t} (\hat{f}_t(x_t^i) - \mathbb{E}\hat{f}_t(X_t^i))^2 + \mathbb{E}(\hat{f}_t(X_t^i) - \mathbb{E}\hat{f}_t(X_t^i))^2 \\
&\le W,
\end{aligned}
\tag{101}
$$

where $X_t^i$ follows the distribution $\mu_t^i$, i.e., $\mu_t^i = \mu_t$.

Therefore, $V_+^2 F \le W$. Then equation (26) in Maurer (2021) yields that for $0 < \gamma \le \beta < 2$,

$$
Ent_F(\gamma) \le \frac{\gamma}{2-\gamma} \log \mathbb{E}e^{\gamma V_+^2 F} \le \frac{\gamma}{2-\gamma} \log \mathbb{E}e^{\gamma W}.
\tag{102}
$$

Next, we are going the prove that $W$ is self-bounding, so that Lemma 32 (i) in Maurer (2021) can be applied to bound $\mathbb{E}e^{\gamma W}$. Assume that the maximum in the definition of $W$ is achieved at $\bar{\boldsymbol{f}} = (\bar{f}_1, \cdots, \bar{f}_T) \in \boldsymbol{\mathcal{F}}$, then for any $y$,

$$
(W - S_y^{t,i}W)_+ \le \frac{n^2}{4b^2 N_t^2}((\bar{f}_t(x_t^i) - \mathbb{E}\bar{f}_t(X_t^i))^2 - (\bar{f}_t(y) - \mathbb{E}\bar{f}_t(X_t^i))^2)_+ \le \frac{n^2}{4b^2 N_t^2}(\bar{f}_t(x_t^i) - \mathbb{E}\bar{f}_t(X_t^i))^2,
$$

therefore

$$
\begin{aligned}
V_+^2 W(x) &= \sum_{t=1}^{T} \sum_{i=1}^{N_t} \mathbb{E}_{y \sim \mu_{t,i}} (W(x) - S_y^{t,i}W(x))_+^2 \\
&\le \frac{1}{16b^4} \sum_{t=1}^{T} \frac{n^4}{N_t^4} \sum_{i=1}^{N_t} \mathbb{E}_{y \sim \mu_{t,i}} \left[ ((\bar{f}_t(x_t^i) - \mathbb{E}\bar{f}_t(X_t^i))^2 - (\bar{f}_t(y) - \mathbb{E}\bar{f}_t(X_t^i))^2)_+^2 \right] \\
&\le \frac{1}{16b^4} \sum_{t=1}^{T} \frac{n^4}{N_t^4} \sum_{i=1}^{N_t} (\bar{f}_t(x_t^i) - \mathbb{E}\bar{f}_t(X_t^i))^4 \\
&\le \frac{1}{4b^2} \sum_{t=1}^{T} \frac{n^2}{N_t^2} \sum_{i=1}^{N_t} (\bar{f}_t(x_t^i) - \mathbb{E}\bar{f}_t(X_t^i))^2 \\
&\le W.
\end{aligned}
\tag{103}
$$

Combining (103) with Lemma 32(i) in Maurer (2021), we have

$$
\log \mathbb{E}[e^{\gamma W}] \le \frac{\gamma^2 \mathbb{E}[W]}{2-\gamma} + \gamma \mathbb{E}[W] = \frac{\gamma \mathbb{E}[W]}{1 - \gamma/2}.
\tag{104}
$$

Plugging (104) into (102) yields that

$$
Ent_F(\gamma) \le \frac{\gamma}{2-\gamma} \log \mathbb{E}[e^{\gamma W}] \le \frac{\gamma}{2-\gamma}\left(\frac{\gamma \mathbb{E}[W]}{1 - \gamma/2}\right) = \frac{\gamma^2}{(1-\gamma/2)^2} \frac{\mathbb{E}[W]}{2}.
\tag{105}
$$

Combining (95) and (105), we can conclude that

$$
\begin{aligned}
\log \mathbb{E} e^{\beta(F-\mathbb{E}F)} &\leq \beta \int_0^\beta \frac{Ent_F(\gamma)}{\gamma^2} d\gamma \\
&\leq \beta \frac{\mathbb{E}[W]}{2} \int_0^\beta \frac{1}{(1-\gamma/2)^2} d\gamma \\
&= \frac{\beta^2}{1-\beta/2} \frac{\mathbb{E}[W]}{2}.
\end{aligned}
\tag{106}
$$

In fact, the above inequality implies that $F$ is a sub-gamma random variable. Thus with the following lemma, we can derive the concentration inequality for $F$.

**Lemma 2.** *Let $Z$ be a random variable, $A, B > 0$ be some constants. If for any $\lambda \in (0, 1/B)$ it holds*

$$
\log \mathbb{E}[e^{\lambda(Z-\mathbb{E}Z)}] \leq \frac{A\lambda^2}{2(1-B\lambda)},
$$

*then for all $x \geq 0$,*

$$
P(Z \geq \mathbb{E}Z + \sqrt{2Ax} + Bx) \leq e^{-x}.
$$

Applying Lemma 2 with $A = \mathbb{E}[W], B = 1/2$ for $F$, we can deduce that with probability at least $1 - e^{-x}$

$$
F \leq \mathbb{E}F + \sqrt{2x\mathbb{E}W} + \frac{x}{2}.
\tag{107}
$$

From the definitions of $F$ and $Z$, i.e. (99) and (98), we have $Z = \frac{2bF}{nT}$, then with probability at least $1 - e^{-x}$

$$
Z \leq \mathbb{E}Z + \frac{2b}{nT}\sqrt{2x\mathbb{E}W} + \frac{bx}{nT}.
\tag{108}
$$

Note that $\mathbb{E}Z \leq 2\mathcal{R}(\mathcal{F})$ and

$$
\begin{aligned}
\mathbb{E}W &= \frac{1}{4b^2} \mathbb{E} \sup_{\boldsymbol{f}\in\boldsymbol{\mathcal{F}}} \sum_{t=1}^T \frac{n^2}{N_t^2} \sum_{i=1}^{N_t} (f_t(X_t^i) - \mathbb{E}f_t(X_t^i))^2 + \mathbb{E}(f_t(X_t^i) - \mathbb{E}f_t(X_t^i))^2 \\
&= \frac{1}{4b^2} \mathbb{E} \sup_{\boldsymbol{f}\in\boldsymbol{\mathcal{F}}} \sum_{t=1}^T \frac{n^2}{N_t^2} \sum_{i=1}^{N_t} \left[ [(f_t(X_t^i) - \mathbb{E}f_t(X_t^i))^2 - \mathbb{E}(f_t(X_t^i) - \mathbb{E}f_t(X_t^i))^2] + 2\mathbb{E}(f_t(X_t^i) - \mathbb{E}f_t(X_t^i))^2 \right] \\
&\leq \frac{1}{4b^2} \left( 2\mathbb{E} \sup_{\boldsymbol{f}\in\boldsymbol{\mathcal{F}}} \sum_{t=1}^T \frac{n^2}{N_t^2} \sum_{i=1}^{N_t} \sigma_t^i (f_t(X_t^i) - \mathbb{E}f_t(X_t^i))^2 + \sup_{\boldsymbol{f}\in\boldsymbol{\mathcal{F}}} 2 \sum_{t=1}^T \frac{n^2}{N_t} \mathbb{E}(f_t(X_t^1) - \mathbb{E}f_t(X_t^1))^2 \right) \\
&\leq \frac{1}{4b^2} (8b\mathbb{E} \sup_{\boldsymbol{f}\in\boldsymbol{\mathcal{F}}} \sum_{t=1}^T \frac{n}{N_t} \sum_{i=1}^{N_t} \sigma_t^i (f_t(X_t^i) - \mathbb{E}f_t(X_t^i)) + 2nTr) \\
&\leq \frac{1}{4b^2} (16bnT\mathcal{R}(\mathcal{F}) + 2nTr) \\
&\leq \frac{4nT\mathcal{R}(\mathcal{F})}{b} + \frac{nTr}{2b^2},
\end{aligned}
\tag{109}
$$

where the first inequality follows from the standard symmetrization technique and the second inequality follows from the contraction property of the Rademacher complexity.

Plugging (109) into the concentration inequality for $Z$, i.e. (108), we have

$$
\begin{aligned}
Z &\leq \mathbb{E}Z + \frac{2b}{nT}\sqrt{2x\mathbb{E}W} + \frac{bx}{nT} \\
&\leq 2\mathcal{R}(\mathcal{F}) + \frac{2b}{nT}\sqrt{2x\left(\frac{4nT\mathcal{R}(\mathcal{F})}{b} + \frac{nTr}{2b^2}\right)} + \frac{bx}{nT} \\
&= 2\mathcal{R}(\mathcal{F}) + 2\sqrt{\frac{8bx\mathcal{R}(\mathcal{F})}{nT} + \frac{xr}{nT}} + \frac{bx}{nT} \\
&\leq 2\mathcal{R}(\mathcal{F}) + 2\sqrt{\frac{8bx\mathcal{R}(\mathcal{F})}{nT}} + 2\sqrt{\frac{xr}{nT}} + \frac{bx}{nT} \\
&\leq 2(1+\alpha)\mathcal{R}(\mathcal{F}) + 2\sqrt{\frac{xr}{nT}} + \left(1 + \frac{4}{\alpha}\right)\frac{bx}{nT},
\end{aligned}
\tag{110}
$$

where the last inequality follows from the inequality $2\sqrt{ab} \leq \alpha a + \frac{b}{\alpha}$ for any $\alpha > 0, a > 0, b > 0$. $\qquad\square$

### B.3 PROOF OF THEOREM 6

In the following, we assume that for any $\boldsymbol{f} = (f_1, \cdots, f_T) \in \mathcal{F}, 0 \leq f_t \leq b \ (1 \leq t \leq T)$.

Define
$$
U_N(\mathcal{F}) := \sup_{\boldsymbol{f}\in\mathcal{F}} (P\boldsymbol{f} - P_N\boldsymbol{f}).
$$

**Lemma 3.** *For normalized function class $\mathcal{F}_r$,*

$$
\mathcal{F}_r := \left\{ \frac{r}{P\boldsymbol{f}^2 \vee r}\boldsymbol{f} : \ \boldsymbol{f} \in \mathcal{F} \right\}
\tag{111}
$$

*and assume that for some fixed constants $K > 1$ and $r > 0$,*

$$
U_N(\mathcal{F}_r) \leq \frac{r}{bK}
\tag{112}
$$

*Then for any $\boldsymbol{f} \in \mathcal{F}$ the following inequality holds:*

$$
P\boldsymbol{f} \leq \frac{K}{K-1}P_N\boldsymbol{f} + \frac{r}{bK}.
\tag{113}
$$

*Proof.* Let us consider two cases:

1: $P\boldsymbol{f}^2 \leq r$,

2: $P\boldsymbol{f}^2 > r$.

For the first case, $\boldsymbol{f} = \frac{r}{P\boldsymbol{f}^2 \vee r}\boldsymbol{f} \in \mathcal{F}_r$, therefore

$$
P\boldsymbol{f} \leq P_N\boldsymbol{f} + U_N(\mathcal{F}_r) \leq P_N\boldsymbol{f} + \frac{r}{K} \leq \frac{K}{K-1}P_N\boldsymbol{f} + \frac{r}{bK}.
$$

For the second case, $\frac{r}{P\boldsymbol{f}^2 \vee r}\boldsymbol{f} = \frac{r}{P\boldsymbol{f}^2}\boldsymbol{f} \in \mathcal{F}_r$, thus

$$
P\frac{r}{P\boldsymbol{f}^2}\boldsymbol{f} \leq P_N\frac{r}{P\boldsymbol{f}^2}\boldsymbol{f} + U_N(\mathcal{F}_r) \leq P_N\frac{r}{P\boldsymbol{f}^2}\boldsymbol{f} + \frac{r}{bK}.
$$

Basic algebraic transformation yields that

$$
P\boldsymbol{f} \leq P_N\boldsymbol{f} + \frac{P\boldsymbol{f}^2}{bK} \leq P_N\boldsymbol{f} + \frac{P\boldsymbol{f}}{K},
$$

which implies

$$
P\boldsymbol{f} \leq \frac{K}{K-1}P_N\boldsymbol{f} \leq \frac{K}{K-1}P_N\boldsymbol{f} + \frac{r}{bK}.
$$

$\qquad\square$

**Lemma 4.** *Let us consider a sub-root function $\psi(r)$ with fixed point $r^*$ and suppose that $\forall r > r^*$,*

$$\psi(r) \geq b\mathcal{R}(\mathcal{F}_r). \tag{114}$$

*Then for any $K > 1$, we have that, with probability at least $1 - e^{-x}$, for $\forall \boldsymbol{f} \in \mathcal{F}$*

$$P\boldsymbol{f} \leq \frac{K}{K-1}P_N\boldsymbol{f} + \frac{32Kr^*}{b} + \frac{(10b + 8bK)x}{nT}. \tag{115}$$

*Proof.* The aim is to find some $r$ such that $U_N(\mathcal{F}_r) \leq \frac{r}{bK}$, then applying Lemma 3 yields the conclusion.

Note that the variance of functions in $\mathcal{F}_r$ is at most $r$. For any $\boldsymbol{f} \in \mathcal{F}_r$, we consider two cases:

1: $P\boldsymbol{f}^2 \leq r$,

2: $P\boldsymbol{f}^2 > r$.

For the first case, $\boldsymbol{f} = \frac{r}{P\boldsymbol{f}^2 \vee r}\boldsymbol{f} \in \mathcal{F}_r$, thus $Var\left(\frac{r}{P\boldsymbol{f}^2 \vee r}\boldsymbol{f}\right) = Var(\boldsymbol{f}) \leq P\boldsymbol{f}^2 \leq r$.

For the second case,

$$Var\left(\frac{r}{P\boldsymbol{f}^2 \vee r}\boldsymbol{f}\right) = Var\left(\frac{r}{P\boldsymbol{f}^2}\boldsymbol{f}\right) \leq P\left(\frac{r}{P\boldsymbol{f}^2}\boldsymbol{f}\right)^2 = \frac{r^2}{P\boldsymbol{f}^2} < r.$$

Then applying Theorem 5 for $U_N(\mathcal{F}_r)$ with $\alpha = 1$, we have that with probability at least $1 - e^{-x}$,

$$U_N(\mathcal{F}_r) \leq 4\mathcal{R}(\mathcal{F}_r) + 2\sqrt{\frac{xr}{nT}} + \frac{5bx}{nT}$$

$$\leq 4\frac{\psi(r)}{b} + 2\sqrt{\frac{xr}{nT}} + \frac{5bx}{nT}$$

$$\leq 4\frac{\sqrt{rr^*}}{b} + 2\sqrt{\frac{xr}{nT}} + \frac{5bx}{nT}$$

$$:= A\sqrt{r} + B,$$

where the third inequality follows from the property of the sub-root function, i.e., $\psi(r)/\sqrt{r} \leq \psi(r^*)/\sqrt{r^*} = \sqrt{r^*}$ for any $r > r^*$ and $A = \frac{4\sqrt{r^*}}{b} + 2\sqrt{\frac{x}{nT}}$, $B = \frac{5bx}{nT}$.

Solving the equation

$$A\sqrt{r} + B = \frac{r}{bK}$$

yields that

$$\sqrt{r} = \frac{bKA + \sqrt{b^2K^2A^2 + 4bKB}}{2}.$$

Thus

$$r \geq \frac{b^2K^2A^2}{2} > r^*$$

and

$$r \leq b^2K^2A^2 + 2bKB.$$

Therefore by Lemma 3, we have

$$P\boldsymbol{f} \leq \frac{K}{K-1}P_N\boldsymbol{f} + \frac{r}{bK}$$

$$\leq \frac{K}{K-1}P_N\boldsymbol{f} + bKA^2 + 2B$$

$$\leq \frac{K}{K-1}P_N\boldsymbol{f} + 2bK\left(\frac{16r^*}{b^2} + \frac{4x}{nT}\right) + \frac{10bx}{nT} \tag{116}$$

$$= \frac{K}{K-1}P_N\boldsymbol{f} + \frac{32Kr^*}{b} + \frac{(10b + 8bK)x}{nT}.$$

$\square$

There remain some problems regarding the selection of the sub-root function $\psi$ and the computation of its fixed point. Just as in the single-task scenario, we can take $\psi$ as the local Rademacher averages of the star-hull of $\mathcal{F}$ around 0.

Specifically, let

$$\psi(r) := 16b\mathbb{E}\mathcal{R}_N\{\boldsymbol{f} : \boldsymbol{f} \in star(\mathcal{F}, 0), P\boldsymbol{f}^2 \leq r\} + \frac{14b^2 \log(nT)}{nT}, \tag{117}$$

where $star(\mathcal{F}, 0) := \{\alpha\boldsymbol{f} : \boldsymbol{f} \in \mathcal{F}, \alpha \in [0, 1]\}$.

Note that the normalized function class $\mathcal{F}_r$ defined in the Lemma 3 is a subset of the function class $\{\boldsymbol{f} : \boldsymbol{f} \in star(\mathcal{F}, 0), P\boldsymbol{f}^2 \leq r\}$, thus $\psi(r) \geq b\mathcal{R}(\mathcal{F}_r)$.

For the first term in the definition of $\psi(r)$, i.e. (117), with the following lemma, we can translate the ball in $L^2(P)$ into the ball in $L^2(P_N)$, so that Dudley's theorem can be applied.

**Lemma 5.** *Let $\mathcal{G}$ be a class of vector-valued functions that map $\mathcal{X}$ into $[-b, b]^T$ with $b > 0$. For every $x > 0$ and $r$ satisfy*

$$r \geq 16b\mathbb{E}\mathcal{R}_N\{\boldsymbol{g} : \boldsymbol{g} \in \mathcal{G}, P\boldsymbol{g}^2 \leq r\} + \frac{14b^2 x}{nT}, \tag{118}$$

*then with probability at least $1 - e^{-x}$*

$$\{\boldsymbol{g} \in \mathcal{G} : P\boldsymbol{g}^2 \leq r\} \subset \{\boldsymbol{g} \in \mathcal{G} : P_N\boldsymbol{g}^2 \leq 2r\}. \tag{119}$$

*Proof.* Define $\mathcal{G}_r := \{\boldsymbol{g}^2 : \boldsymbol{g} \in \mathcal{G}, P\boldsymbol{g}^2 \leq r\}$.

Note that $\|\boldsymbol{g}^2\|_\infty \leq b^2, Var(\boldsymbol{g}^2) \leq P\boldsymbol{g}^4 \leq b^2 P\boldsymbol{g}^2 \leq b^2 r$. Then applying the Theorem 5 for $\mathcal{G}_r$ with $\alpha = 1$ yields that with probability at least $1 - e^{-x}$, for any $\boldsymbol{g} \in \mathcal{G}$ such that $\boldsymbol{g}^2 \in \mathcal{G}_r$,

$$P_N\boldsymbol{g}^2 \leq P\boldsymbol{g}^2 + 4\mathbb{E}\mathcal{R}_N\{\boldsymbol{g}^2 : \boldsymbol{g} \in \mathcal{G}, P\boldsymbol{g}^2 \leq r\} + 2\sqrt{\frac{b^2 xr}{nT}} + \frac{5b^2 x}{nT}$$

$$\leq r + 8b\mathbb{E}\mathcal{R}_N\{\boldsymbol{g} : \boldsymbol{g} \in \mathcal{G}, P\boldsymbol{g}^2 \leq r\} + \frac{r}{2} + \frac{7b^2 x}{nT}$$

$$\leq 2r,$$

where the second inequality follows from the contraction property of the Rademacher complexity and the mean inequality. $\square$

**Remark 7.** *Although the contraction property used in the proof of Lemma 5 is slightly different from the standard form (see Lemma 5.7 in Mohri et al. (2018)), it is just an adaptation of the standard one.*

*Specifically, let $\Phi_i$ be $l_i$-Lipschitz functions from $\mathbb{R}$ to $\mathbb{R}$ for $i = 1, \cdots, m$ and $\sigma_1, \cdots, \sigma_m$ be Rademacher random variables. Then for any set $A \subset \mathbb{R}^m$, the following inequality holds.*

$$\mathbb{E}_\sigma \sup_{a \in A} \sum_{i=1}^m \sigma_i \Phi_i(a_i) \leq \mathbb{E}_\sigma \sup_{a \in A} \sum_{i=1}^m \sigma_i l_i a_i.$$

*For completeness, we give a brief proof.*

*By the Fubini's theorem, we have*

$$\mathbb{E}_\sigma \sup_{a \in A} \sum_{i=1}^m \sigma_i \Phi_i(a_i) = \mathbb{E}_{\sigma_1, \cdots, \sigma_{m-1}} \mathbb{E}_{\sigma_m}[\sup_{a \in A} u_{m-1}(a) + \sigma_m \Phi_m(a_m)],$$

*where $u_{m-1}(a) = \sum\limits_{i=1}^{m-1} \sigma_i \Phi_i(a_i)$.*

*From the proof of Lemma 5.7 in Mohri et al. (2018), we know*

$$\mathbb{E}_{\sigma_m}[\sup_{a \in A} u_{m-1}(a) + \sigma_m \Phi_m(a_m)] \leq \mathbb{E}_{\sigma_m}[\sup_{a \in A} u_{m-1}(a) + \sigma_m l_m a_m].$$

*Proceeding in the same way for all other $\sigma_i(i \neq m)$ leads to the conclusion. In fact, we have used the conclusion with $\Phi_{t,i}(x) = \frac{x^2}{N_t}$ in the proof of Lemma 5.*

With Lemma 5, we can bound $r^*$ as follows.

**Lemma 6.**

$$r^* \leq 16b\mathbb{E}\mathcal{R}_N\{\boldsymbol{f} : \boldsymbol{f} \in star(\boldsymbol{\mathcal{F}}, 0), P_N\boldsymbol{f}^2 \leq 2r^*\} + \frac{16b^2 + 14b^2\log(nT)}{nT}. \tag{120}$$

*Proof.* From Lemma 5 and the fact that

$$r^* = \psi(r^*) = 16b\mathbb{E}\mathcal{R}_N\{\boldsymbol{f} : \boldsymbol{f} \in star(\boldsymbol{\mathcal{F}}, 0), P\boldsymbol{f}^2 \leq r^*\} + \frac{14b^2\log(nT)}{nT},$$

we can deduce that with probability at least $1 - \frac{1}{nT}$,

$$\{\boldsymbol{f} : \boldsymbol{f} \in star(\boldsymbol{\mathcal{F}}, 0), P\boldsymbol{f}^2 \leq r^*\} \subset \{\boldsymbol{f} : \boldsymbol{f} \in star(\boldsymbol{\mathcal{F}}, 0), P_N\boldsymbol{f}^2 \leq 2r^*\}.$$

Therefore,

$$r^* \leq 16b\left[\mathbb{E}\mathcal{R}_N\{\boldsymbol{f} : \boldsymbol{f} \in star(\boldsymbol{\mathcal{F}}, 0), P_N\boldsymbol{f}^2 \leq 2r^*\} + \frac{b}{nT}\right] + \frac{14b^2\log(nT)}{nT}$$

$$= 16b\mathbb{E}\mathcal{R}_N\{\boldsymbol{f} : \boldsymbol{f} \in star(\boldsymbol{\mathcal{F}}, 0), P_N\boldsymbol{f}^2 \leq 2r^*\} + \frac{16b^2 + 14b^2\log(nT)}{nT}.$$

$\square$

Now, we are ready to use the Dudley's theorem to bound the first term in the right.

Specifically, define $\boldsymbol{\mathcal{F}}_{s,r} := \{\boldsymbol{f} : \boldsymbol{f} \in star(\boldsymbol{\mathcal{F}}, 0), P_N\boldsymbol{f}^2 \leq 2r\}$, with the samples $(X_t^i)_{(t,i)=(1,1)}^{(T,N_t)}$ fixed, define a random process $(X_{\boldsymbol{f}})_{\boldsymbol{f} \in \boldsymbol{\mathcal{F}}_{s,r}}$ as

$$X_{\boldsymbol{f}} := \frac{1}{T}\sum_{t=1}^{T}\frac{1}{N_t}\sum_{i=1}^{N_t}\sigma_t^i f_t(X_t^i) \; for \; \boldsymbol{f} = (f_1, \cdots, f_T) \in \boldsymbol{\mathcal{F}}_{s,r}. \tag{121}$$

From the fact that $\sigma_t^i$ is sub-gaussian, we can deduce that for any $\lambda \in \mathbb{R}$ and $\boldsymbol{f}' = (f_1', \cdots, f_T') \in \boldsymbol{\mathcal{F}}_{s,r}$

$$\mathbb{E}e^{\lambda(X_{\boldsymbol{f}} - X_{\boldsymbol{f}'})} = \mathbb{E}e^{\frac{\lambda}{T}\sum\limits_{t=1}^{T}\frac{1}{N_t}\sum\limits_{i=1}^{N_t}\sigma_t^i(f_t(X_t^i) - f_t'(X_t^i))}$$

$$\leq e^{\frac{\lambda^2}{2T^2}\sum\limits_{t=1}^{T}\frac{1}{N_t^2}\sum\limits_{i=1}^{N_t}(f_t(X_t^i) - f_t'(X_t^i))^2}$$

$$\leq e^{\frac{\lambda^2}{2}K^2 d^2(\boldsymbol{f},\boldsymbol{f}')},$$

where $K = \frac{1}{\sqrt{nT}}$ and

$$d(\boldsymbol{f}, \boldsymbol{f}') := \sqrt{\frac{1}{T}\sum_{t=1}^{T}\frac{1}{N_t}\sum_{i=1}^{N_t}(f_t(X_t^i) - f_t'(X_t^i))^2}. \tag{122}$$

It implies that $\|X_{\boldsymbol{f}} - X_{\boldsymbol{f}'}\|_{\psi_2} \leq CKd(\boldsymbol{f}, \boldsymbol{f}')$ with a universal constant $C$.

Then using Dudley's theorem yields that

$$\mathbb{E}\sup_{\boldsymbol{f} \in \boldsymbol{\mathcal{F}}_{s,r}} X_{\boldsymbol{f}} \leq CK\int_0^{diam(\boldsymbol{\mathcal{F}}_{s,r})}\sqrt{\log\mathcal{N}(\boldsymbol{\mathcal{F}}_{s,r}, d, \epsilon)}d\epsilon \leq CK\int_0^{2\sqrt{r}}\sqrt{\log\mathcal{N}(\boldsymbol{\mathcal{F}}_{s,r}, d, \epsilon)}d\epsilon, \tag{123}$$

where $diam(\boldsymbol{\mathcal{F}}_{s,r}) := \sup_{\boldsymbol{f},\boldsymbol{f}' \in \boldsymbol{\mathcal{F}}_{s,r}} d(\boldsymbol{f}, \boldsymbol{f}')$.

*Proof of Theorem 6:* In the following, we assume that $\mathcal{F}$ is a parameterized hypothesis function class to be determined. When considering the framework of PINNs for the linear second order elliptic equation as MTL, the function class in MTL associated with $\mathcal{F}$ is defined as

$$\boldsymbol{\mathcal{F}} := \{\boldsymbol{u} = \left(|\Omega|(Lu(x) - f(x))^2, |\partial\Omega|(u(y) - g(y))^2\right) : u \in \mathcal{F}\}. \tag{124}$$

Note that here we use notation $u$ to represent a function in $\mathcal{F}$ and $\boldsymbol{u}$ to denote the corresponding vector-valued function associated with $u$.

Then the empirical loss can be written as

$$
\mathcal{L}_N(u) = \frac{|\Omega|}{N_1} \sum_{k=1}^{N_1} \left( -\sum_{i,j=1}^{d} a_{ij}(X_k)\partial_{ij}u(X_k) + \sum_{i=1}^{d} b_i(X_k)\partial_i u(X_k) + c(X_k)u(X_k) - f(X_k) \right)^2
$$
$$
+ \frac{|\partial\Omega|}{N_2} \sum_{k=1}^{N_2} (u(Y_k) - g(Y_k))^2
$$
$$
= 2P_N \boldsymbol{u},
$$

where $N = (N_1, N_2)$ and $n = \min(N_1, N_2)$.

The aim is to seek $u_N \in \mathcal{F}$ which minimizes $\mathcal{L}_N$. It is equivalent to seek $\boldsymbol{u}_N \in \mathcal{F}$ which minimizes $P_N \boldsymbol{u}$ i.e.,

$$
\boldsymbol{u}_N \in \arg\min_{\boldsymbol{u} \in \mathcal{F}} P_N \boldsymbol{u}. \tag{125}
$$

Assume that $u^*$ is the solution of the linear second order elliptic PDE and there is a constant $M$ such that $|a_{ij}|, |b_i|, |c|, |g|, |u^*|, |\partial_i u^*|, |\partial_{ij} u^*| \leq M$ and $|u|, |\partial_i u|, |\partial_{ij} u| \leq M$ for any $u \in \mathcal{F}$, $1 \leq i, j \leq d$.

Then $\sup_{u \in \mathcal{F}} \max(|\Omega|(Lu - f)^2, |\partial\Omega|(u - g)^2) \leq c(|\Omega|d^2 M^4 + |\partial\Omega|M^2) := b$ with a universal constant $c$.

Therefore, with probability at least $1 - e^{-t}$

$$
P_N \boldsymbol{u}_N \leq P_N \boldsymbol{u}_{\mathcal{F}} \leq P\boldsymbol{u}_{\mathcal{F}} + 2\sqrt{\frac{tVar(\boldsymbol{u}_{\mathcal{F}})}{2n}} + \frac{2bt}{2n}
$$
$$
\leq P\boldsymbol{u}_{\mathcal{F}} + 2\sqrt{\frac{tbP\boldsymbol{u}_{\mathcal{F}}}{2n}} + \frac{bt}{n} \tag{126}
$$
$$
\leq \frac{3}{2}P\boldsymbol{u}_{\mathcal{F}} + \frac{2bt}{n},
$$

where $\boldsymbol{u}_{\mathcal{F}} = \left( |\Omega|(Lu_{\mathcal{F}} - f)^2, |\partial\Omega|(u_{\mathcal{F}} - g)^2 \right)$, $u_{\mathcal{F}} \in \arg\min_{u \in \mathcal{F}} \|u - u^*\|_{H^2(\Omega)}^2$ and the second inequality follows from Theorem 5 by taking $\mathcal{F} = \{\boldsymbol{u}_{\mathcal{F}}\}$ and $\alpha = 4, T = 2$, which can be seen as a vector version of the Bernstein inequality. Here, we define the approximation error as $\epsilon_{app} := \|u_{\mathcal{F}} - u^*\|_{H^2(\Omega)}$.

Then applying Lemma 4 with $K = 2$ yields that with probability at least $1 - 2e^{-t}$

$$
P\boldsymbol{u}_N \leq 2P_N \boldsymbol{u}_N + \frac{64r^*}{b} + \frac{13bt}{n}
$$
$$
\leq 3P\boldsymbol{u}_{\mathcal{F}} + \frac{64r^*}{b} + \frac{17bt}{n}, \tag{127}
$$

which implies that

$$
\mathcal{L}(u_N) = 2P_N \boldsymbol{u}_N \leq 3\mathcal{L}(u_{\mathcal{F}}) + \frac{128r^*}{b} + \frac{34bt}{n}. \tag{128}
$$

Note that $\mathcal{L}(u_{\mathcal{F}})$ can be bounded by the approximation error, since for any $u \in H^2(\Omega)$

$$\mathcal{L}(u) = \int_{\Omega} (Lu - f)^2 dx + \int_{\partial\Omega} (u - g)^2 dy$$

$$= \int_{\Omega} \left( -\sum_{i,j=1}^{d} a_{ij}\partial_{ij}u + \sum_{i=1}^{d} b_i\partial_i u + cu - f \right)^2 dx + \int_{\partial\Omega} (u - g)^2 dy$$

$$= \int_{\Omega} \left( -\sum_{i,j=1}^{d} a_{ij}\partial_{ij}(u - u^*) + \sum_{i=1}^{d} b_i\partial_i(u - u^*) + c(u - u^*) \right)^2 dx + \int_{\partial\Omega} (u - u^*)^2 dy$$

$$\leq 3 \int_{\Omega} \left( -\sum_{i,j=1}^{d} a_{ij}\partial_{ij}(u - u^*) \right)^2 + \left( \sum_{i=1}^{d} b_i\partial_i(u - u^*) \right)^2 + (c(u - u^*))^2 dx + \int_{\partial\Omega} (u - u^*)^2 dy$$

$$\leq 3d^2 M^2 \|u - u^*\|_{H^2(\Omega)}^2 + C(Tr, \Omega)^2 \|u - u^*\|_{H^1(\Omega)}^2$$

$$\leq (3d^2 M^2 + C(Tr, \Omega)^2) \|u - u^*\|_{H^2(\Omega)}^2, \tag{129}$$

where in the last inequality, we use the boundedness of $a_{ij}, b_i, c$ and the Sobolev trace theorem with the constant $C(Tr, \Omega)$ that depends only on the domain $\Omega$.

Thus,

$$\mathcal{L}(u_{\mathcal{F}}) \leq (3d^2 M^2 + C(Tr, \Omega)^2)\epsilon_{app}^2 \tag{130}$$

and with probability at least $1 - 2e^{-t}$

$$\mathcal{L}(u_N) = 2P_N \boldsymbol{u}_N \leq 3(3d^2 M^2 + C(Tr, \Omega)^2)\epsilon_{app}^2 + \frac{128r^*}{b} + \frac{34bt}{n}. \tag{131}$$

It remains only to bound the fixed point $r^*$. With Lemma 6, it suffices to bound the covering number of $\mathcal{F}$ under $d$, which is done in the Lemma 16. Thus, we have the following results.

(1) For the two-layer neural networks, we know

$$\log \mathcal{N}(\mathcal{F}, d, \epsilon) \leq cmd \log \left( \frac{b}{\epsilon} \right), \tag{132}$$

where $c$ is a universal constant.

Therefore

$$r^* \leq cb\sqrt{\frac{md}{n}} \int_0^{2\sqrt{r^*}} \sqrt{\log\left(\frac{b}{\epsilon}\right)} d\epsilon + \frac{cb^2 \log n}{n}$$

$$= cb^2 \sqrt{\frac{md}{n}} \int_0^{2\sqrt{\frac{r^*}{b^2}}} \sqrt{\log\left(\frac{1}{\epsilon}\right)} d\epsilon + \frac{cb^2 \log n}{n}$$

$$\leq cb\sqrt{\frac{mdr^*}{n}} \sqrt{\log\left(\frac{2b}{\sqrt{r^*}}\right)} + c\frac{cb^2 \log n}{n} \tag{133}$$

$$\leq cb\sqrt{\frac{mdr^*}{n}} \sqrt{\log n} + \frac{cb^2 \log n}{n},$$

where second inequality follows from Lemma 13.

It implies that

$$r^* \leq \frac{cb^2 md \log n}{n}. \tag{134}$$

(2) For the deep neural networks, we know

$$\log \mathcal{N}(\mathcal{F}, d, \epsilon) \leq CK^d \log\left(\frac{K}{\epsilon}\right), \tag{135}$$

where $C$ is a constant independent of $K$.

Similar to that in (1), we have

$$r^* \leq \frac{CK^d(\log K + \log n)}{n} \tag{136}$$

with a constant $C$ independent of $K, N, n$. $\qquad\square$

## C   AUXILIARY LEMMAS

**Lemma 7** (Bernstein inequality). *Let $X_i, 1 \leq i \leq n$ be i.i.d. centred random variables a.s. bounded by $b < \infty$ in absolute value. Set $\sigma^2 = \mathbb{E}X_1^2$ and $S_n = \frac{1}{n}\sum_{i=1}^{n} X_i$. Then, for all $t > 0$,*

$$P\left(S_n \geq \sqrt{\frac{2\sigma^2 t}{n}} + \frac{bt}{3n}\right) \leq e^{-t}.$$

**Lemma 8** (Hoeffding inequality). *Let $X_i, 1 \leq i \leq n$ be i.i.d. centred random variables a.s. bounded by $b < \infty$ in absolute value. Set $S_n = \frac{1}{n}\sum_{i=1}^{n} X_i$, then for all $t > 0$,*

$$P\left(|S_n| \geq b\sqrt{\frac{2t}{n}}\right) \leq 2e^{-t}.$$

**Lemma 9** (Bounded difference inequality). *Let $X_1, \cdots, X_m \in \mathcal{X}^m$ be a set of $m \geq 1$ independent random variables and assume that there exists $c_1, \cdots, c_m$ such that $f : \mathcal{X}^m \to \mathbb{R}$ satisfies the following conditions:*

$$|f(x_1, \cdots, x_i, \cdots, x_m) - f(x_1, \cdots, x_i', \cdots, x_m)| \leq c_i,$$

*for all $i \in [m]$ and any points $x_1, \cdots, x_m, x_i' \in \mathcal{X}$. Let $f(S)$ denote $f(X_1, \cdots, X_m)$, then, for all $\epsilon > 0$, the following inequalities hold:*

$$P(f(S) - \mathbb{E}(f(S)) \geq \epsilon) \leq \exp\left(\frac{-2\epsilon^2}{\sum_{i=1}^{m} c_i^2}\right),$$

$$P(f(S) - \mathbb{E}(f(S)) \leq -\epsilon) \leq \exp\left(\frac{-2\epsilon^2}{\sum_{i=1}^{m} c_i^2}\right)$$

**Lemma 10** (Theorem 1 in Makovoz (1996)). *Let $\Phi := \{\phi_1, \phi_2, \cdots\}$ be an arbitrary bounded sequence of elements of the Hilbert space $H$. For every $f \in H$ of the form*

$$f = \sum_i c_i \phi_i, \quad \sum_i |c_i| < \infty,$$

*and for every natural number $n$, there is a $g = \sum_i a_i \phi_i$ with at most $n$ non-zero coefficients $a_i$ and with $\sum_i |a_i| \leq \sum_i |c_i|$, for which*

$$\|f - g\| \leq 2\epsilon_n(\Phi)n^{-1/2}\sum_i |c_i|.$$

The definition of metric entropy $\epsilon_n$ is given in Proposition 8.

**Lemma 11** (Covering number of $\partial B_1^d(1)$ in the $L^1$ norm). *For any $\epsilon > 0$,*

$$\mathcal{N}(\partial B_1^d(1), |\cdot|_1, \epsilon) \leq 2\left(\frac{12}{\epsilon}\right)^{d-1}.$$

*Proof.* By the symmetry of $\partial B_1^d(1)$, it suffices to consider the set

$$S := \{(x_1, \cdots, x_d) \in \partial B_1^d(1), x_i \geq 0, 1 \leq i \leq d\}, \tag{137}$$

as $\mathcal{N}(\partial B_1^d(1), |\cdot|_1, \epsilon) \le 2^d \mathcal{N}(S, |\cdot|_1, \epsilon)$.

Note that for $(x_1, \cdots, x_d) \in \partial B_1^d(1)$, $x_d$ is determined by $x_1, \cdots, x_{d-1}$. Thus the problem of estimating the covering number of $\partial B_1^d(1)$ can be reduced to estimating the covering number of

$$S_1 := \{(x_1, \cdots, x_{d-1}) : x_1 + \cdots + x_{d-1} \le 1, x_i \ge 0, 1 \le i \le d-1\}, \tag{138}$$

which is a subset of $B_1^{d-1}(1)$.

By Lemma 5.7 in Wainwright (2019), we know $\mathcal{N}(B_1^{d-1}(1), |\cdot|_1, \epsilon) \le (\frac{2}{\epsilon}+1)^{d-1} \le (\frac{3}{\epsilon})^{d-1}$. Thus, there exists a $\frac{\epsilon}{2}$- cover of $B_1^{d-1}(1)$ with cardinality $(\frac{6}{\epsilon})^{d-1}$ which we denote by $C$. Although $C$ is also a $\frac{\epsilon}{2}$- cover of $S_1$, the elements in $C$ may not belong to $S_1$. To fix this issue, we can transform $C$ to a subset of $S_1$ and the transformation doesn't change the property that $C$ is a $\frac{\epsilon}{2}$- cover of $S_1$. Specifically, for $(y_1, \cdots, y_{d-1}) \in C$, we do the transformation as follows

$$(y_1, \cdots, y_{d-1}) \to (y_1 I_{\{y_1 \ge 0\}}, \cdots, y_{d-1} I_{\{y_{d-1} \ge 0\}}).$$

Note that
$$y_1 I_{\{y_1 \ge 0\}} + \cdots + y_{d-1} I_{\{y_{d-1} \ge 0\}} \le |y_1| + \cdots + |y_{d-1}| \le 1, \tag{139}$$

and for any $(x_1, \cdots, x_{d-1}) \in S_1$

$$|x_1 - y_1 I_{\{y_1 \ge 0\}}| + \cdots |x_{d-1} - y_{d-1} I_{\{y_{d-1} \ge 0\}}| \le |x_1 - y_1| + \cdots + |x_{d-1} - y_{d-1}|, \tag{140}$$

which imply that after transformation, it is a subset of $S_1$ and also a $\frac{\epsilon}{2}$- cover of $S_1$. For simplicity, we still denote it by $C$.

Now we are ready to give a $\epsilon$-cover of $S$ via extending $C$ to a subset of $\partial B_1^d(1)$. Define $C_e := \{(y_1, \cdots, y_d) : (y_1, \cdots, y_{d-1}) \in C, y_d = 1 - (y_1 + \cdots + y_{d-1})\}$.

Thus for any $(x_1, \cdots, x_d) \in S$, since $(x_1, \cdots, x_{d-1}) \in S_1$ and $C$ is a $\frac{\epsilon}{2}$-cover of $S_1$, there exists a element of $C$, we denote it by $(z_1, \cdots, z_{d-1})$, such that

$$|x_1 - z_1| + \cdots + |x_{d-1} - z_{d-1}| \le \frac{\epsilon}{2}. \tag{141}$$

Note that for $z_d = 1 - (z_1 + \cdots + z_{d-1})$, $(z_1, \cdots, z_d) \in C_e$ and

$$\begin{aligned}
&|x_1 - z_1| + \cdots + |x_{d-1} - z_{d-1}| + |x_d - z_d| \\
&= |x_1 - z_1| + \cdots + |x_{d-1} - z_{d-1}| + |x_1 - z_1 + \cdots + x_{d-1} - z_{d-1}| \\
&\le 2(|x_1 - z_1| + \cdots + |x_{d-1} - z_{d-1}|) \\
&\le \epsilon,
\end{aligned}$$

which implies that $C_e$ is a $\epsilon$-cover of $S$.

Recall that $|C_e| = |C| = (\frac{6}{\epsilon})^{d-1}$, then $\mathcal{N}(\partial B_1(1), |\cdot|_1, \epsilon) \le 2^d (\frac{6}{\epsilon})^{d-1} = 2 (\frac{12}{\epsilon})^{d-1}$.

Note that in this lemma, our goal is not to investigate the optimal upper bound, but to give an upper bound with explicit dependence on the dimension. $\qquad\square$

**Lemma 12** (Equivalence between metric entropy and covering number). *Let $(T, d)$ be a metric space and there is a continuous and strictly increasing function $f : \mathbb{R}_+ \to \mathbb{R}_+$ such that for any $\epsilon > 0$,*

$$\mathcal{N}(T, d, \epsilon) \le f(\epsilon),$$

*Then for any $\epsilon > 0$,*

$$\epsilon_n(T) \le f^{-1}(n),$$

*where $f^{-1}$ represents the inverse of $f$.*

*Proof.* It's obvious, since $\mathcal{N}(T, d, f^{-1}(n)) \le f(f^{-1}(n)) = n$. $\qquad\square$

**Lemma 13.** *For any $0 < x \le 1$, we have*

$$\int_0^x \sqrt{\log \frac{1}{\epsilon}} d\epsilon \le 2x \sqrt{\log \frac{4}{x}}.$$

*Proof.* For $0 < x \le 1$, let $f(x) = \sqrt{x \log \frac{1}{x}}$, $g(x) = \sqrt{x}$, $h(x) = x \log \frac{1}{x}$, then $f(x) = g(h(x))$. Note that $g$ is increasing, concave and $h$ is concave, thus

$$
\begin{aligned}
f(\lambda x + (1 - \lambda)y) &= g(h(\lambda x + (1 - \lambda)y)) \\
&\ge g(\lambda h(x) + (1 - \lambda)h(y)) \\
&\ge \lambda g(h(x)) + (1 - \lambda)g(h(y)) \\
&= \lambda f(x) + (1 - \lambda)f(y),
\end{aligned}
$$

which means $f$ is concave in $[0, 1]$.

Let $\epsilon = y^{\frac{3}{2}}$, then

$$
\begin{aligned}
\int_0^x \sqrt{\log \frac{1}{\epsilon}} d\epsilon &= (\frac{3}{2})^{\frac{3}{2}} \int_0^{x^{\frac{2}{3}}} \sqrt{y \log \frac{1}{y}} dy \\
&\le (\frac{3}{2})^{\frac{3}{2}} x^{\frac{2}{3}} \sqrt{\frac{x^{\frac{2}{3}}}{2} \log \frac{2}{x^{\frac{2}{3}}}} \\
&= (\frac{3}{2})^{\frac{3}{2}} x \sqrt{\frac{1}{3} \log \frac{2^{\frac{3}{2}}}{x}} \\
&\le 2x \sqrt{\log \frac{4}{x}},
\end{aligned}
$$

where the first inequality follows from Jensen's inequality. $\qquad\square$

**Lemma 14** (The remaining part of the proof of Theorem 3). *For the function class $\mathcal{F}$ and*

$$
\mathcal{G} := \{(|\nabla u(x)|^2 - 2f(x)u(x)) - (|\nabla u^*(x)|^2 - 2f(x)u^*(x)) : u \in \mathcal{F}\},
$$

*we assume that for any $\epsilon > 0$,*

$$
\mathcal{N}(\mathcal{F}, \|\cdot\|_{L^2(P_n)}, \epsilon) \le \left(\frac{b}{\epsilon}\right)^a \ a.s. \ and \ \mathcal{N}(\mathcal{G}, \|\cdot\|_{L^2(P_n)}, \epsilon) \le \left(\frac{\beta}{\epsilon}\right)^\alpha \ a.s.
$$

*for some positive constants $a, b, \alpha, \beta$ with $b > \sup_{f \in \mathcal{F}} |f|$, $\beta > \sup_{g \in \mathcal{G}} |g|$.*

*Then we have that with probability at least $1 - e^{-t}$*

$$
\begin{aligned}
\sup_{u \in \mathcal{F}_\delta} & (\mathcal{E}(u) - \mathcal{E}(u^*)) - (\mathcal{E}_n(u) - \mathcal{E}_n(u^*)) \\
&\le C(\frac{\alpha M^2 \log(2\beta\sqrt{n})}{n} + \sqrt{\frac{M^2\delta\alpha \log(2\beta\sqrt{n})}{n}} + \sqrt{\frac{M^2\delta t}{n}} \\
&\quad + \frac{M^2 t}{n} + \sqrt{\frac{aM^2\delta}{n}} \log \frac{4b}{M}),
\end{aligned} \qquad (142)
$$

*where*

$$
\mathcal{F}_\delta := \{u \in \mathcal{F} : \|u - u^*\|_{H^1(\Omega)}^2 \le \delta\}
$$

*and $C$ is a universal constant.*

*Proof.* As before, rearranging $\sup_{u \in \mathcal{F}_\delta} (\mathcal{E}(u) - \mathcal{E}(u^*)) - (\mathcal{E}_n(u) - \mathcal{E}_n(u^*))$ yields that

$$
\begin{aligned}
&\sup_{u \in \mathcal{F}_\delta} (\mathcal{E}(u) - \mathcal{E}(u^*)) - (\mathcal{E}_n(u) - \mathcal{E}_n(u^*)) \\
&= \sup_{u \in \mathcal{F}(\delta)} \Bigg[ \int_\Omega \big[ (|\nabla u(x)|^2 - 2f(x)u(x)) - (|\nabla u^*(x)|^2 - 2f(x)u^*(x)) \big] dx \\
&\qquad - \frac{1}{n} \sum_{i=1}^n \big[ (|\nabla u(X_i)|^2 - 2f(X_i)u(X_i)) - (|\nabla u^*(X_i)|^2 - 2f(X_i)u^*(X_i)) \big] \\
&\qquad + \left( \int_\Omega u(x)dx \right)^2 - \left( \frac{1}{n} \sum_{i=1}^n u(X_i) \right)^2 + \left( \frac{1}{n} \sum_{i=1}^n u^*(X_i) \right)^2 \Bigg] \\
&\leq \sup_{g \in \mathcal{G}(\delta)} (P - P_n)g + \sup_{u \in \mathcal{F}(\delta)} \left[ \left( \int_\Omega u(x)dx \right)^2 - \left( \frac{1}{n} \sum_{i=1}^n u(X_i) \right)^2 \right] + \left( \frac{1}{n} \sum_{i=1}^n u^*(X_i) \right)^2 \\
&:= \psi_n^{(1)}(\delta) + \psi_n^{(2)}(\delta) + \psi_n^{(3)}(\delta),
\end{aligned}
\tag{143}
$$

where

$$
\mathcal{G}(\delta) := \{ (|\nabla u(x)|^2 - 2f(x)u(x)) - (|\nabla u^*(x)|^2 - 2f(x)u^*(x)) : u \in \mathcal{F}, \|u - u^*\|_{H^1(\Omega)}^2 \leq \delta \}.
$$

Applying the Hoeffding inequality for $\psi_n^{(3)}(\delta)$, we can obtain that with probability at least $1 - e^{-t}$

$$
\psi_n^{(3)}(\delta) = \left( \frac{1}{n} \sum_{i=1}^n u^*(X_i) \right)^2 \leq \frac{2M^2 t}{n}.
\tag{144}
$$

For $\psi_n^{(2)}(\delta)$, we can deduce that

$$
\begin{aligned}
\psi_n^{(2)}(\delta) &= \sup_{u \in \mathcal{F}(\delta)} \left[ \left( \int_\Omega u(x)dx \right)^2 - \left( \frac{1}{n} \sum_{i=1}^n u(X_i) \right)^2 \right] \\
&= \sup_{u \in \mathcal{F}(\delta)} [(Pu)^2 - (P_n u)^2] \\
&= \sup_{u \in \mathcal{F}(\delta)} [(Pu)^2 - ((P_n u - Pu) + Pu)^2] \\
&= \sup_{u \in \mathcal{F}(\delta)} [2(Pu)((P - P_n)u) - (P_n u - Pu)^2] \\
&\leq \sqrt{\delta} \sup_{u \in \mathcal{F}(\delta)} |(P - P_n)u|,
\end{aligned}
\tag{145}
$$

where the last inequality follows from the fact that for any $u \in \mathcal{F}(\delta)$,

$$
|Pu| = \left| \int_\Omega u \, dx \right| = \left| \int_\Omega (u - u^*) dx \right| \leq \left( \int_\Omega (u - u^*)^2 dx \right)^{\frac{1}{2}} \leq \sqrt{\delta}.
$$

Therefore, to bound $\psi_n^{(2)}(\delta)$, it suffices to bound the empirical process $\sup_{u \in \mathcal{F}(\delta)} |(P - P_n)u|$. By applying the bounded difference inequality and the symmetrization technique, we can deduce that with probability at least $1 - e^{-t}$

$$
\begin{aligned}
\sup_{u \in \mathcal{F}(\delta)} |(P - P_n)u| &\leq \mathbb{E} \sup_{u \in \mathcal{F}(\delta)} |(P - P_n)u| + M\sqrt{\frac{2t}{n}} \\
&\leq 2\mathbb{E} \sup_{u \in \mathcal{F}(\delta)} \left| \frac{1}{n} \sum_{i=1}^n \epsilon_i u(X_i) \right| + M\sqrt{\frac{2t}{n}} \\
&\leq 2\mathbb{E} \sup_{u \in \mathcal{F}} \left| \frac{1}{n} \sum_{i=1}^n \epsilon_i u(X_i) \right| + M\sqrt{\frac{2t}{n}}.
\end{aligned}
\tag{146}
$$

The first term is the expectation of the empirical process and it can be easily bounded by using Dudley's theorem.

Specifically,

$$
\begin{aligned}
\mathbb{E} \sup_{u \in \mathcal{F}} \left| \frac{1}{n} \sum_{i=1}^{n} \epsilon_i u(X_i) \right| &= \mathbb{E}_X \mathbb{E}_\epsilon \sup_{u \in \mathcal{F} \cup (-\mathcal{F})} \frac{1}{n} \sum_{i=1}^{n} \epsilon_i u(X_i) \\
&\leq \mathbb{E}_X \left[ \frac{12}{\sqrt{n}} \int_0^M \sqrt{\log \mathcal{N}(\mathcal{F} \cup (-\mathcal{F}), \| \cdot \|_{L^2(P_n)}, u)} du \right] \\
&\leq \mathbb{E}_X \left[ \frac{12}{\sqrt{n}} \int_0^M \sqrt{\log 2 \mathcal{N}(\mathcal{F}, \| \cdot \|_{L^2(P_n)}, u)} du \right] \\
&\leq \frac{12}{\sqrt{n}} \int_0^M \sqrt{\log 2 + a \log \frac{b}{u}} du \\
&\leq \frac{12}{\sqrt{n}} \left( \sqrt{\log 2} M + \sqrt{a} b \int_0^{\frac{M}{b}} \sqrt{\log \frac{1}{u}} du \right) \\
&\leq \frac{12}{\sqrt{n}} \left( \sqrt{\log 2} M + 2\sqrt{a} M \sqrt{\log \frac{4b}{M}} \right) \\
&\leq C \sqrt{\frac{aM^2}{n} \log \frac{4b}{M}},
\end{aligned}
\tag{147}
$$

where the fifth inequality follows by the fact that $b > M$ and Lemma 13.

Now, it remains only to bound $\psi_n^1(\delta)$.

Recall that

$$
\mathcal{G}(\delta) = \{ (|\nabla u(x)|^2 - 2f(x)u(x)) - (|\nabla u^*(x)|^2 - 2f(x)u^*(x)) : u \in \mathcal{F}, \|u - u^*\|_{H^1(\Omega)}^2 \leq \delta \}.
$$

Therefore, we can deduce that $|g| \leq 6M^2$ and $Var(g) \leq P(g^2) \leq 4M^2\delta$ for any $g \in \mathcal{G}(\delta)$. Then, from Talagrand's inequality for empirical processes (Theorem 2.1 in Bartlett et al. (2005) with $\alpha = 1$), we obtain that with probability at least $1 - e^{-t}$

$$
\sup_{g \in \mathcal{G}(\delta)} (P - P_n) g \leq 4 \mathbb{E} \mathcal{R}_n(\mathcal{G}(\delta)) + \sqrt{\frac{8M^2 t \delta}{n}} + \frac{16M^2 t}{n}.
\tag{148}
$$

Note that $Pg^2 \leq 4M^2\delta$ for any $g \in \mathcal{G}(\delta)$, therefore

$$
\mathbb{E} \mathcal{R}_n(\mathcal{G}(\delta)) \leq \mathbb{E} \mathcal{R}_n(g \in \mathcal{G} : Pg^2 \leq 4M^2\delta).
$$

The right term frequently appears in the articles related to the LRC and can be more easily handled than the term on the left.

By applying Corollary 2.1 in Lei et al. (2016) under the assumption for the empirical covering number of $\mathcal{G}$, we know

$$
\mathbb{E} \mathcal{R}_n(g \in \mathcal{G} : Pg^2 \leq 4M^2\delta) \leq C \left( \frac{\alpha M^2 \log(2\beta \sqrt{n})}{n} + \sqrt{\frac{M^2 \delta \alpha \log(2\beta \sqrt{n})}{n}} \right),
\tag{149}
$$

where $C$ is a universal constant.

Combining the upper bounds for $\psi_n^{(1)}(\delta), \psi_n^{(2)}(\delta)$ and $\psi_n^{(3)}(\delta)$, i.e. (144), (145), (147) and (149), the conclusion holds. $\qquad\square$

**Lemma 15.** *For the empirical covering number of $\mathcal{F}$ and $\mathcal{G}$ defined in the Lemma 14, we can deduce that*

*(1) when $\mathcal{F} = \mathcal{F}_{m,1}(B)$, we have*

$$\mathcal{N}(\mathcal{F}, L^2(P_n), \epsilon) \leq \left(\frac{cB}{\epsilon}\right)^{m(d+1)} \ and \ \mathcal{N}(\mathcal{G}, L^2(P_n), \epsilon) \leq \left(\frac{c\max(MB, B^2)}{\epsilon}\right)^{cmd}, \quad (150)$$

*where $M$ is a upper bound for $|f|$ and $c$ is a universal constant.*

*(2) when $\mathcal{F} = \Phi(N, L, B)$, we have*

$$\mathcal{N}(\mathcal{F}, L^2(P_n), \epsilon) \leq \left(\frac{Cn}{\epsilon}\right)^{CN^2L^2(\log N \log L)^3} \ and \ \mathcal{N}(\mathcal{G}, L^2(P_n), \epsilon) \leq \left(\frac{Cn}{\epsilon}\right)^{CN^2L^2(\log N \log L)^3},$$
$$(151)$$

*where $C$ is a constant independent of $N, L$ and $n \geq CN^2L^2(\log N \log L)^3$.*

*Proof.* (1) For the function class of two-layer neural networks, recall that

$$\mathcal{F}_{m,1}(B) = \left\{\sum_{i=1}^{m} \gamma_i \sigma(\omega_i \cdot x + t_i) : \sum_{i=1}^{m} |\gamma_i| \leq B, |\omega_i|_1 = 1, t_i \in [-1, 1)\right\}.$$

Due to the Lipschitz continuity of $\sigma$, we can just consider the covering number in the $L^\infty$ norm.

Without loss of generality, we can assume that $B = 1$. Then for

$$u_k(x) = \sum_{i=1}^{m} \gamma_i^k \sigma(\omega_i^k \cdot x + t_i^k) \in \mathcal{F}_{m,1}(1), k = 1, 2,$$

we have

$$|u_1(x) - u_2(x)| = |\sum_{i=1}^{m} \gamma_i^1 \sigma(\omega_i^1 \cdot x + t_i^1) - \gamma_i^2 \sigma(\omega_i^2 \cdot x + t_i^2)|$$

$$\leq \sum_{i=1}^{m} |\gamma_i^1 \sigma(\omega_i^1 \cdot x + t_i^1) - \gamma_i^2 \sigma(\omega_i^2 \cdot x + t_i^2)|$$

$$= \sum_{i=1}^{m} |(\gamma_i^1 - \gamma_i^2)\sigma(\omega_i^1 \cdot x + t_i^1) + \gamma_i^2(\sigma(\omega_i^1 \cdot x + t_i^1) - \sigma(\omega_i^2 \cdot x + t_i^2))|$$

$$\leq \sum_{i=1}^{m} 2|\gamma_i^1 - \gamma_i^2| + |\gamma_i^2|(|\omega_i^1 - \omega_i^2|_1 + |t_i^1 - t_i^2|),$$

where the last inequality follows from that $\sigma$ is bounded by 2 in absolute value and is 1-Lipschitz continuous.

Therefore, when

$$\sum_{i=1}^{m} |\gamma_i^1 - \gamma_i^2| \leq \frac{\epsilon}{4} \ and \ |\omega_i^1 - \omega_i^2|_1 \leq \frac{\epsilon}{4}, |t_i^1 - t_i^2| \leq \frac{\epsilon}{4}, 1 \leq i \leq m,$$

we have that $\sup_{x \in \Omega} |u_1(x) - u_2(x)| \leq \epsilon$, which implies

$$\mathcal{N}(\mathcal{F}_{m,1}(1), L^2(P_n), \epsilon) \leq \mathcal{N}(\mathcal{F}_{m,1}(1), L^\infty, \epsilon) \leq \left(\frac{c}{\epsilon}\right)^m \left(\frac{c}{\epsilon}\right)^{m(d-1)} \left(\frac{c}{\epsilon}\right)^m = \left(\frac{c}{\epsilon}\right)^{m(d+1)},$$

where $c$ is a universal constant.

Therefore, $\mathcal{N}(\mathcal{F}_{m,1}(B), L^2(P_n), \epsilon) \leq \left(\frac{cB}{\epsilon}\right)^{m(d+1)}$, where we assume that $B \geq 1$.

Recall that

$$\mathcal{G} = \{(|\nabla u(x)|^2 - 2f(x)u(x)) - (|\nabla u^*(x)|^2 - 2f(x)u^*(x)) : u \in \mathcal{F}\}.$$

Since $u^*$ is fixed, the estimation for the term $f(x)u(x)$ can be conducted in the same manner as for $\mathcal{F}$. Therefore, we only need to estimate the first term.

For

$$u_k = \sum_{i=1}^{m} \gamma_i^k \sigma(\omega_i^k \cdot x + t_i^k) \in \mathcal{F}_m(1), k = 1, 2$$

we have

$$\||\nabla u_1|^2 - |\nabla u_2|^2\|_{L^2(P_n)}$$

$$\leq 2\||\nabla u_1 - \nabla u_2\|_{L^2(P_n)}$$

$$\leq 2\|\sum_{i=1}^{m} |\gamma_i^1 \omega_i^1 I_{\{\omega_i^1 \cdot x + t_i^1 \geq 0\}} - \gamma_i^2 \omega_i^2 I_{\{\omega_i^2 \cdot x + t_i^2 \geq 0\}}\|_{L^2(P_n)}$$

$$\leq 2\sum_{i=1}^{m} \||\gamma_i^1 \omega_i^1 I_{\{\omega_i^1 \cdot x + t_i^1 \geq 0\}} - \gamma_i^2 \omega_i^2 I_{\{\omega_i^2 \cdot x + t_i^2 \geq 0\}}\|_{L^2(P_n)}$$

$$= 2\sum_{i=1}^{m} \||(\gamma_i^1 - \gamma_i^2)\omega_i^1 I_{\{\omega_i^1 \cdot x + t_i^1 \geq 0\}} + \gamma_i^2(\omega_i^1 I_{\{\omega_i^1 \cdot x + t_i^1 \geq 0\}} - \omega_i^2 I_{\{\omega_i^2 \cdot x + t_i^2 \geq 0\}})\|_{L^2(P_n)}$$

$$\leq 2\sum_{i=1}^{m} |\gamma_i^1 - \gamma_i^2| + 2\sum_{i=1}^{m} |\gamma_i^2|\|\omega_i^1 I_{\{\omega_i^1 \cdot x + t_i^1 \geq 0\}} - \omega_i^2 I_{\{\omega_i^2 \cdot x + t_i^2 \geq 0\}}\|_{L^2(P_n)}$$

$$\leq 2\sum_{i=1}^{m} |\gamma_i^1 - \gamma_i^2| + 2\sum_{i=1}^{m} |\gamma_i^2|(|\omega_i^1 - \omega_i^2|_1 + \|I_{\{\omega_i^1 \cdot x + t_i^1 \geq 0\}} - I_{\{\omega_i^2 \cdot x + t_i^2 \geq 0\}}\|_{L^2(P_n)}),$$

where the first inequality follows from that $|\nabla u_k| \leq |\nabla u_k|_1 \leq 1$ for $k = 1, 2$ and the second, third, fourth and the last inequalities follow from the triangle inequality.

Thus if

$$\sum_{i=1}^{m} |\gamma_i^1 - \gamma_i^2| \leq \frac{\epsilon}{4} \ and \ |\omega_i^1 - \omega_i^2|_1 + \|I_{\{\omega_i^1 \cdot x + t_i^1 \geq 0\}} - I_{\{\omega_i^2 \cdot x + t_i^2 \geq 0\}}\|_{L^2(P_n)} \leq \frac{\epsilon}{4}, 1 \leq i \leq m,$$

we can deduce that $\||\nabla u_1|^2 - |\nabla u_2|^2\|_{L^2(P_n)} \leq \epsilon$.

Based on same method in the proof of Proposition 8, the $L^2(P_n)$ covering number of the function class $\{|\nabla u|^2 : u \in \mathcal{F}\}$ can be bounded as

$$\left(\frac{c}{\epsilon}\right)^m \left(\frac{c}{\epsilon}\right)^{(d-1+2d)m} = \left(\frac{c}{\epsilon}\right)^{3md}.$$

Combining the result for $\mathcal{F}$, we obtain that

$$\mathcal{N}(\mathcal{G}, L^2(P_n), \epsilon) \leq \left(\frac{c\max(MB, B^2)}{\epsilon}\right)^{cmd},$$

where $M$ is a upper bound for $|f|$ and $c$ is a universal constant.

(2) Note that the empirical covering number $\mathcal{N}(\mathcal{F}, L^2(P_n), \epsilon)$ can be bounded by the uniform covering number $\mathcal{N}(\mathcal{F}, n, \epsilon)$, which is defined as

$$\mathcal{N}(\mathcal{F}, n, \epsilon) := \sup_{Z_n \in \mathcal{X}^n} \mathcal{N}(\mathcal{F}|_{Z_n}, \epsilon, \|\cdot\|_\infty),$$

where $Z_n = (z_1, \cdots, z_n)$ and $\mathcal{F}|_{Z_n} := \{(f(z_1), \cdots, f(z_n)) : f \in \mathcal{F}\}$.

As for the uniform covering number, it can be estimated using the pseudo-dimension $Pdim(\mathcal{F})$. Specifically, let $\mathcal{F}$ be a class of function from $\mathcal{X}$ to $[-B, B]$. Then for any $\epsilon > 0$, we have

$$\mathcal{N}(\mathcal{F}, n, \epsilon) \leq \left(\frac{2enB}{\epsilon Pdim(\mathcal{F})}\right)^{Pdim(\mathcal{F})}$$

for $n \geq Pdim(\mathcal{F})$ (See Theorem 12.2 in Anthony et al. (1999)).

From Bartlett et al. (2019) and Yang et al. (2023b), we know that

$$Pdim(\Psi) \leq CN^2L^2 \log L \log N \ and \ Pdim(D\Psi) \leq CN^2L^2 \log L \log N$$

with a constant $C$ independent with $N, L$, where $\Psi$ is the function class of ReLU neural networks with width $N$ and depth $L$.

Therefore, we can deduce that for $\mathcal{F} = \Phi(N, L, B)$, we have

$$\mathcal{N}(\mathcal{F}, L^2(P_n), \epsilon) \leq \left(\frac{Cn}{\epsilon}\right)^{CN^2L^2(\log N \log L)^3}$$

and

$$\mathcal{N}(\mathcal{G}, L^2(P_n), \epsilon) \leq \left(\frac{Cn}{\epsilon}\right)^{CN^2L^2(\log N \log L)^3}$$

with a constant $C$ independent of $N, L$ and $n \geq CN^2L^2(\log N \log L)^3$, as the width and depth of $\Phi(N, L, B)$ are $\mathcal{O}(N \log N)$ and $\mathcal{O}(L \log L)$ respectively. $\qquad\square$

**Lemma 16** (Estimation of the covering numbers for PINNs)**.**

*(1) For $\mathcal{F} = \mathcal{F}_{m,2}(B)$ with $B = \mathcal{O}(M)$, we have*

$$\log \mathcal{N}(\boldsymbol{\mathcal{F}}, d, \epsilon) \leq cmd \log\left(\frac{b}{\epsilon}\right)$$

*with a universal constant $c$.*

*(2) For $\mathcal{F} = \Phi(L, W, S, B; H)$ with $L = \mathcal{O}(1), W = \mathcal{O}(K^d), S = \mathcal{O}(K^d), B = 1, H = \mathcal{O}(1)$, we have*

$$\log \mathcal{N}(\boldsymbol{\mathcal{F}}, d, \epsilon) \leq CK^d \log\left(\frac{K}{\epsilon}\right),$$

*where $C$ is a constant independent of $K$.*

*Proof.* Recall that

$$(Lu - f)^2 = \left(-\sum_{i,j=1}^d a_{ij}(x)\partial_{i,j}u(x) + \sum_{i=1}^d b_i(x)\partial_i u(x) + c(x)u(x) - f(x)\right)^2$$

and

$$\boldsymbol{\mathcal{F}} = \{\boldsymbol{u} = (|\Omega|(Lu(x) - f(x))^2, |\partial\Omega|(u(y) - g(y))^2) : \ u \in \mathcal{F}\}.$$

(1) For the two functions $\boldsymbol{u} = (|\Omega|(Lu - f)^2, |\partial\Omega|(u - g)^2), \bar{\boldsymbol{u}} = (|\Omega|(L\bar{u} - f)^2, |\partial\Omega|(\bar{u} - g)^2) \in \boldsymbol{\mathcal{F}}$, where $u, \bar{u}$ belong to $\mathcal{F}_{m,2}(B)$ and are of the form

$$u(x) = \sum_{k=1}^m \gamma_k \sigma_2(\omega_k \cdot x + t_k), \ \bar{u}(x) = \sum_{k=1}^m \bar{\gamma}_k \sigma_2(\bar{\omega}_k \cdot x + \bar{t}_k)$$

respectively. We write $\boldsymbol{u}, \bar{\boldsymbol{u}}$ as $(u_1, u_2)$ and $(\bar{u}_1, \bar{u}_2)$ for simplicity.

As for the samples from $\Omega$ and $\partial\Omega$, we denote their empirical measure as

$$P_{N_1} := \frac{1}{N_1}\sum_{i=1}^{N_1}\delta_{X_i} \ and \ P_{N_2} := \frac{1}{N_2}\sum_{i=1}^{N_2}\delta_{Y_i},$$

respectively.

Now, we are ready to estimate $d(\boldsymbol{u}, \bar{\boldsymbol{u}})$, recall that

$$d(\boldsymbol{u}, \bar{\boldsymbol{u}}) = \sqrt{\frac{1}{2}}\sqrt{\|u_1 - \bar{u}_1\|_{L^2(P_{N_1})}^2 + \|u_2 - \bar{u}_2\|_{L^2(P_{N_2})}^2}$$

$$\leq \sqrt{\frac{1}{2}}(\|u_1 - \bar{u}_1\|_{L^2(P_{N_1})} + \|u_2 - \bar{u}_2\|_{L^2(P_{N_2})}),$$

which allows us to estimate these two terms separately.

From the boundedness of related functions, we have

$$\|u_1 - \bar{u}_1\|_{L^2(P_{N_1})} = \||\Omega|(Lu - f)^2 - |\Omega|(L\bar{u} - f)^2\|_{L^2(P_{N_1})}$$
$$\leq cd^2 M^2 |\Omega| \|L(u - \bar{u})\|_{L^2(P_{N_1})}$$

and

$$\|u_2 - \bar{u}_2\|_{L^2(P_{N_2})} = \||\partial\Omega|(u - g)^2 - |\partial\Omega|(\bar{u} - g)^2\|_{L^2(P_{N_2})}$$
$$\leq cM|\partial\Omega| \|u - \bar{u}\|_{L^2(P_{N_2})}.$$

Therefore, it can be turned to bound $\|L(u - \bar{u})\|_{L^2(P_{N_1})}$ and $\|u - \bar{u}\|_{L^2(P_{N_2})}$.

For $\|L(u - \bar{u})\|_{L^2(P_{N_1})}$, applying the triangle inequality yields

$$\|L(u - \bar{u})\|_{L^2(P_{N_1})} = \| -\sum_{i,j=1}^{d} a_{ij}\partial_{i,j}(u - \bar{u}) + \sum_{i=1}^{d} b_i\partial_i(u - \bar{u}) + c(u - \bar{u})\|_{L^2(P_{N_1})}$$

$$\leq \|\sum_{i,j=1}^{d} a_{ij}\partial_{i,j}(u - \bar{u})\|_{L^2(P_{N_1})} + \|\sum_{i=1}^{d} b_i\partial_i(u - \bar{u})\|_{L^2(P_{N_1})} + \|c(u - \bar{u})\|_{L^2(P_{N_1})}$$

$$:= A_1 + A_2 + A_3.$$

Note that $\partial_i u, u$ are Lipschitz continuous with respect to the parameters, thus for $A_2$, we have

$$A_2 = \|\sum_{i=1}^{d} b_i\partial_i(u - \bar{u})\|_{L^2(P_{N_1})}$$

$$\leq \|\sum_{i=1}^{d} b_i\partial_i(u - \bar{u})\|_{L^\infty(\Omega)}$$

$$= \|\sum_{i=1}^{d} 2b_i \left( \sum_{k=1}^{m} \gamma_k\omega_k^i\sigma(\omega_k \cdot x + t_k) - \bar{\gamma}_k\bar{\omega}_k^i\sigma(\bar{\omega}_k \cdot x + \bar{t}_k) \right)\|_{L^\infty(\Omega)}$$

$$= \|\sum_{i=1}^{d} 2b_i \left( \sum_{k=1}^{m} (\gamma_k - \bar{\gamma}_k)\omega_k^i\sigma(\omega_k \cdot x + t_k) + \bar{\gamma}_k\omega_k^i\sigma(\omega_k \cdot x + t_k) - \bar{\gamma}_k\bar{\omega}_k^i\sigma(\bar{\omega}_k \cdot x + \bar{t}_k) \right)\|_{L^\infty(\Omega)}$$

$$\leq 4M \sum_{k=1}^{m} |\gamma_k - \bar{\gamma}_k| + 2M \sum_{i=1}^{d} \|\sum_{k=1}^{m} \bar{\gamma}_k\omega_k^i\sigma(\omega_k \cdot x + t_k) - \bar{\gamma}_k\bar{\omega}_k^i\sigma(\bar{\omega}_k \cdot x + \bar{t}_k)\|_{L^\infty(\Omega)},$$

where the last inequality follows from the facts that $|b_i| \leq M, 1 \leq i \leq d$ and $\omega_k = (\omega_k^1, \cdots, \omega_k^d)$, $\sum_{i=1}^{d} |\omega_k^i| = 1$. And we denote the second term by $A_{22}$, then

$$A_{22} = 2M \sum_{i=1}^{d} \|\sum_{k=1}^{m} \bar{\gamma}_k\omega_k^i\sigma(\omega_k \cdot x + t_k) - \bar{\gamma}_k\bar{\omega}_k^i\sigma(\bar{\omega}_k \cdot x + \bar{t}_k)\|_{L^\infty(\Omega)}$$

$$= 2M \sum_{i=1}^{d} \|\sum_{k=1}^{m} \bar{\gamma}_k(\omega_k^i - \bar{\omega}_k^i)\sigma(\omega_k \cdot x + t_k) + \bar{\gamma}_k\bar{\omega}_k^i(\sigma(\omega_k \cdot x + t_k) - \sigma(\bar{\omega}_k \cdot x + \bar{t}_k))\|_{L^\infty(\Omega)}$$

$$\leq 4M \sum_{i=1}^{d} \sum_{k=1}^{m} |\bar{\gamma}_k||\omega_k^i - \bar{\omega}_k^i| + 2M \sum_{i=1}^{d} \sum_{k=1}^{m} |\bar{\gamma}_k||\bar{\omega}_k^i|(|\omega_k - \bar{\omega}_k|_1 + |t_k - \bar{t}_k|)$$

$$= 4M \sum_{k=1}^{m} |\bar{\gamma}_k||\omega_k - \bar{\omega}_k|_1 + 2M \sum_{k=1}^{m} |\bar{\gamma}_k|(|\omega_k - \bar{\omega}_k|_1 + |t_k - \bar{t}_k|),$$

where the inequality follows from the triangle inequality and the facts that $\sigma$ is 1-Lipschitz continuous and $\|\sigma\|_{L^\infty([-2,2])} \leq 2$.

Combining the results for $A_2$, we have

$$A_2 \leq 4M \sum_{k=1}^{m} |\gamma_k - \bar{\gamma}_k| + 4M \sum_{k=1}^{m} |\bar{\gamma}_k||\omega_k - \bar{\omega}_k|_1 + 2M \sum_{k=1}^{m} |\bar{\gamma}_k|(|\omega_k - \bar{\omega}_k|_1 + |t_k - \bar{t}_k|).$$

Similarly, we have

$$A_3 = \|c(u - \bar{u})\|_{L^2(P_{N_1})}$$

$$\leq 4M \sum_{k=1}^{m} |\gamma_k - \bar{\gamma}_k| + 4M \sum_{k=1}^{m} |\bar{\gamma}_k|(|\omega_k - \bar{\omega}_k|_1 + |t_k - \bar{t}_k|)$$

and

$$\|u - \bar{u}\|_{L^2(P_{N_2})} \leq 4 \sum_{k=1}^{m} |\gamma_k - \bar{\gamma}_k| + 4 \sum_{k=1}^{m} |\bar{\gamma}_k|(|\omega_k - \bar{\omega}_k|_1 + |t_k - \bar{t}_k|).$$

As $A_1$ involves the second derivative of $\sigma_2$, the method described above cannot be applied. However, we can borrow the idea from the proof of Proposition 8.

$$A_1 = \| \sum_{i,j=1}^{d} a_{ij} \partial_{i,j}(u - \bar{u})\|_{L^2(P_{N_1})}$$

$$= 2\| \sum_{k=1}^{m} \gamma_k \omega_k^T A \omega_k I_{\{\omega_k \cdot x + t_k \geq 0\}} - \bar{\gamma}_k \bar{\omega}_k^T A \bar{\omega}_k I_{\{\bar{\omega}_k \cdot x + \bar{t}_k \geq 0\}}\|_{L^2(P_{N_1})}$$

$$= 2\| \sum_{k=1}^{m} (\gamma_k \omega_k^T A \omega_k - \bar{\gamma}_k \bar{\omega}_k^T A \bar{\omega}_k) I_{\{\omega_k \cdot x + t_k \geq 0\}} + \bar{\gamma}_k \bar{\omega}_k^T A \bar{\omega}_k (I_{\{\omega_k \cdot x + t_k \geq 0\}} - I_{\{\bar{\omega}_k \cdot x + \bar{t}_k \geq 0\}})\|_{L^2(P_{N_1})}$$

$$\leq 2 \sum_{k=1}^{m} |\gamma_k \omega_k^T A \omega_k - \bar{\gamma}_k \bar{\omega}_k^T A \bar{\omega}_k| + 2 \sum_{k=1}^{m} |\bar{\gamma}_k \bar{\omega}_k^T A \bar{\omega}_k| \| I_{\{\omega_k \cdot x + t_k \geq 0\}} - I_{\{\bar{\omega}_k \cdot x + \bar{t}_k \geq 0\}}\|_{L^2(P_{N_1})}.$$

For the first term, we have

$$\sum_{k=1}^{m} |\gamma_k \omega_k^T A \omega_k - \bar{\gamma}_k \bar{\omega}_k^T A \bar{\omega}_k| \leq \sum_{k=1}^{m} |(\gamma_k - \bar{\gamma}_k)\omega_k^T A \omega_k| + |\bar{\gamma}_k(\omega_k^T A \omega_k - \bar{\omega}_k^T A \bar{\omega}_k)|$$

$$\leq \sum_{k=1}^{m} M|\gamma_k - \bar{\gamma}_k| + |\bar{\gamma}_k||\omega_k^T A(\omega_k - \bar{\omega}_k) + \bar{\omega}_k^T A(\omega_k - \bar{\omega}_k)|$$

$$\leq M \left( \sum_{k=1}^{m} |\gamma_k - \bar{\gamma}_k| + 2|\bar{\gamma}_k||\omega_k - \bar{\omega}_k|_1 \right),$$

where the inequalities follow from the triangle inequality and the fact that for any $x \in \partial B_1^d(1), y \in \mathbb{R}^d$ and matrix $A \in \mathbb{R}^{d \times d}$ with $|A(i,j)| \leq M(1 \leq i,j \leq d)$, we have $|x^T A y| = |(A^T x)^T y| \leq |A^T x|_\infty |y|_1 \leq M|y|_1$.

Thus we obtain the final upper bound for $A_1$.

$$A_1 \leq 2M \sum_{k=1}^{m} (|\gamma_k - \bar{\gamma}_k| + 2|\bar{\gamma}_k||\omega_k - \bar{\omega}_k|_1) + 2M \sum_{k=1}^{m} |\bar{\gamma}_k| \| I_{\{\omega_k \cdot x + t_k \geq 0\}} - I_{\{\bar{\omega}_k \cdot x + \bar{t}_k \geq 0\}}\|_{L^2(P_{N_1})}.$$

Combining all results above, we can deduce that

$$d(\boldsymbol{u}, \hat{\boldsymbol{u}}) \leq c(d^2 M^3 |\Omega| + M|\partial\Omega|)(\sum_{k=1}^{m} (|\gamma_k - \bar{\gamma}_k| + |\bar{\gamma}_k||\omega_k - \bar{\omega}_k|_1)$$

$$+ \sum_{k=1}^{m} |\bar{\gamma}_k| \| I_{\{\omega_k \cdot x + t_k \geq 0\}} - I_{\{\bar{\omega}_k \cdot x + \bar{t}_k \geq 0\}}\|_{L^2(P_{N_1})}).$$

Similar to bounding the empirical covering number of $\mathcal{G}$ for the two-layer neural networks in Lemma 15(1), the covering number of $\mathcal{F}$ under $d$ is

$$\left( \frac{c(d^2 M^3 |\Omega| + M|\partial\Omega|)B}{\epsilon} \right)^{cmd} \leq \left( \frac{c(d^2 M^4 |\Omega| + M^2 |\partial\Omega|)}{\epsilon} \right)^{cmd} \leq \left( \frac{cb}{\epsilon} \right)^{cmd},$$

where $c$ is a universal constant.

(2) Note that $d(\boldsymbol{u}, \bar{\boldsymbol{u}}) \leq C\|u - \bar{u}\|_{C^2(\bar{\Omega})}$, then Proposition 1 Belomestny et al. (2024) implies that

$$\log \mathcal{N}(\mathcal{F}, \|\cdot\|_{C^2(\bar{\Omega})}, \epsilon) \leq CK^d \log\left( \frac{K}{\epsilon} \right),$$

where $C$ is a constant independent of $K$.

Therefore, the conclusion holds. $\qquad\square$

**Lemma 17** (Agmon et al. (1959)). *For $u \in H^{\frac{1}{2}}(\Omega) \cap L^2(\partial\Omega)$,*

$$
\begin{aligned}
\|u\|^2_{H^{\frac{1}{2}}(\Omega)} &\leq C \left\| - \sum_{i,j=1}^d a_{ij}\partial_{ij}u + \sum_{i=1}^d b_i\partial_i u + cu \right\|^2_{H^{-\frac{3}{2}}(\Omega)} + C\|u\|^2_{L^2(\partial\Omega)} \\
&\leq C_\Omega \left( \| - \sum_{i,j=1}^d a_{ij}\partial_{ij}u + \sum_{i=1}^d b_i\partial_i u + cu \|^2_{L^2(\Omega)} + \|u\|^2_{L^2(\partial\Omega)} \right),
\end{aligned}
\tag{152}
$$

*where $C_\Omega$ is a constant that depends only on $\Omega$.*

# D DISCUSSION

## D.1 OVER-PARAMETERIZED SETTING

In the context of over-parameterization, the generalization bounds for two-layer neural networks may become less meaningful due to the term $m/n$. However, fortunately, the function class of two-layer neural networks in Proposition 2 and Proposition 4 forms a convex hull of a function class with a covering number similar to that of VC-classes. Consequently, we can extend the convex hull entropy theorem (Theorem 2.6.9 in Vaart & Wellner (2023)) to the $H^1$ norm, allowing us to derive generalization bounds that are independent of the network's width. Theorem 10 is a modification of Theorem 2.6.9 in Vaart & Wellner (2023) to obtain explicit dependence on the dimension.

**Lemma 18.** *Let $\mathcal{F}$ be arbitrary set consisting of $n$ measurable function $f : \Omega \to \mathbb{R}$ of finite $H^1(Q)$-diameter $diam(\mathcal{F})$. Then for every $\epsilon > 0$, we have*

$$\mathcal{N}(\epsilon\, diam(\mathcal{F}), conv(\mathcal{F}), H^1(Q)) \leq \left( e + \frac{en\epsilon^2}{2} \right)^{\frac{2}{\epsilon^2}}.$$

*Proof.* Assume that $\mathcal{F} = \{f_1, \cdots, f_n\}$. For given $\lambda$ in the $n$-dimensional simplex. Let $Y_1, \cdots, Y_k$ be i.i.d. random elements such that $P(Y_1 = f_j) = \lambda_i$ for $j = 1, \cdots, k$ and $k$ is natural number to be determined. Then we have

$$\mathbb{E}Y_i = \sum_{j=1}^n \lambda_j f_j \ and \ \nabla\mathbb{E}Y_i = \mathbb{E}\nabla Y_i = \sum_{j=1}^n \lambda_j \nabla f_j.$$

Let $\bar{Y}_k = \frac{1}{k}\sum_{i=1}^k Y_i$, then the independence implies

$$\mathbb{E}\|\bar{Y}_k - \mathbb{E}Y_1\|^2_{H^1(Q)} = \frac{1}{k^2}\sum_{i=1}^k \mathbb{E}\|Y_i - \mathbb{E}Y_1\|^2_{H^1(Q)} \leq \frac{1}{k}(\text{diam}(\mathcal{F}))^2.$$

Therefore, Markov inequality implies that there is at least one realization of $\bar{Y}_k$ that have $H^1(Q)$-distance at most $k^{-1/2}\text{diam}(\mathcal{F})$ to the convex combination $\sum_{j=1}^n \lambda_j f_j$. Note that every realization

has the form $k^{-1} \sum_{i=1}^{k} f_{i_k}$, where some functions $f_j$ in the set $\mathcal{F}$ may be used multiple times. As such forms are at most $C_{n+k-1}^{k}$, we can deduce that

$$\mathcal{N}(k^{-1/2}\text{diam}(\mathcal{F}), \text{conv}(\mathcal{F}), H^1(Q)) \leq C_{n+k-1}^{k} \leq e^k(1+\frac{n}{k})^k,$$

where the last inequality follows from Stirling's inequality.

For $0 < \epsilon < 1$, we can take $k = \lceil \frac{1}{\epsilon^2} \rceil$, then the monotonicity of the function $e^k(1+\frac{n}{k})^k$ and the fact $k \leq \frac{1}{\epsilon^2} + 1 \leq \frac{2}{\epsilon^2}$ imply that

$$e^k \left(1 + \frac{n}{k}\right)^k \leq \left(e + \frac{en\epsilon^2}{2}\right)^{\frac{2}{\epsilon^2}}. \tag{153}$$

For $\epsilon > 1$, the right term in (153) is larger than 1, thus the conclusion holds directly. $\qquad \square$

**Theorem 10.** *Let $Q$ be a probability on $\Omega$, and let $\mathcal{F}$ be a class of measurable functions with $\|F\|_{Q,2} := \sup_{f \in \mathcal{F}} \|f\|_{H^1(Q)} < \infty$ and*

$$\mathcal{N}(\epsilon\|F\|_{Q,2}, \mathcal{F}, H^1(Q)) \leq C\left(\frac{1}{\epsilon}\right)^V, \ 0 < \epsilon < 1$$

*for some $V \geq 1$. Then we have*

$$\log \mathcal{N}(\epsilon\|F\|_{Q,2}, \text{conv}(\mathcal{F}), H^1(Q)) \leq KV(C^{\frac{1}{V}} + 2)^{\frac{2V}{V+2}}\left(\frac{1}{\epsilon}\right)^{\frac{2V}{V+2}},$$

*where $K$ is a universal constant.*

*Proof.* Note that every element in the convex hull of $\mathcal{F}$ has distance $\epsilon$ to the convex hull of an $\epsilon$-net over $\mathcal{F}$. Accordingly, given a fixed $\epsilon$, it suffices to consider scenarios where the set $\mathcal{F}$ is finite.

Set $W = \frac{1}{2} + \frac{1}{V}$ and $L = C^{1/V}\|F\|_{Q,2}$. Then the assumption implies that $\mathcal{F}$ can be covered by $n$ balls of radius at most $Ln^{-1/V}$ for every natural number $n$. Form sets $\mathcal{F}_1 \subset \mathcal{F}_2 \subset \cdots \subset \mathcal{F}$ such that for each $n$, the set $\mathcal{F}_n$ is a maximal, $Ln^{-1/V}$-separated net over $\mathcal{F}$. Thus $\mathcal{F}_n$ has at most $n$ elements. We will show by induction that there exist constant $C_k$ and $D_k$ depending only on $C$ and $V$ such that $\sup_k C_j \vee D_k < \infty$ and for $q \geq 3V$,

$$\log \mathcal{N}(C_k Ln^{-W}, \text{conv}(\mathcal{F}_{nk^q}), H^1(Q)) \leq D_k n, \ n, k \geq 1.$$

The proof consists of a nested induction argument. The outer layer is induction on $k$ and the inner layer is induction on $n$.

First, we apply induction for $n$, i.e., for $k = 1$, we will prove the conclusion for each $n$. For fixed $n_0 = 10$, it suffices to choose $C_1 Ln_0^{-W} = C_1 10^{-W} \geq \|F\|_{Q,2}$ so that the statement is trivially ture for $n \leq n_0 = 10$, i.e., $C_1 \geq 10^W C^{-1/V}$. For $10 < n \leq 100$, set $m = \lfloor \frac{n}{10} \rfloor$, thus $1 \leq m \leq 10$. By the definition of $\mathcal{F}_m$, each $f \in \mathcal{F}_n - \mathcal{F}_m$ has distance at most $Lm^{-1/V}$ of some element $\pi_m f$ of $\mathcal{F}_m$. Thus each element of $\text{conv}(\mathcal{F})$ can be written as

$$\sum_{f \in \mathcal{F}_n} \lambda_f f = \sum_{f \in \mathcal{F}_m} \mu_f f + \sum_{f \in \mathcal{F}_n - \mathcal{F}_m} \lambda_f(f - \pi_m f),$$

where $\mu_f \geq 0$ and $\sum \mu_f = \sum \lambda_f = 1$. Taking $\mathcal{G}$ as the set of function $f - \pi_m f$ with $f$ ranging over $\mathcal{F}_n - \mathcal{F}_m$, thus $\text{conv}(\mathcal{F}_n) \subset \text{conv}(\mathcal{F}_m) + \text{conv}(\mathcal{G}_n)$ for a set $\mathcal{G}_n$ consisting of at most $n$ elements, each of norm smaller than $Lm^{-1/V}$, then $\text{diam}(\mathcal{G}_n) \leq 2Lm^{-1/V}$. Applying Lemma 17 for $\mathcal{G}_n$ with $\epsilon$ defined by $m^{-1/V}\epsilon = \frac{1}{4}C_1 n^{-W}$, i.e., $\epsilon\text{diam}(\mathcal{G}_n) \leq \frac{1}{2}C_1 Ln^{-W}$, we can find a $\frac{1}{2}C_1 Ln^{-W}$-net over $\text{conv}(\mathcal{G}_n)$ consisting of at most

$$\left(e + \frac{en\epsilon^2}{2}\right)^{2/\epsilon^2} = \left(e + \frac{eC_1^2}{32}\left(\frac{m}{n}\right)^{\frac{2}{V}}\right)^{\frac{32n}{C_1^2}\left(\frac{n}{m}\right)^{\frac{2}{V}}} \leq \left(e + \frac{eC_1^2}{32}\left(\frac{1}{20}\right)^{\frac{2}{V}}\right)^{\frac{32n}{C_1^2}20^{\frac{2}{V}}}$$

elements, where the inequality follows from the facts that $(e + enx)^{\frac{1}{x}}$ is increasing with respect to $x > 0$ and $\lfloor \frac{n}{10} \rfloor \geq \frac{1}{2} \frac{n}{10}$ for $n \geq 10$. Applying the induction hypothesis to $\mathcal{F}_m$ to find a $C_1 L m^{-W}$-net over $\text{conv}(\mathcal{F}_m)$ consisting of at most $e^m$ elements, where we choose $D_1 = 1$. This defines a partition of $\text{conv}(\mathcal{F}_m)$ into $m$-dimensional sets of radius at most $C_1 L m^{-W}$. Without loss of generality, we can assume that $\mathcal{F}_m = \{f_{i_1}, f_{i_2}, \cdots, f_{i_m}\}$. For any fixed element $h$ in the $C_1 L m^{-W}$-net over $\text{conv}(\mathcal{F}_m)$, assume that $h = \lambda_1 f_{i_1} + \cdots \lambda_m f_{i_m}$ for $\lambda = (\lambda_1, \cdots, \lambda_m) \in \mathbb{R}^m$. And we denote the ball centered at $h$ with $H^1(Q)$ radius $C_1 L m^{-W}$ by

$$H := \{\bar{\lambda} = (\bar{\lambda}_1, \cdots, \bar{\lambda}_m) \in \mathcal{A} : \bar{h} = \bar{\lambda}_1 f_{i_1} + \cdots + \bar{\lambda}_m f_{i_m}, \|\bar{h} - h\|_{H^1(Q)} \leq C_1 L m^{-W}\},$$

where $\mathcal{A}$ is a subset of $\mathbb{R}^m$.

Note that

$$\|h - \bar{h}\|_{H^1(Q)} = \|\lambda_1 f_{i_1} + \cdots \lambda_m f_{i_m} - \bar{\lambda}_1 f_{i_1} - \cdots - \bar{\lambda}_m f_{i_m}\|_{H^1(Q)}$$
$$\leq |\lambda_1 - \bar{\lambda}_1| \|f_{i_1}\|_{H^1(Q)} + \cdots + |\lambda_m - \bar{\lambda}_m| \|f_{i_m}\|_{H^1(Q)}$$
$$\leq (|\lambda_1 - \bar{\lambda}_1| + \cdots + |\lambda_m - \bar{\lambda}_m|) \|F\|_{Q,2}.$$

Thus if $\|\lambda - \bar{\lambda}\|_1 \leq C_1 C^{1/V} m^{-W}$, then $\|h - \bar{h}\|_{H^1(Q)} \leq C_1 L m^{-W}$. Therefore, $\mathcal{A} \subset \{\bar{\lambda} \in \mathbb{R}^m : \|\bar{\lambda} - \lambda\|_1 \leq C_1 C^{1/V} m^{-W}\}$. By Lemma 5.7 in Wainwright (2019), we can find a $\frac{1}{2} C_1 C^{1/V} n^{-W}$-net of $\mathcal{A}$ under the distance $\|\cdot\|_1$ consisting of at most

$$\left( \frac{6 C_1 C^{1/V} m^{-W}}{\frac{1}{2} C_1 C^{1/V} n^{-W}} \right)^m = (12(\frac{n}{m})^W)^m \leq \left(12(20)^W\right)^{\frac{n}{10}}$$

elements. Moreover, it yields a $\frac{1}{2} C_1 L n^{-W}$-net of $H$ under $H^1(Q)$. Select a function from each of the given sets. Then, construct all possible combinations of the sums $f + g$ by preceding procedure, where $f$ is associated with $\text{conv}(\mathcal{F}_m)$ and $g$ is associated with $\text{conv}(\mathcal{G}_n)$. These form a $C_1 L n^{-W}$-net over $\text{conv}(\mathcal{F}_n)$ of cardinality bounded by

$$e^{n/10}(12(20)^W)^{n/10}\left(e + \frac{eC_1^2}{32}\left(\frac{1}{20}\right)^{\frac{2}{V}}\right)^{\frac{32(20)^{\frac{2}{V}} n}{C_1^2}}.$$

This is bounded by $e^n$ for some suitable choice of $C_1$. Specifically, note that for $V \geq 1$, the term attains the maximum at $V = 1$, thus it is bounded by

$$e^{n/10}(12(20)^{\frac{3}{2}})^{n/10}\left(e + \frac{eC_1^2}{32 \cdot 400}\right)^{\frac{32 \cdot 400 n}{C_1^2}}.$$

We can just take $C_1 = 1000$. This concludes the proof for $k = 1$ and $10 < n \leq 100$. Proceeding in the same way yields that the conclusion holds for every $n$.

We continue by induction on $k$. By a similar construction as before, $\text{conv}(\mathcal{F}_{nk^q}) \subset \text{conv}(\mathcal{F}_{n(k-1)^q}) + \text{conv}(\mathcal{G}_{n,k})$ for a set $\text{conv}(\mathcal{G}_{n,k})$ containing at most $nk^q$ elements, each of norm smaller than $L(n(k-1)^q)^{-1/V}$, so that $\text{conv}(\mathcal{G}_{n,k}) \leq 2 L n^{-1/V} k^{-q/V} 2^{q/V}$. Applying Lemma 17 to $\text{conv}(\mathcal{G}_{n,k})$ with $\epsilon = 2^{-1} k^{q/V-2} 2^{-q/V} n^{-1/2}$, we can find an $L k^{-2} n^{-W}$-net over $\text{conv}(\mathcal{G}_{n,k})$ consisting of at most

$$(e + \frac{enk^q \epsilon^2}{2})^{\frac{2}{\epsilon^2}} = \left(e + \frac{ek^{q+\frac{2q}{V}-4}}{2^{\frac{2q}{V}+3}}\right)^{n2^{\frac{2q}{V}+3} k^{4-\frac{2q}{V}}}$$

elements. Apply the induction hypothesis to obtain a $C_{k-1} L n^{-W}$-net over the set $\text{conv}(\mathcal{F}_{n(k-1)^q})$ with respect to $H^1(Q)$ consisting at most $e^{D_{k-1} n}$ elements. Combine the nets as before to obtain a $C_{k-1} L n^{-W}$-net over $\text{conv}(\mathcal{F}_{nk^q})$ consisting of at most $e^{D_k n}$ elements, for

$$C_k = C_{k-1} + \frac{1}{k^2},$$

$$D_k = D_{k-1} + 2^{\frac{2q}{V}+3} \frac{1 + \log(1 + 2^{-\frac{2q}{V}-3} k^{q+\frac{2q}{V}-4})}{k^{2(\frac{q}{V}-2)}}.$$

For $2(\frac{q}{V} - 2) \geq 2$, the resulting sequences $C_k$ and $D_k$ are bounded. By setting $q = 3V$, i.e., $2(\frac{q}{V} - 2) = 2$, we have

$$D_k = D_{k-1} + 2^9 \frac{1 + \log(1 + 2^{-9} k^{3V+2})}{k^2}.$$

Therefore, for any $k$, we can deduce that $C_k \leq C_1 + 2$ and $D_k \leq D_1 + KV$, where $K$ is a universal constant. Recall that $C_1 = \max(10^W C^{-1/V}, 1000)$, thus $\sup_k C_k \leq \max(10^W C^{-1/V}, 1000) + 2$.

Finally,

$$\log \mathcal{N}(\epsilon \|F\|_{Q,2}, \mathrm{conv}(\mathcal{F}), H^1(Q)) \leq \sup_k D_k \left(\frac{C_k C^{\frac{1}{V}}}{\epsilon}\right)^{\frac{2V}{V+2}} \leq KV(C^{\frac{1}{V}} + 2)^{\frac{2V}{V+2}} \left(\frac{1}{\epsilon}\right)^{\frac{2V}{V+2}},$$

where $K$ is a universal constant. $\qquad\square$

For the function class of two-layer neural networks considered in the DRM, i.e.,

$$\mathcal{F} = \{\sigma(\omega \cdot x + t), -\sigma(\omega \cdot x + t), 0 : |\omega|_1 = 1, t \in [-1, 1)\},$$

thus for any probability measure $Q$ on $[0,1]^d$, we have $\|F\|_{Q,2} \leq 3$ and

$$\mathcal{N}(\epsilon \|F\|_{Q,2}, \mathcal{F}, H^1(Q)) \leq C(d+1)(4e)^{d+1} \left(\frac{C}{\epsilon}\right)^{3d},$$

where $C$ is a universal constant.

Then, applying Theorem 10 yields that

$$\log \mathcal{N}(\epsilon \|F\|_{Q,2}, \mathrm{conv}(\mathcal{F}), H^1(Q)) \leq Kd \left(\frac{1}{\epsilon}\right)^{\frac{6d}{3d+2}},$$

where $K$ is a universal constant.

As a result, in Theorem 9 for deriving the generalization error for the static Schrödinger equation, we can deduce that the fixed point $r^*$ satisfies

$$r^* \lesssim d^{\frac{3}{2}} \left(\frac{1}{n}\right)^{\frac{1}{2} + \frac{1}{2(3d+1)}},$$

which yields a meaningful generalization bound in the setting of over-parameterization.

### D.2 OTHER BOUNDARY CONDITIONS FOR DEEP RITZ METHOD

Let $\Omega \subset [0,1]^d$ be a convex bounded open set and $\partial\Omega$ be the boundary of $\Omega$. Consider the elliptic equation on $\Omega$ with Neumann boundary condition:

$$-\Delta u + wu = h \text{ on } \Omega, \quad \frac{\partial u}{\partial n} = g \text{ on } \partial\Omega, \tag{154}$$

where

$$h \in L^\infty(\Omega), \quad g \in H^{\frac{1}{2}}(\partial\Omega), \quad w \in L^\infty(\Omega). \tag{155}$$

From the variation method, the Ritz functional can be defined by

$$\mathcal{E}(u) = \int_\Omega \left(\frac{1}{2}\|\nabla u\|_2^2 + \frac{1}{2} w|u|^2 - hu\right) dx - \int_{\partial\Omega} (gTu) ds, \tag{156}$$

where $T$ is the trace operator.

Then we can deduce that then unique weak solution $u^* \in H^1(\Omega)$ of (154) is the unique minimizer of $\mathcal{E}$ over $H^1(\Omega)$. Moreover, the Ritz functional possesses similar strongly convex property as described in Proposition 1. Specifically, for any $u \in H^1(\Omega)$,

$$\|u - u^*\|_{H^1(\Omega)}^2 \lesssim \mathcal{E}(u) - \mathcal{E}(u^*) \lesssim \|u - u^*\|_{H^1(\Omega)}^2. \tag{157}$$

At this point, to derive the fast rate for equation (156), we can employ the LRC from the multi-task learning setting. This is due to the strongly convex property of the Ritz functional (156), which is similar to the approach used to derive faster generalization bounds for the static Schrödinger equation. Specifically, Theorem B.3 in Yousefi et al. (2018) can be seen as a generalization of Theorem 3.3 in Bartlett et al. (2005), thus combining it with the error decomposition in (77) can lead to the conclusion for the Ritz functional (156). For the sake of brevity, we omit the proof here.

For other boundary conditions, such as the Dirichlet and Robin conditions, see Duan et al. (2021a) and Chen et al. (2024) for discussions on the similarly strong convexity of the Ritz functional, as in equation (157).

### D.3 LIMITATIONS AND FUTURE WORK

- In the paper, we have made the assumption that all related functions are bounded, as required for the localization analysis. However, these assumptions can sometimes be strict. Therefore, it is crucial to investigate settings where the boundedness is not imposed.

- Utilizing ReLU neural networks in the DRM presents optimization challenges due to the non-differentiability of the ReLU function's derivative. One potential approach is to employ randomized methods to tackle the objective functions, like using random neural networks. Despite this, methods for deriving improved generalization error remain valid under stronger assumptions. For instance, when the solutions belong to $\mathcal{B}^3(\Omega)$, employing ReLU$^2$ neural networks allows us to leverage gradient descent or stochastic gradient descent methods.

- For the PINNs, the loss functions play a crucial role for solving PDEs. It is worth paying more attention to the design of loss functions for different PDEs. Moreover, extending the results in Section 3 to other types of PDEs and other PDE solvers involving neural networks is also a topic for future research.

- The optimization error is beyond the scope of this paper. Gao et al. (2023); Luo & Yang (2020) have considered the optimization error of the two-layer neural networks for the PINNs inspired by the work Du et al. (2018). However, it remains open of the optimization aspect for the DRM.

- The requirements of the function class of deep neural networks may be impractical. Achieving these requirements in practice might be accomplished by restricting the weights of the networks, but doing so can make optimization more difficult. Thus, it is worth exploring whether there are more efficient methods.

- The solution theory of PDEs in the Barron spaces remains unclear. Lu et al. (2021c) has addressed the problem for the Poisson and static Schrödinger equations in the Spectral Barron spaces, yielding a priori estimates similar to the standard Sobolev regularity estimate. As for the Barron spaces, Chen et al. (2023) has studied the regularity of solutions to the whole-space static Schrödinger equation in $\mathcal{B}^s(\mathbb{R}^d)$. However, the results of Lu et al. (2021c) and Chen et al. (2023) do not work for $\mathcal{B}^s(\Omega)$. Despite this, at least, there exists solutions in the $\mathcal{B}^s(\Omega)$, as $H^{\frac{d}{2}+s+\epsilon}(\Omega) \subset \mathcal{B}^s(\Omega)$ for any $\epsilon > 0$.

