# OpenReview forum: "Refined Generalization Analysis of the Deep Ritz Method and Physics-Informed Neural Networks"
_ICLR.cc/2025/Conference — Submitted to ICLR 2025_

### Official Review · Reviewer_xW5b · 2024-10-25

**Soundness:** 3
**Presentation:** 3
**Contribution:** 2
**Rating:** 5
**Confidence:** 4

**Summary:**

In this paper, the authors derive refined generalization bounds for the Deep Ritz Method (DRM) and Physics-Informed Neural Networks (PINNs), building on the work of Lu et al. (2021c in the manuscript).  The authors derive sharper bounds for the Poisson and Scrodinger's equations, and then present their modified framework within a multi-task learning perspective.

**Strengths:**

The strength of the paper is that the authors present mathematical analysis of physics-informed neural networks and similar (related) methods. The author's work fits within the current push for developing and growing the mathematical foundations of PINNs and related ideas.

**Weaknesses:**

The reviewer did not find any mathematical errors in the work, but was surprised that the authors did not point to the other mathematical works that exist (in the field).  The most well-known on the mathematical side would be Weinan E and collaborators:

https://link.springer.com/article/10.1007/s00365-021-09549-y

Weinan and others (some of which the authors reference part of their work, like:
https://arxiv.org/abs/2106.07539).

People have since built on Weinan's ideas:  A Deep Double Ritz Method (D
RM) for solving Partial Differential Equations using Neural Networks by Uriiarte et al., CMAME 2023.

**Questions:**

+ How does the analysis differ (substantially) from the work of Weinan E (paper above) and subsequent papers? In particular, when comparing the author's theoretical framework or results to Weinan's section 2 or the CMAME Section 3, can you highlight how your results/conclusions are different and what changes your work makes in terms of the results.

+ Clarify how the author's work is distinct from the CMAME paper, as an 'extension' of the Lu and Weinan E ideas?  Although not identical, can the authorrs provide details on how their work is novel and how it leads to improved or different results over those of Weinan E and/or the CMAME paper.

+ Please compare the framework with the multi-head (multitask) work of Karniadakis :  https://www.semanticscholar.org/paper/L-HYDRA%3A-Multi-Head-Physics-Informed-Neural-Zou-Karniadakis/ec7289f0cb03f0987a0f84391de278f83654bb09   (either argue how they are different or show numerical results).

**Details Of Ethics Concerns:**

No ethical issues

---

> ### Author Response · Authors · 2024-11-20
>
> Thank you for taking the time to review our article and for your valuable suggestions. Let us address your questions point by point.
>
> **Q1**: Differences with the Weinan's section 2 (i.e., ref [1]) and CMAME section 3 (i.e., ref [2]).
>
> **A1**: We have cited Weinan's work in line 238 and explained that our definition of the Barron space differs. Nevertheless, we acknowledge the lack of a detailed comparison with Weinan's work, i.e., [1]. Section 2 of [1] (and [3]) only presents approximation results under the $L^2$ norm, with a slow rate of $m^{- \frac{1}{2}}$, where $m$ is the width of a two-layer neural network. Although our definition of the Barron space differs from that in [1], our method can be naturally applied to [1], yielding approximation results under the $H^1$ norm with a faster rate, that is, $m^{-(\frac{1}{2}+\frac{1}{3d})}$. Regarding why we are able to achieve better approximation results, [1] employed the Monte Carlo method, which can only yield a slow approximation rate of $m^{- \frac{1}{2}}$. In contrast, we utilized Lemma 10. After applying Lemma 10, the challenge lies in estimating the metric entropy of the target function class in the $H^1$ norm, which we addressed using a new decoupling technique. Moreover, in addition to approximation error, we have also provided generalization error, while [1] only focuses on approximation error.
>
> As for comparing with Section 3 of [2], [2] does not provide any theoretical analysis regarding approximation results, focusing only on the experimental part. It may be interesting to consider the theoretical analysis of the Deep Double Ritz Method presented in [2].
>
> In summary, for the Barron space, we have provided a faster approximation rate in a higher-order Sobolev norm, and the associated constants are independent of the dimension. Moreover, our method can also be naturally extended to other Barron spaces defined differently.
>
> **Q2**: Novelty of the method for deriving the fast approximation rates.
>
> **A2**: [1] employed the Monte Carlo method to derive the approximation rates, hence could only obtain a slow rate of $m^{- \frac{1}{2}}$ in the $L^2$ norm. In contrast, we utilized Lemma 10 (line 2141-2150). As mentioned in A1, the difficulty lies in the estimation of the metric entropy under the $H^1$ norm of target function class, since it is not Lipschitz continuous with respect to the parameters. To address this challenge, we employed a novel decoupling technique, which involves splitting the estimation of the metric entropy in the $H^1$ norm into the independent estimation of the metric entropy in the $L^2$ norm. Specifically in the estimation of the metric entropy under the $H^1$ norm, for $(\omega _1,t _1),(\omega _2,t _2)\in \partial B _1^d(1)\times [-1,1]$, we have
> $$  || \sigma(\omega _1 \cdot x+t _1)-\sigma(\omega _2 \cdot x+t _2)|| _{H^1(\Omega)}^2  \leq 4(|\omega _1-\omega _2| _1^2+|t _1-t _2|^2)+2\int _{\Omega} |I _{ \{\omega _1 \cdot x+t _1 \geq 0\}}- I _{ \{\omega _2 \cdot x+t _2 \geq 0\} }|^2dx .     $$
> It is challenging to handle the first and second terms simultaneously due to the discontinuity of indicator functions, thus we turn to handle two terms separately.  Note that the first term is related to the covering of $\partial B _1^d(1)\times [-1,1] $ and the second term is related to the covering of a VC-class of functions. In this way, by dealing with these two terms separately, we can address this difficulty. For more details about the proof, one can refer to Proposition 8 starting from line 928. Moreover, our method can be naturally applied to [1], because the functions defined in the Barron space in [1] have a direct integral representation, which allows Lemma 10 from our paper to be directly applied.
>
> **Q3**: Comparison with [4].
>
> **A3**: In [4], the authors view solving different PDEs as different tasks and consider solving a single PDE as one task. However, in our article, when solving a single PDE, we consider minimizing the internal residual and the boundary residual as different tasks, thus our criteria for classifying different tasks differ from those in [4]. Furthermore, [4] does not provide theoretical analysis, while we offer detailed error analysis.
>
>
> **References**:
>
> [1]: W. E, C. Ma, and L. Wu. The Barron space and the flow-induced function spaces for neural
> network models. Constructive Approximation, 55(1):369–406, 2022.
>
> [2]: C. Uriarte, D. Pardo, I. Muga, J. Munoz-Matute, A deep double ritz
> method (d2rm) for solving partial differential equations using neural
> networks, Computer Methods in Applied Mechanics and Engineering
> 405 (2023) 115892.
>
> [3]: Chen, Z., Lu, J., and Lu, Y. On the representation of solutions to elliptic PDEs in Barron spaces. Advances in Neural Information Processing Systems, 34, 2021
>
> [4]: Zongren Zou and George Em Karniadakis. L-HYDRA: Multi-head physics-informed
> neural networks. arXiv preprint arXiv:2301.02152, 2023.

---

> ### Comment · Area_Chair_Co33 · 2024-11-26
>
> Please check if the authors' response addresses your concerns.

---

> > ### Author Response · Authors · 2024-12-01
> >
> > Dear Reviewer,
> >
> > We sincerely appreciate your time and effort in reviewing our work. We fully understand your schedule may be quite busy right now. As the deadline for the Author-Reviewer discussion period is approaching, we would greatly value the opportunity to engage in further discussion with you. We look forward to your feedback on whether our responses effectively address your concerns and if there are any additional questions or points you would like to discuss.
> >
> > Thank you again for your thoughtful consideration.
> >
> > Best regards,
> >
> > The Authors.

---

### Official Review · Reviewer_C4fn · 2024-10-29

**Soundness:** 3
**Presentation:** 2
**Contribution:** 2
**Rating:** 5
**Confidence:** 4

**Summary:**

This paper derived new generalization bounds for solving elliptic Partial Differential Equations (PDEs) via two deep learning based methods - the Deep Ritz Method (DRM) and Physics Informed Neural Network (PINN). For DRM, this paper obtained generalization bounds for the Poisson equation with Neuman boundary condition when the solution is in either some Barron space or some Sobolev space. For PINN, this paper considered linear second order elliptic PDEs with Dirichlet boundary condition. By utilizing results from statistical learning theory under the Multi-Task Learning (MTL) framework, the authors also obtained generalization bounds for PINN when the solution is in some Barron space or some Sobolev space. As a side product of the main results, this paper also obtained better rates of approximating functions in certain Barron spaces via two-layer neural networks.

**Strengths:**

1. This paper is presented in a clear way for readers to follow. Also, a thorough review of related work on variants of Rademacher complexity done by researchers from the statistical learning theory community is included. Proofs of essential lemmas are also presented to ensure the rigorousness of the entire manuscript.

2. In terms of specific contribution, this paper obtained finer bounds for approximating certain Barron functions via two layer neural networks, which lead to better generalization bounds on solving elliptic PDEs via DRM and PINN under the circumstances when the true solution is in some Barron space compared to existing work [1].

**Weaknesses:**

1. The reviewer's main concern is that some contributions mentioned in this article are not described in a clear way. Specifically, the reviewer thinks that it might be worthwhile for the authors to include a separate paragraph to compare the contributions made in this manuscript and existing work [2]. For specific questions about the main contributions and novelty of this paper compared to [2], the authors may refer to the first four bulletin points in the "Questions" section below.

2. Regarding presentation of this paper, there are a few grammatic issues that can be potentially resolved. For instance, the sentence from line 40 to 41 can be possibly rephrased as "The Deep Ritz method, on the other hand, incorporates the variational formulation into training the neural networks due to the widespread use of the variational formulation in traditional methods".

3. Given that solving PDEs via machine learning based methods is now a popular field, the authors might consider performing a brief literature review by citing a few important work (other than DRM and PINN) [3-9] in the first part of subsection 1.1 (Related Works) for the sake of completeness. One may refer to the related works section in [2] and [10] as possible examples.

**Questions:**

1. One thing the authors emphasized as a contribution is that this manuscript derived a generalization bound for the Poisson equation compared to [2]. However, it seems to the reviewer that Proposition 1, which provides a variational formulation of Poisson equation, actually requires extra assumption ($\int_{\Omega}f dx = 0$) compared to [2]. Hence, the review wonders whether this is a fair comparison and would appreciates it if the authors can elaborate on why they think the main reason is actually because the expectation of empirical loss is not equal to the population loss under the variational formulation (line 111-112, line 129-130).

2. In Remark 2, the authors state that the convergence rate associated with the DRM gets improved from $n^{-\frac{2k-2}{d+2k-4}}$ to $n^{-\frac{2k-2}{d+2k-2}}$, which seems to be incorrect as the second rate is larger than the first one. Also, the same rate (up to $\log n$ factors) has also been achieved in [2] via the peeling method. Therefore, the reviewer would appreciate it if the author could specify the major novelty/contribution of this paper for the DRM.

3. For the generalization bounds on PINN, the authors claimed that results presented in Section 4 are more general compared to [2] as this paper doesn't require strong convexity of the objective function. However, it seems to the reviewer that this cannot be claimed as a major novelty as Lemma 17 serves a similar role as Theorem B.2 in [2] (Also, here the authors still need the strict elliptic condition). Furthermore, even though the convergence rate $n^{-\frac{2k-4}{d+2k-4}}$ of PINN attained here is the same as that of [2], the norm used for measuring the error seems to be different - (30) implies that the norm used here is the $H^{\frac{1}{2}}$ norm, while [2] uses the $H^1$ norm. This will for sure influence the convergence rate and statistical optimality, so the reviewer would appreciate it if the authors could provide some intuition on the change of norms here.

4. Given that [2] provides not only upper bounds but also informational theoretical lower bounds on the expected estimation error, which certifies statistical optimality under certain regimes, would it be possible for the authors to provide some intuition on the lower bounds for the cases when the true solution is in some Barron spaces? Essentially speaking, are the bounds presented in (13) and (27) statistically optimal?

5. In the abstract and introduction (line 15-16 and line 125-127), the authors claimed that sharper generalization bounds are derived for solving both the Poisson equation and the Schrödinger equation via the DRM. However, it seems that Theorem 3 in Section 2 only contains results for the Poisson equation?

References:

[1] Lu, Y., Lu, J. and Wang, M., 2021, July. A priori generalization analysis of the deep Ritz method for solving high dimensional elliptic partial differential equations. In Conference on learning theory (pp. 3196-3241). PMLR.

[2] Lu, Y., Chen, H., Lu, J., Ying, L. and Blanchet, J., 2021. Machine learning for elliptic pdes: Fast rate generalization bound, neural scaling law and minimax optimality. In International Conference on Learning Representations, 2022. URL https://openreview.net/forum?id=mhYUBYNoGz.

[3] Sirignano, J. and Spiliopoulos, K., 2018. DGM: A deep learning algorithm for solving partial differential equations. Journal of computational physics, 375, pp.1339-1364.

[4] Han, J., Jentzen, A. and E, W., 2018. Solving high-dimensional partial differential equations using deep learning. Proceedings of the National Academy of Sciences, 115(34), pp.8505-8510.

[5] Khoo, Y., Lu, J. and Ying, L., 2021. Solving parametric PDE problems with artificial neural networks. European Journal of Applied Mathematics, 32(3), pp.421-435.

[6] Zang, Y., Bao, G., Ye, X. and Zhou, H., 2020. Weak adversarial networks for high-dimensional partial differential equations. Journal of Computational Physics, 411, p.109409.

[7] Chen, Y., Hosseini, B., Owhadi, H. and Stuart, A.M., 2021. Solving and learning nonlinear PDEs with Gaussian processes. Journal of Computational Physics, 447, p.110668.

[8] Li, Z., Kovachki, N., Azizzadenesheli, K., Liu, B., Bhattacharya, K., Stuart, A. and Anandkumar, A., 2020. Fourier neural operator for parametric partial differential equations. arXiv preprint arXiv:2010.08895.

[9] Lu, L., Jin, P., Pang, G., Zhang, Z. and Karniadakis, G.E., 2021. Learning nonlinear operators via DeepONet based on the universal approximation theorem of operators. Nature machine intelligence, 3(3), pp.218-229.

[10] Lu, Y., Blanchet, J. and Ying, L., 2022. Sobolev acceleration and statistical optimality for learning elliptic equations via gradient descent. Advances in Neural Information Processing Systems, 35, pp.33233-33247.

---

> ### Author Response · Authors · 2024-11-21
> **The first part of the reply.**
>
> Thank you for taking the time to review our article and for your insightful comments and suggestions. Let us address your questions point by point.
>
> **Q1**: Comparison with [2] on the Deep Ritz Method aspect.
>
> **A1**: First, the results of [2] are unrealistic, meaning there is still a gap between theory and practice. Specifically, the strong convexity of the DRM objective in Proposition 2.1 of [2] shows that for any $u\in H _0^1(\Omega)$,
> $$ \mathcal{E}^{DRM}(u)-\mathcal{E}^{DRM}(u^{* })\lesssim ||u-u^{* }|| _{H^1}^2 \lesssim \mathcal{E}^{DRM}(u)-\mathcal{E}^{DRM}(u^{* }), $$
> where $u^{* }\in H _0^1(\Omega)$ is the true solution and $\mathcal{E}^{DRM}$ is the Ritz functional. However, this only holds for the functions in $H _0^1(\Omega)$ and the neural network functions fail to belong to $H_0^1(\Omega)$. If we force the neural network functions to belong to $H _0^1(\Omega)$, for example, by multiplying them with a function that vanishes on the boundary, then the approximation results in [2] no longer hold. In this article, we consider the zero Neumann boundary conditions instead of the zero Dirichlet boundary conditions in [2]. The assumption that $\int _{\Omega}fdx=0$ is made to ensure the existence of the solution under the zero Neumann boundary condition. In fact, when dealing with zero Dirichlet boundary conditions, we would prefer to add a boundary penalty term, that is, to consider the following loss function:
> $$ \mathcal{E}^{DRM}(u)=\int _{\Omega} ||\nabla u|| _2^2+V|u|^2dx-2\int _{\Omega}fudx+\lambda \int _{\partial \Omega} u^2ds. $$
> Then combining this with our Section D.2 (line 2845-2871), which discusses the non-zero Neumann boundary condition and the results in [3], we may achieve a better generalization bound that is more aligned with reality under the zero Dirichlet boundary condition.
>
> Second, the method in [2] for deriving fast rates does not work for the setting of Poisson equation, since the expectation of empirical loss is not equal to the variational formulation. This fact also limits the use of some popular methods, such as the local Rademacher complexity. As for why the method in [2] and the method in the original paper of local Rademacher complexity (i.e., [5]) are not feasible, for example, the core lemma in [2], i.e., Lemma B.4, requires the condition (from [5]) that
> $$R _n(\{f\in \mathcal{F}| \mathbb{E}[f]\leq r\}) \leq \phi(r),$$
> where $R _n$ is the empirical Rademacher complexity and $\phi$ is a sub-root function. In the subsequent proof of [2], an appropriate class of functions class $\mathcal{F}$ can be chosen such that $\mathbb{E}[f] = \mathcal{E}^{DRM}(u)-\mathcal{E}^{DRM}(u^{* })$ for some $u$. By doing so, the strong convexity of the Ritz functional can be utilized. However, in the setting of Poisson equation, the Ritz function cannot be expressed in the form of an expectation, i.e., there does not exist a function class $\mathcal{F}$ such that $\mathbb{E}[f] = \mathcal{E}^{DRM}(u)-\mathcal{E}^{DRM}(u^{* })$. Therefore, in this work, we use a new error decomposition method and new peeling method to derive the fast rates.
>
> **Q2**: Contribution of this paper to the DRM
>
> **A2**: We apologize for the error in Remark 2. In fact, it should be "the convergence rate associated with the DRM improves from $n^{-\frac{2k-2}{d+4k-4}}$ to $n^{-\frac{2k-2}{d+2k-2}}$." We have corrected this in the revised version, and thank you for pointing out this error. Regarding the contributions of this paper to DRM, in Answer 1 we have already explained the difficulties in dealing with the Poisson equation and our new method. Although this method also works for the static Schrödinger equation, it is too complicated. On the other hand, the case of the static Schrödinger equation represents a large class of important problems with strongly convex properties like equation (6), such as the $L^2$ regression problem under bounded noise in [1]. In fact, the methods used for the Poisson equation in this article, as well as those in [2] and [1], are applicable to the class of problems mentioned above. However, these methods are too complicated. In this article, by combining a new error decomposition with the use of local Rademacher complexity, we provide a simpler method for solving this class of problems. Finally, we also discuss other boundary conditions for the DRM (see Section D.2, line 2845-2871) and show that our methods can be extended to a broader class of PDEs. Moreover, in the Discussion section starting on line 2873, we investigate the complexity of over-parameterized two-layer neural networks when approximating functions in Barron space and demonstrate meaningful generalization errors in the setting of over-parameterization.

---

> > ### Author Response · Authors · 2024-11-21
> > **The remaining part of the reply.**
> >
> > **Q3**: About the norm used for the generalization bounds of PINNs.
> >
> > **A3**: Although the results in [2] are presented in the $H^2$ norm, our results are in the $H^{\frac{1}{2}}$ norm, as mentioned in Answer 1, the results in [2] require a strongly convex structure, which only holds for functions in $H _0^1$. This makes their results unrealistic. Secondly, the scenario in [2] is single-task, while ours is multi-task. This is because, within the framework of PINNs, [2] only minimizes the internal residual.
> >
> > Regarding the strong convexity of the objective function, Lemma 17 does not ensure it, unlike Theorem B.2 in [2]. Lemma 17 can only yield that
> > $$ ||u-u^{* }|| _{H^{\frac{1}{2} }(\Omega)}^2\lesssim L(u)-L(u^{* }) \lesssim ||u-u^{* }|| _{H^2(\Omega)}^2.$$
> > Note that the left side is the $H^{\frac{1}{2}}$ norm while the right side is the $H^2$ norm, and this is not the strong convexity we need. Next, we will explain why the strict elliptic condition is necessary and why our conclusion is in terms of the $H^{\frac{1}{2}}$ norm. It should be noted that for PINNs, we have provided two forms of generalization bounds. One is the generalization bound for the PINN's loss (equations 27 and 29 in Theorem 6), which does not require the strict elliptic condition or the strong convexity. The other is the generalization bound in the $H^{\frac{1}{2}}$ norm, which is a direct corollary of Theorem 6 and Lemma 17. Lemma 17, however, requires the strict elliptic condition as well as certain smoothness of the boundary. Regarding why we can only provide an error estimate in the $H^{\frac{1}{2}}$ norm, a weak Sobolev norm, it is because we follow the original PINNs framework, where both the internal residual and boundary residual are measured using the $L^2$ norm. As discussed between lines 468-482 in this article, if we use the loss function in equation (32) (line 478), we might obtain results in a stronger Sobolev norm. For the choice of PINN's loss, reference can also be made to article [4].
> >
> > **Q4**: Missing the lower bounds.
> >
> > **A4**: We are aware of the significance of statistical optimality in the context of Barron spaces and we will address this in our future research.
> >
> > **Q5**: The results for the static Schrödinger equation via the DRM.
> >
> > **A5**: Apologies, due to page limitations and the similarity between the conclusions of the static Schrödinger equation and the Poisson equation, we have placed it in the appendix. Please refer to Theorem 9 (lines 1288-1310) for details.
> >
> > **For the weaknesses**: We acknowledge the need for a more explicit comparison between our work and existing literature, particularly in relation to [2]. To address this, we will include a dedicated paragraph in the "Related Works" section that contrasts our contributions with those in [2]. This will highlight the novel aspects of our research and how it advances the field. Thank you for your feedback on the grammatical aspects of our paper. We will carefully review and address these issues to ensure the presentation is as clear and polished as possible. Additionally, we appreciate your recommendation to expand our literature review. We will include key works beyond DRM and PINNs in the "Related Works" section, ensuring a comprehensive overview of the field.
> >
> > **References**:
> >
> > [1]: Farrell, M., T. Liang, and S. Misra (2021). Deep neural networks for estimation and inference. Econometrica 89, 181–213
> >
> > [2]: Lu, Y., Chen, H., Lu, J., Ying, L. and Blanchet, J., 2021. Machine learning for elliptic pdes: Fast rate generalization bound, neural scaling law and minimax optimality. In International Conference on Learning Representations, 2022.
> >
> > [3]: Muller, J. and Zeinhofer, M., Error Estimates for the Deep Ritz Method with Boundary Penalty,  Proceedings of Mathematical and Scientific Machine Learning, B. Dong, Q. Li, L. Wang, and Z.Q.J. Xu, Eds., Vol. 190 of Proceedings of Machine Learning
> > Research, PMLR, pp. 215–230, 2022.
> >
> > [4]: Chuwei Wang, Shanda Li, Di He, and Liwei Wang. Is $L^2$ physics informed loss always suitable for training physics-informed neural network? Advances in Neural Information Processing Systems, 35:8278–8290, 2022.
> >
> > [5]: P. L. Bartlett, O. Bousquet, and S. Mendelson. Local Rademacher complexities. The Annals of Statistics, 33(4):1497–1537, 2005.

---

> ### Comment · Area_Chair_Co33 · 2024-11-26
>
> Please check if the authors' response addresses your concerns.

---

> ### Comment · Reviewer_C4fn · 2024-11-28
>
> The reviewer would like to thank the authors for their detailed response. However, it seems to the reviewer that current organization of the whole manuscript still seems to be a bit disorganized and disconnected, especially for the "Related Works" section. This makes it hard for the readers to tell the main contribution and focus of this paper. Therefore, the reviewer is inclined to maintain the score of 5 for now.

---

> > ### Author Response · Authors · 2024-12-01
> >
> > Thank you very much for your feedback. We understand your concerns about the organization and clarity of our paper, particularly in the "Related Works" section, and we agree that a well-structured manuscript is crucial for conveying our research contributions effectively. In the revised version, we will supplement the "Related Works" section with other machine learning-based methods, such as the ones you mentioned [3-9].

---

### Official Review · Reviewer_sz2Y · 2024-11-01

**Soundness:** 2
**Presentation:** 1
**Contribution:** 1
**Rating:** 3
**Confidence:** 3

**Summary:**

The paper theoretically analyzes the generalization error of two prominent deep learning methods for solving PDEs: the Deep Ritz method and PINNs. The analysis is conducted on simple second-order linear PDEs, and under certain assumptions, they obtain tighter bounds compared to previous studies. However, the assumptions made are somewhat impractical, and the results raise questions about the significance of this research beyond its mathematical implications and what it contributes to the field of deep learning.

**Strengths:**

The authors have performed a mathematical analysis of the Deep Ritz method and PINNs, which continue to attract considerable interest. This mathematical analysis can aid in understanding the characteristics of the models.

**Weaknesses:**

A robust mathematical analysis can be more powerful than empirical observations. However, there are several areas of concern in this study:

1.	What is the intention behind studying the two methods using different mathematical tools for different PDEs? There is no connection drawn between the two methods within the paper, which leads to confusion about the main message of the study.

2.	Due to the complexity of deep learning models, it is a natural approach to start the analysis with simpler scenarios. However, the assumption that the solutions of the PDEs lie within the Barron space is too strong. The Barron space is specifically designed for functions that are best approximated by two-layer MLPs and is suitable for analyzing two-layer neural network structures, but it is too narrow to represent the space of PDE solutions.

3.	The primary goal of Deep Ritz and PINNs is not just to minimize the loss but to find the solution to PDEs. Thus, instead of focusing on how much a minimizer of the empirical loss reduces the expectation loss, the analysis should focus on how close the empirical loss minimizer is to the true solution. It is crucial because both methods include unbounded derivative operators in their loss functions, meaning that a small difference in loss values does not necessarily imply that the functions are close. Although the paper briefly addresses this in equations (16) and (30), it does so under very restrictive assumptions, casting doubt on the implications of the results.

4.	While the theoretical analysis uses the $ReLU^k$ activation function (also known as RePU) for convenience, this also has limitations. The theory is developed in the context of simple two-layer networks, but practical implementations often involve deeper networks, where the power of the RePU increases with depth, leading to floating-point precision issues. This limitation should be acknowledged in the paper.
5.	The PDEs considered in this study are too simple. Such simple PDEs can be solved much faster and more accurately using classical methods such as FEM, FDM, Discrete Galerkin, and FVM. Deep learning-based methods should focus on more complex PDEs or inverse problems. While it is important to start from a mathematical understanding of simpler problems, it is also important to recognize that these methods are not just mathematical objects. Research in this area must integrate PDEs, classical numerical methods, and deep learning, rather than excluding any one aspect. Furthermore, the significance of this research in the field of deep learning remains unclear beyond its mathematical implications.

6.	Eq (16) combines the results of Prop 1 and Thm 3, but it appears to omit the Poincaré constant from Prop 1, which is dimension-dependent.


7.	Limitations are not addressed in the paper.

**Questions:**

See weakness above.

---

> ### Author Response · Authors · 2024-11-20
> **The first part of the reply**
>
> Thank you for taking the time to review our article. Let us address your questions point by point.
>
> **Q1**: What is the intention behind studying the two methods using different mathematical tools for different PDEs?
>
> **A1**: We consider these two methods, DRM and PINNs, because they are both very popular for solving PDEs using neural networks. The reason we used different approaches for the generalization bounds of these methods is due to the nature of each method. Section 2 deals with DRM, which involves only one task and has strong convexity, while Section 3 deals with PINNs, which involves two tasks and lacks strong convexity. Of course, the methods for handling DRM and PINNs can also be combined, as demonstrated in Section D.2, starting on line 2844, where we considered non-zero Neumann conditions.
>
> **Q2**: The assumption that the solutions of the PDEs lie within the Barron space is too strong.
>
> **A2**: In this article, we consider not only the case where the solutions of the PDEs lie in the Barron space but also the case where the solutions belong to the general Sobolev space. The consideration of the Barron space scenario is aimed at demonstrating that the Deep Ritz Method (DRM) and Physics-Informed Neural Networks (PINNs) do not suffer from the curse of dimensionality in this context. That is, the constants in the generalization bounds are at most polynomially related to the dimension, and the convergence rate is almost independent of the dimension. In fact, higher-order Sobolev spaces can be embedded in Barron spaces, for example, $H^{s+\frac{d}{2}+\epsilon}(\Omega) \subset \mathcal{B}^{s}(\Omega)$ for any $s>0, \epsilon>0$ and for any bounded domain $\Omega\subset \mathbb{R}^d$. Additionally, [1] and [2] have considered the solution theory of certain PDEs in Barron spaces,  demonstrating when the solutions of certain PDEs are in the Barron space.
>
> **Q3**: Assumptions are restrictive when deriving the generalization bounds under certain Sobolev norms.
>
> **A3**: For the DRM, we only require boundedness and strong convexity. The assumption of boundedness is important in the field of machine learning for deriving better generalization bounds. Regarding the assumption of strong convexity, the conditions required by the traditional Ritz method can lead to strongly convex structures similar to equations (4) and (6). Specifically, let $H$ be a real Hilbert space with norm denoted by $||\cdot|| _{H}$ and let $\langle \cdot, \cdot \rangle$ denote the inner product. Let $a(u, v)$ be a real bilinear form on $H\times H$, $f$ be an element of $H$ and consider the following problem: find $u\in H$ such that
> $$ a(u, v) = \langle f, v\rangle, \ \forall v \in H.$$
> When the conditions of the Lax-Milgram theorem are met, that is $|a(u,v)|\leq M||u||  _{H}||v|| _{H}$ for any $u,v \in H$ and $a(v,v)\geq \alpha ||v|| _{H}^2$, and $a$  is symmetric, we have that the solution $u^{* }$ of the above variational problem is also the only element of $H$ that minimizes the following quadratic functional (also called energy functional) on $H$:
> $$ J(u)= a(u,u)-2\langle f, u\rangle.$$
> At this point, we naturally have for any $u\in H$,
> $$ ||u-u^{* }|| _{H}^2 \lesssim J(u)-J(u^{* }) \lesssim ||u-u^{* }|| _{H}^2,$$
> since $J(u)-J(u^{* })=a(u-u^{* }, u-u^{* })$, this is exactly the strongly convex structure we need. For the PINNs, equation (30) only requires the strict elliptic condition and smoothness of the boundary. In fact, these assumptions are not restrictive.
>
> **Q4**: Limitation of RePU activation functions.
>
> **A4**: Thank you for highlighting the limitations of using the RePU activation function in our theoretical analysis. We acknowledge that in deeper networks, the power of RePU increases with depth, which can lead to floating-point precision issues. We will point out this limitation in the paper.
>
> **Q5**: This article considers PDEs that are too simple and traditional methods can solve them faster and more accurately.
>
> **A5**: Classical methods may suffer from the curse of dimensionality when solving high-dimensional PDEs. In contrast, part of our results are meaningful in high dimensions as well. As we mentioned in answer 3, the results regarding DRM in the article can be generalized to a broader class of PDEs. The strong convexity required can naturally be derived from the conditions of the Lax-Milgram theorem, which are also needed by the traditional Ritz method. Similarly, for PINNs, our conclusions do not require strong conditions, so they can be naturally extended to other types of PDEs.
>
> **Q6**: The Poincaré constant is dimension-dependent.
>
> **A6**: The Poincaré constant of the domain $[0, 1]^d$ is dimension-independent, since $[0,1]^d$ is a space of tensor-product form. Please see Theorem 4.2 in [3] or the content in line 1223-1232 of this paper for more details.

---

> > ### Author Response · Authors · 2024-11-20
> > **The remaining part of the reply.**
> >
> > **Q7**: Limitations are not addressed in the paper.
> >
> > **A7**: Apologies, in the revised version, we will add a section to describe the organization of this paper. In fact, we have discussed the limitations of this work in Section D.3, starting from line 2872. In this section, we outline the constraints and challenges encountered during our study, providing a comprehensive view of the scope and applicability of our findings.
> >
> > **References**:
> >
> >
> > [1]: Chen, J. Lu, Y. Lu and S. Zhou (2023), A regularity theory for static schrodinger equations on in spectral barron spaces, SIAM Journal on Mathematical Analysis 55(1), 557–570.
> >
> > [2]: Lu, Y., Lu, J., and Wang, M., A Priori Generalization Analysis of the Deep Ritz Method for Solving High Dimensional Elliptic Partial Differential Equations, Conference on Learning Theory, PMLR, pp. 3196–3241, 2021.
> >
> > [3]: Bonnefont, M. Poincaré inequality with explicit constant in dimension d ≥ 1, 2022. Global Sensitivity Analysis and Poincaré Inequalities (Summer school, Toulouse).

---

> ### Comment · Area_Chair_Co33 · 2024-11-26
>
> Please check if the authors' response addresses your concerns.

---

> ### Comment · Reviewer_sz2Y · 2024-11-26
> **Thank you for your answer**
>
> I appreciate the authors' efforts in addressing the concerns I raised. However, I am still unclear about the main focus and contributions of the paper. The discussions on DRM, PINNs, and the expressivity of neural networks seem somewhat disconnected, making it difficult to discern the central message of the paper.
>
> While mathematical analysis of deep learning models is important, I would like to remind you that the majority of readers of the ICLR proceedings may not be mathematically inclined. With this in mind, I remain uncertain about the key takeaway of this study for the broader ICLR audience. DRM and PINNs have been explored in previous works, so it would be helpful to clarify what new insights and contributions your work offers over existing research, particularly for the broader deep learning community. I am also unclear about how your mathematical contributions provide a useful tool compared to existing research.
>
> You also mention that the model addresses the curse of dimensionality. However, I believe this claim may be premature. Deep learning models involve complex interdependencies between network expressiveness, optimization, and generalization. It is not entirely clear that generalization bounds alone can be used to definitively argue that your model resolves the curse of dimensionality. For instance, the convergence of gradient descent in PINNs is known to be exponentially dependent on the dimension (C.H.Song at el., How does PDE order affect the convergence of PINNs?, NeurIPS, 2024.)
>
> I believe it would be helpful to clearly position your work within the broader context of deep learning-based PDE solvers and to clarify its implications for general advancements in deep learning research. As of now, I still have some reservations regarding the paper's contribution to the field and its potential impact on the broader community.

---

> > ### Author Response · Authors · 2024-11-27
> > **Thank you for your feedback**
> >
> > Thank you for your feedback. We hope the answers below can address your concerns. If you have any further concerns, we would be more than happy to address them as well.
> >
> > **Q1**: What is the connection between of the discussions on DRM, PINNs, and the expressivity of neural networks?
> >
> > **A1**: The goal of DRM and PINNs is to ensure that the empirical loss minimizer can get sufficiently close to the solution, and this depends on the approximation properties of neural networks. Furthermore, when considering the generalization bounds of DRM and PINNs in this article, we also need to utilize the approximation results of neural networks under some Sobolev norm.
> >
> > **Q2**: What are the new insights and contributions of this work for the broader machine learning community?
> >
> > **A2**: The primary purpose of this paper is to provide better generalization bounds (i.e., fast rates) for DRM and PINNs compared to existing work. Deriving fast rates has been a hot topic and a challenging issue in the field of machine learning theory.  When considering the Poisson equation, the situation is different from most cases in machine learning because the expectation of empirical loss is not equal to the Ritz functional. Thus, we need to employ new peeling techniques to derive the fast rates. As for the static Schrödinger equation, the scenario represents a large class of problems with a similar strongly convex structure like equation (6) (line 199), and for such problems, we provide a simple and effective theoretical framework to achieve better generalization bounds. In the section on PINNs, we have enhanced the Talagrand concentration inequality for multi-task scenarios using the entropy method, which we believe will benefit the field of multi-task learning. Lastly, we discussed the generalization bounds in the over-parameterized scenario, providing meaningful generalization bounds in certain cases, which is still a challenging issue in the field of deep learning.
> >
> > In summary, our paper introduces new or simpler methods for deriving fast rates within the field of machine learning theory. Concurrently, it provides novel insights into the theories of PINNs and DRM.
> >
> >
> > **Q3**: The issue of the curse of dimensionality.
> >
> > **A3**: In scenarios where the solutions to PDEs belong to the Barron space, we considered only the approximation error and generalization error without taking into account the optimization error, and asserted that DRM and PINNs can overcome the curse of dimensionality. This is because considering these three types of errors simultaneously has always been a challenging problem in the field of machine learning and also a focus of our future research.
> >
> > Based on the neural tangent kernel (NTK) framework, [1] takes into account the order of PDEs and the dimension, and provides conditions for the convergence of gradient descent in training PINNs. Regarding the dependency on dimension, the result in Theorem 3.2 of [1] is at most $\mathcal{O}((C _{d+k}^k)^{14})$, where $k$ is the order of the PDE. Therefore, when $k$ is independent of $d$, the result is polynomially dependent on $d$. Furthermore, the method in [1] can only obtain the training error under the over-parameterized scenario, but cannot derive the generalization error. Since Theorems 3 and 6 in our work consider scenarios that are not over-parameterized, this requires new methods to analyze the training error.
> >
> > **Q4**: What is the potential impact of this work on the broader community?
> >
> > **A4**: In addition to the impact on the field of machine learning theory that we mentioned in answer 2, our approach to handling PINNs can be naturally extended to variations of PINNs, such as the deep least-squares methods discussed in [2].
> >
> >
> > **References**:
> >
> > [1]: C.H.Song at el., How does PDE order affect the convergence of PINNs?, NeurIPS, 2024.
> >
> > [2]: Cai at el., Deep least-squares methods: An unsupervised learning-based numerical method for solving elliptic PDEs, Journal of Computational Physics, 2020

---

> > > ### Author Response · Authors · 2024-12-01
> > >
> > > Dear Reviewer,
> > >
> > > We sincerely appreciate your time and effort in reviewing our work. We fully understand your schedule may be quite busy right now. As the deadline for the Author-Reviewer discussion period is approaching, we would greatly value the opportunity to engage in further discussion with you. We look forward to your feedback on whether our responses effectively address your concerns and if there are any additional questions or points you would like to discuss.
> > >
> > > Thank you again for your thoughtful consideration.
> > >
> > > Best regards,
> > >
> > > The Authors.

---

### Official Review · Reviewer_5WcL · 2024-11-04

**Soundness:** 3
**Presentation:** 3
**Contribution:** 3
**Rating:** 5
**Confidence:** 4

**Summary:**

Authors utilize the localized analysis to refine generalization bounds for both the Deep Ritz Method (DRM) and Physics-Informed Neural Networks (PINNs). For the DRM, authors provide theoretical results for the Poisson and Schrödinger equations. For PINN, the theoretical results obtained for the linear second order elliptic equation.

**Strengths:**

* The proposed theoretical results show that the function $\mathcal{B}^2$ can be well approximated in the $H^1$ norm by MLP (two hidden layers with ReLU as the activation function) network with controlled weights.
* For the DRM, authors obtain more precise generalization bounds for the Poisson and Schrödinger equations with Neumann boundary condition, irrespective of whether the solutions are in Barron spaces or Sobolev spaces.
* For the PINNs, authors provide a generalization error for PINN loss of the linear second order elliptic equation in the $H^{1/2}$ norm.
* Theoretical estimations for both methods with neural networks are correct for both cases if the PDE solution is in the Barron space or the Sobolev space.

**Weaknesses:**

* Considering PDEs(the Poisson equation and the static Schrödinger equation with Neumann boundary condition) defined on d-dimensional cube are simple and not interested in community. Is it possible to extend the list of PDEs where proposed theory can be used?
* It will be good for understanding to add some numerical experiments.

**Questions:**

See the weaknesses.

---

> ### Author Response · Authors · 2024-11-20
>
> Thank you for taking the time to review our article and for your recognition of it. Let us address your questions point by point.
>
> **Q1**: Extension to other PDEs for the Deep Ritz Method (DRM).
>
> **A1**: We first explain why we consider these two simple PDEs, and then illustrate how to generalize our results to more general PDEs. When considering the Poisson equation, the situation is different from most cases in the field of machine learning, because the expectation of empirical loss is not equal to the Ritz functional. Therefore, we need to employ new peeling techniques to obtain better generalization bounds. As for the static Schrödinger equation, the scenario represents a large class of problems with a similar strongly convex structure like equation (6) (line 199), and for such problems, we provide a simple and effective theoretical framework to achieve better generalization bounds.
>
> Then we will demonstrate how our theory can be extended to a broader range of PDEs. In the discussion section starting on line 2844, we have illustrated how to extend our results to non-zero Neumann boundary condition. And for other boundary conditions, our results can also be similarly extended. As for extending to other PDEs, note that our method relies on two essential conditions, one is the boundedness condition and another is the strongly convex structure (see line 188 or line 199). For strongly convex structure, it is not only the Poisson equation and the static Schrödinger equation that meet the conditions. In fact, even the traditional Ritz method has similar requirements, which stem from the need to satisfy the Lax-Milgram theorem. Specifically, let $H$ be a real Hilbert space with norm denoted by $||\cdot|| _{H}$ and let $\langle \cdot, \cdot \rangle$ denote the inner product. Let $a(u, v)$ be a real bilinear form on $H\times H$, $f$ be a element of $H$ and consider the following problem: find $u\in H$ such that
> $$ a(u, v) = \langle f, v\rangle, \ \forall v \in H.$$
> When the conditions of the Lax-Milgram theorem are met, that is $|a(u,v)|\leq M||u|| _{H}||v|| _{H}$ for any $u,v \in H$ and $a(v,v)\geq \alpha ||v|| _{H}^2$, and $a$  is symmetric, we have that the solution $u^{* }$ of the above variational problem is also the only element of $H$ that minimizes the following quadratic functional (also called energy functional) on $H$:
> $$ J(u)= a(u,u)-2\langle f, u\rangle.$$
> Hence, for any $u\in H$, we naturally have
> $$ ||u-u^{* }|| _{H}^2 \lesssim J(u)-J(u^{* }) \lesssim ||u-u^{* }|| _{H}^2,$$
> since $J(u)-J(u^{* })=a(u-u^{* }, u-u^{* })$, and this is exactly the strongly convex structure we need. As a simple example, we can replace the Laplacian operator in the static Schrödinger equation with operator $A$, satisfying
> $$Au(x)= \sum\limits _{i,j=1}^{d}\frac{\partial}{\partial x _j} (a _{ij}\frac{\partial u}{\partial x _i}(x) ),$$
> where $a _{i,j}\in L^{\infty}(\Omega)$, $a _{ij}=a _{ji}$, the matrix ($a _{i,j}$) is uniformly (a.e.) positive definite.
>
> In summary, the assumption of strong convexity is not overly restrictive, as even the traditional Ritz method requires it.
>
> **Q2**: No numerical experiments.
>
> **A2**: Apologies, this article primarily focuses on the theoretical aspects; in the future, we plan to design efficient algorithms to validate our conclusions.

---

> ### Comment · Area_Chair_Co33 · 2024-11-26
>
> Please check if the authors' response addresses your concerns.

---

> > ### Author Response · Authors · 2024-12-01
> >
> > Dear Reviewer,
> >
> > We sincerely appreciate your time and effort in reviewing our work. We fully understand your schedule may be quite busy right now. As the deadline for the Author-Reviewer discussion period is approaching, we would greatly value the opportunity to engage in further discussion with you. We look forward to your feedback on whether our responses effectively address your concerns and if there are any additional questions or points you would like to discuss.
> >
> > Thank you again for your thoughtful consideration.
> >
> > Best regards,
> >
> > The Authors.

---

### Meta-Review · Area_Chair_Co33 · 2024-12-18

**Metareview:**

The manuscript studies the generalization error of machine learning based solvers for partial differential equations. While the work seems solid, overall the reviewers find the novelty marginal compared with existing literature. After reading the manuscript, the metareviewer agrees with the majority of the reviewer comments. The manuscript is probably better suited for a venue more focused on numerical solutions to PDEs.

**Additional Comments On Reviewer Discussion:**

The authors have tried to address most of the comments by the reviewers, in particular regarding literature. However, some concerns of the reviewers remain.

---

### Decision · Program_Chairs · 2025-01-22

Reject